# A FOUNDATION MODEL WITH MULTI-VARIATE PARALLEL ATTENTION TO GENERATE NEURONAL ACTIVITY

**Francesco S. Carzaniga** *
IBM Research – Zurich
Bern University Hospital

**Michael Hersche**
IBM Research – Zurich

**Abu Sebastian**
IBM Research – Zurich

**Kaspar Schindler**
Bern University Hospital

**Abbas Rahimi**
IBM Research – Zurich

## ABSTRACT

Learning from multi-variate time-series with heterogeneous channel configurations remains a fundamental challenge for deep neural networks, particularly in clinical domains such as intracranial electroencephalography (iEEG), where channel setups vary widely across subjects. In this work, we introduce multi-variate parallel attention (MVPA), a novel self-attention mechanism that disentangles content, temporal, and spatial attention, enabling flexible, generalizable, and efficient modeling of time-series data with varying channel counts and configurations. We use MVPA to build MVPFormer, a generative foundation model for human electrophysiology, trained to predict the evolution of iEEG signals across diverse subjects. To support this and future efforts by the community, we release the SWEC iEEG dataset, the largest publicly available iEEG dataset to date, comprising nearly 10,000 hours of recordings from heterogeneous clinical sources. MVPFormer leverages MVPA to achieve strong generalization across subjects, demonstrating expert-level performance in several iEEG tasks. MVPFormer surpasses state-of-the-art (SOTA) Transformer baselines in seizure detection across the SWEC, the MAYO, and the FNUSA datasets, while also achieving SOTA performance on four Brain TreeBank iEEG decoding tasks (volume, pitch, onset, and speech). We further validate MVPA on standard time-series forecasting and classification tasks, where it matches or exceeds the performance of existing attention-based models. Together, our contributions establish MVPA as a general-purpose attention mechanism for heterogeneous time-series and MVPFormer as the first open-source, open-weights, and open-data iEEG foundation model with SOTA clinical performance. The code is available at `https://github.com/IBM/multi-variate-parallel-transformer`. The SWEC iEEG dataset is available at `https://huggingface.co/datasets/NeuroTec/SWEC_iEEG_Dataset`.

## 1 INTRODUCTION

The increasing availability of multi-variate time-series data across domains, from financial data to sensor networks to clinical recordings, has driven demand for general-purpose neural architectures capable of learning from such data (Nie et al., 2023; Jin et al., 2024; Wang et al., 2024b; Guetschel et al., 2024). A fundamental challenge in this setting is channel heterogeneity: different sensors (or channels) often carry information that is both structurally and semantically non-uniform, while the number and the location of channels may vary across instances. This is especially pronounced in intracranial electroencephalography (iEEG; Nunez & Srinivasan 2006), where each subject's electrode layout is unique and tailored to clinical needs. iEEG models (Kuhlmann et al., 2018; Cho & Jang, 2020; Thuwajit et al., 2022; Wang et al., 2023; Saab et al., 2024) often require subject-specific adaptation to account for new setups, yet they still struggle to generalize. Consequently, effec-

---

*Correspondence to `frc@zurich.ibm.com`.

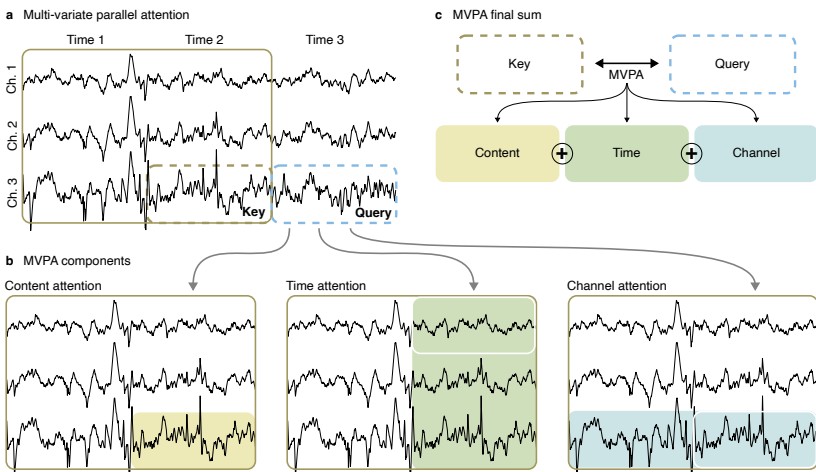

Figure 1: **Multi-variate parallel attention (MVPA). (a)** The input signal is divided into temporal and spatial segments. Each query-key interaction is computed for all keys within a local window. **(b)** MVPA decomposes attention into three components: content-based, computed per segment without positional encoding; time-based, shared across channels and dependent only on temporal distance; and channel-based, shared across time steps and dependent only on spatial distance. **(c)** The final attention is the sum of these three components, each capturing a distinct aspect of the data.

tive learning from multi-variate time-series demands models that are flexible and channel-agnostic, without sacrificing locality or the ability to generalize.

In this work, we introduce multi-variate parallel attention (MVPA, Figure 1), a novel self-attention mechanism addressing the structural challenges of channel heterogeneity. MVPA decomposes attention into three components: content-, time-, and channel-based components. Thus, it allows the model to separately learn the semantics of the signal, its temporal dynamics, and spatial (inter-channel) structure. MVPA enables flexible and efficient processing of time-series data, without relying on fixed channel positions or global positional encodings.

To highlight MVPA's ability to handle heterogeneous and clinically relevant time-series, we apply it to the particularly challenging domain of iEEG. Indeed, as mentioned above iEEG recordings present an ideal testbed for models designed to handle variable multi-channel structure. We use MVPA to build MVPFormer, a foundation model for human electrophysiology trained via generative pre-training to predict the evolution of brain signals.

MVPFormer is trained on the SWEC iEEG dataset, the largest available iEEG corpus to date with nearly 10,000 hours of multi-channel recordings (or 540,000 channel-hours), collected over a decade in clinical settings and made publicly available as part of this work. Using this long-term ictal iEEG dataset, we show that MVPFormer not only models neuronal activity during both normal and ictal states, but also generalizes across patients zero-shot within the same task, and outperforms previous approaches on clinically relevant benchmarks. At the same time, MVPFormer also enables diverse downstream applications through finetuning, including seizure detection on multiple other institutional datasets (Nejedly et al., 2020) and four iEEG decoding tasks from the Brain TreeBank dataset (Wang et al., 2024a). Remarkably, MVPFormer surpasses an equivalent purely discriminative version which has not undergone generative pre-training, strengthening the validity of foundation models in iEEG.

We further evaluate MVPA on classical time-series benchmarks, including ETTh and Weather (Zhou et al., 2021; Wu et al., 2021) for forecasting and EthanolConcentration, FaceDetection, and others (Liu & Wang, 2024) for classification. Here, MVPA matches or outperforms state-of-the-art (SOTA) models. These results establish MVPA as a competitive attention mechanism for general-purpose time-series beyond iEEG.

Our contributions are: (1) **Multi-variate parallel attention (MVPA)**, a novel self-attention mechanism that separately attends to content, temporal, and spatial structure, enabling strong generaliza-

tion across multi-variate time-series with heterogeneous channels; (2) **MVPFormer**, a foundation model for human electrophysiology, powered by MVPA and trained on the largest iEEG corpus available, showing superior generalization across subjects and clinical tasks compared to models which use vanilla attention like Brant-2 (Yuan et al., 2024a); (3) The release of the **SWEC iEEG dataset**, the largest iEEG dataset publicly available to date, with almost 10,000 hours of highly curated and labeled iEEG recordings.

Moreover, we make all our contributions open source, realizing the first open-data, open-code, and open-weights iEEG foundation model.

## 2 MULTI-VARIATE PARALLEL ATTENTION (MVPA)

This section introduces multi-variate parallel attention (MVPA), the first contribution of this work, as shown in Figure 1. We start with vanilla attention for 1D sequences, then present dual-coded attention for 2D sequences, which has higher computational costs. Building on this, we derive MVPA, which *efficiently* attends to both temporal and spatial aspects of multi-variate time-series data.

### 2.1 MVPA OVERVIEW

Vanilla attention (Vaswani et al., 2017) operates on 1D sequences of embeddings $(\boldsymbol{x}_1, \boldsymbol{x}_2, ..., \boldsymbol{x}_T)$ of dimension $d$ $(\boldsymbol{x}_k \in \mathbb{R}^d)$. It computes the attention between two tokens at positions $(i, j)$ as follows[1]:

$$\boldsymbol{a}_{i,j}^{\text{vanilla}} = (\boldsymbol{x}_i + \mathcal{S}_i)^T W_q^T W_k (\boldsymbol{x}_j + \mathcal{S}_j),$$

where $\boldsymbol{x}_i$ is the query token and $\boldsymbol{x}_j$ the key token. $S$ is the positional encoding, a vector with the same dimensionality $(d)$ that helps to distinguish between different positions in the sequence. $W_q$ and $W_k$ are the learnable query and key matrices.

While vanilla attention has been successfully applied to 1D sequences, its extension to multi-variate time series (i.e., 2D sequences) is not obvious. Specifically, we aim to process sequences of the form $(\boldsymbol{x}_{1,1}, \boldsymbol{x}_{1,2}, ..., \boldsymbol{x}_{c,t}, ..., \boldsymbol{x}_{C,T})$, where $c$ indicates the space and $t$ the time dimension, while maintaining the embedding dimensionality (i.e., $\boldsymbol{x}_{c,t} \in \mathbb{R}^d$). One approach is to flatten the 2D data to a 1D sequence (e.g., as done by the Vision Transformer; Dosovitskiy et al. 2021); however, this will yield a loss in spatial structure. Instead, we introduce two separate learnable positional codebooks, representing space $(\mathcal{C})$ and time $(\mathcal{T})$. By equipping self-attention with this dual encoding we can treat the two dimensions individually, which is fundamental in recovering their interplay and would not be possible with vanilla attention:

$$\boldsymbol{a}_{c,t,c',t'}^{\text{dual}} = (\boldsymbol{x}_{c,t} + \mathcal{T}_t + \mathcal{C}_c)^T W_q^T W_k (\boldsymbol{x}_{c',t'} + \mathcal{T}_{t'} + \mathcal{C}_{c'}).$$

Dual attention allows us to exploit the relationship between time and space at the attention level, the most basic computational unit of a Transformer. We believe this allows the architecture to model the time-series at a lower level, and hence more effectively. However, the dual attention mechanism is computationally expensive, as it computes second-order correlations between time and space.

For this reason, we want to squash these cross-correlations. Specifically, we want to push as much of the spatio-temporal computation as possible to the lower levels of processing without overwhelming it. In contrast, all Transformer models equipped with vanilla attention require ancillary structures to process any relation between time and space (Nie et al., 2023; Zhang & Yan, 2023; Wen et al., 2022). Inspired by Transformer-XL (Dai et al., 2019), we encode the relative distance in the two dimensions between the segments separately and introduce new learnable bias terms $(u, v, w)$. In contrast to Transformer-XL, however, MVPA operates on 2D signals by treating the two dimensions differently and providing a sub-quadratic solution to learn these interactions (see Appendix A.2 for a detailed comparison).

To do so, we operate the following modifications to disentangle space and time:

- $(\mathcal{T}_t + \mathcal{C}_c)^T W_q^T W_k \boldsymbol{x}_{c',t'} \rightarrow u^T W_{k_e} \boldsymbol{x}_{c',t'}$;

---

[1]For better readability, we describe the attention computation for a single head without activation. In practice, we generalize it to multi-head attention and apply a consecutive softmax non-linearity.

- $(\mathcal{T}_t + \mathcal{C}_c)^T W_q^T W_k [\mathcal{T}_{t'} | \mathcal{C}_{c'}] \rightarrow [v|w]^T W_{k_{[t|c]}} [\mathcal{T}_{t-t'} | \mathcal{C}_{c-c'}]$;

which characterize the relative error with respect to the full quadratic dual-encoding attention.

Finally, after removing the second-order effects we rearrange the expanded equation into three related groups:

$$
\begin{aligned}
\boldsymbol{a}_{c,t,c',t'}^{\text{MVPA}} = \boldsymbol{x}_{c,t}^T W_q^T W_{k_e} \boldsymbol{x}_{c',t'} + u^T W_{k_e} \boldsymbol{x}_{c',t'} && \text{Content-based attention} && (1) \\
+ \boldsymbol{x}_{c,t}^T W_q^T W_{k_t} \mathcal{T}_{t-t'} + v^T W_{k_t} \mathcal{T}_{t-t'} && \text{Time-based attention} && (2) \\
+ \boldsymbol{x}_{c,t}^T W_q^T W_{k_c} \mathcal{C}_{c-c'} + w^T W_{k_c} \mathcal{C}_{c-c'} && \text{Channel-based attention} && (3)
\end{aligned}
$$

and compute the final softmax attention as:

$$
A = \frac{\text{softmax}(\boldsymbol{a}^{\text{MVPA}}) V}{\sqrt{d}} \tag{4}
$$

The three terms above are the attentional components of MVPA. Content-based attention only attends to the content of query and key, without any positional encoding. In this component we compute the relationship between two raw segment embeddings, so we modulate the final attention output without considering any structure of the signal. Time-based attention only attends to the query and the distance in time with the key. In this component, only the relative distance in time is considered, allowing for arbitrary signal lengths without loss of generality. Finally, channel-based attention only attends to the query and the distance in space with the key. Similarly to the time-based component, also the distance in the channel-based component is relative.

This feature is particularly interesting for the channel-based component, given the heterogeneity of possible channel setups. Specifically, the channel component uncovers the hidden connection map between the spatial locations from its initial random initialization, as shown in Appendix G.11. In fact, while the use of the absolute position of the electrodes has produced notable work (Jirsa et al., 2023), much of the literature has shown that such information might not be necessary (Schindler et al., 2006; 2008). Therefore, our relative encoding scheme affords us maximum flexibility while not sacrificing performance, as MVPFormer outperforms the SOTA both on the seizure detection task (see Section 5.2.1) and on the four tasks of the Brain TreeBank dataset (see Section 5.2.2), which explicitly provides the absolute channel positions.

## 2.2 EFFICIENT IMPLEMENTATION OF MVPA

As MVPA's computational cost is still quadratic in space and time, we employ several techniques to further reduce the complexity and enable the efficient processing of very large signals. We present here the main techniques, while the details can be found in Appendix A.

Efficiently computing the time- and channel-based terms requires two main techniques. First, we recognize that it is unnecessary to compute the full attention matrix, which is quadratic in the context length (i.e., both time and space). By design, all elements of the time-based attention are the same for each channel (see Figure 1b, the green components are all equal), and all elements of the channel-based attention are the same for each time point (see Figure 1c, the blue components are all equal). Hence, complexity is quadratic in one dimension and constant in the other. We then simply repeat the elements along the right dimension at no additional cost. Second, we employ the shifting operation (Dai et al., 2019) to compute all relative embeddings in one pass.

Content attention remains the most expensive component. To further reduce the cost, we make use of a local attention window (Child et al., 2019) that focuses on the most recent $L$ (in our case 10 segments, or 50 seconds) time points. Since time-based attention is not limited, the lookup window still spans the entire context. Thus, for $L \ll T$, the total complexity of MVPA is $O(T^2 \times C + T \times C^2)$, quadratic in each dimension but subquadratic in the context length. Combining all techniques, MVPA pushes the effective total context length on an NVIDIA A100-80GB GPU to over 10,000 (e.g., 100 channels and 100 time segments).

Additionally, we use grouped query attention (Ainslie et al., 2023) to reduce the number of heads without loss of performance. Moreover, we develop FlashMVPA based on FlashAttention (Dao et al., 2022; Dao, 2024), implemented in the OpenAI Triton language, providing us with lower-level access to CUDA primitives and superior performance (see Appendix A).

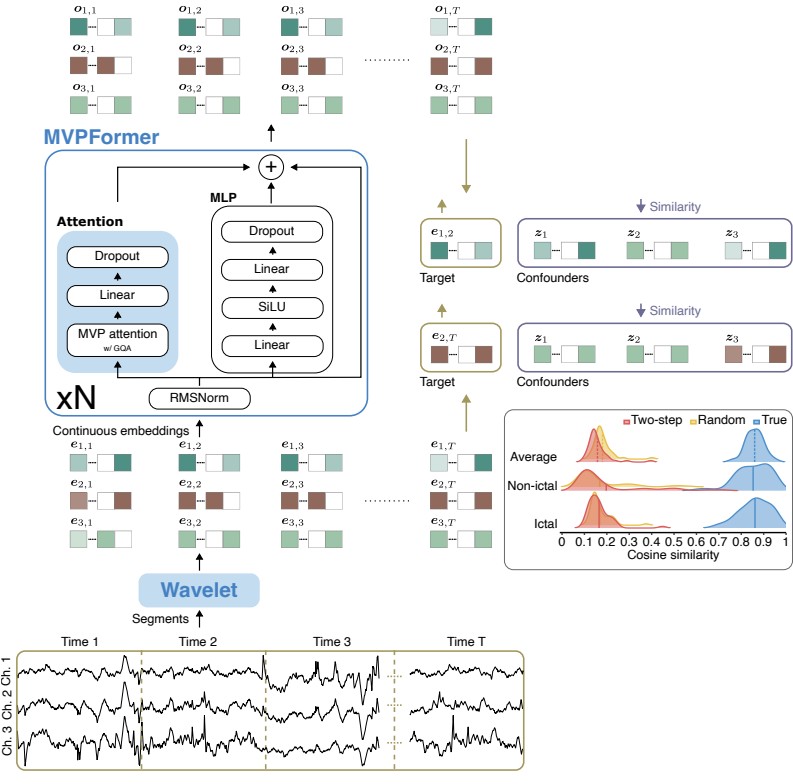

Figure 2: **MVPFormer architecture and forward pass.** iEEG signals are segmented in time and space, encoded via a wavelet-based encoder, and arranged into a 2D embedding grid. These continuous embeddings are processed by MVPA to model temporal, spatial, and content-based dependencies. MVPFormer predicts the next-in-time embedding while reducing similarity to confounders from the same or other subjects. Notched in the bottom right is the resulting cosine similarity with the true target and the confounders after training. The two-step target is the signal twice removed in the future.

## 3 MVPFORMER

MVPFormer is our novel Transformer-based predictive unimodal foundational model equipped with MVPA, that processes heterogeneous multi-variate iEEG data (see Figure 2). While it is customary for language-based Transformer models to employ a finite vocabulary of words, such a definition is non-trivial for iEEG. At the same time, recent works have challenged this discrete paradigm in favor of continuous latent representations (LeCun, 2022; Tack et al., 2025; Tschannen et al., 2025) and continuous chain-of-thought mechanisms (Hao et al., 2024; Geiping et al., 2025). In a similar vein, MVPFormer predicts the development of neuronal activity in a *continuous* embedding space governed by a wavelet encoder. We build MVPFormer following the foundational paradigm, with a pre-training dedicated to predicting the future iEEG embedding using a contrastive loss function. Moreover, we show that a successive fine-tuning using LoRA (Hu et al., 2022) and a simple classification head allows MVPFormer to perform downstream classification tasks. In particular, our results indicate that a model trained in this fashion surpasses an equivalent purely discriminative model (i.e., without generative pre-training), strengthening the validity of foundation models in the iEEG domain as well. Appendix B provides the full details on the architecture.

### 3.1 TRAINING

**Generative pre-training** MVPFormer is pre-trained to generate neuronal activity by predicting successive input segments in time. During pre-training, random input segments from batched windows are used as confounding targets ($Z = \{z_1, ..., z_n\}$), which are plausible but different from the

true target. We compute the contrastive loss as follows

$$\mathcal{L}_{c,t} = -\log \frac{\exp(\text{sim}(\boldsymbol{o}_{c,t}, \boldsymbol{e}_{c,t+1})/\tau)}{\sum_{\boldsymbol{z}_k \in Z} \exp(\text{sim}(\boldsymbol{o}_{c,t}, \boldsymbol{z}_k)/\tau)}, \tag{5}$$

where $\boldsymbol{o}_{c,t}$ is the model's output embedding and $\boldsymbol{e}_{c,t+1}$ the ground-truth next-state embedding. Summing over every $c$, $t$ gives us the optimization target for the generative task. The temperature $\tau$ is 0.1 and the number of confounders $n$ is 30. The contrastive setting (Chen et al., 2020) provides a clearer distinction between segments that are effectively similar, allowing MVPFormer to better model the dynamics of the signal without incurring into the typical pitfalls of L2 distance (such as neural collapse; Han et al. 2022). Given this generative setup, MVPFormer predicts the development in the latent space, rather than the raw signals themselves. See Appendix C.1 for more details.

**Validation of pre-training** Given the architecture of MVPFormer, we need to ensure that the true target and the confounders are sufficiently well-separated in cosine similarity. We evaluate MVPFormer's ability to predict embeddings of future iEEG signals by comparing the predicted embedding at time $t$ to the ground truth embedding at $t + 1$. To do so, we introduce two references: (1) the embedding at $t + 2$, which is highly correlated with $t + 1$, and (2) a random future segment sampled within the next two minutes. Given the high auto-correlation of iEEG signals, a naïve prediction model could simply predict again time $t$, and be moderately successful due to its similarity with $t + 1$. Our results (see Figure 2 and Appendix F) show that the wavelet-based encoder ensures signal features are well preserved, mapping even mildly similar signals to distinct embeddings.

**LoRA fine-tuning for downstream tasks** For downstream tasks, we use a small classification head (i.e., a linear layer). This layer has input size equal to the decoder's block output size, and output size equal to the dimensionality of the classification task (i.e., 2 for seizure classification). The input to this classification head is either the channel-averaged (for seizure detection) or the channel-concatenated output of the last signal segment in time (for the pitch, volume, onset, and speech tasks). The output of the classification head is then passed through a softmax to compute the binary cross-entropy loss. We further adopt LoRA (Hu et al., 2022) to perform parameter-efficient fine-tuning. We only fine-tune the $q$ and $v$ layers of the self-attention in the base MVPFormer model, with a LoRA rank of 8 and and alpha of 16. This leads to a number of trainable parameters during fine-tuning of approximately 0.1% of the base model.

## 4 SWEC IEEG DATASET

The lack of publicly available large-scale iEEG datasets has been a significant obstacle to the development of iEEG foundation models. In fact, while EEG datasets are abundant (Tangermann et al., 2012; Shoeb, 2010) and large (Obeid & Picone, 2016), with many thousands of hours, such resources are lacking in the iEEG domain. Due to barriers tied to data collection and privacy, available iEEG datasets cover few hours (35 subjects and 290 hours; Nejedly et al. 2020) and subjects (10 subjects and 43 hours; Wang et al. 2024a), while larger datasets are kept private Yuan et al. (2024b).

In an effort towards addressing this issue, together with this work we open-source the SWEC iEEG dataset, a large-scale iEEG dataset consisting of a total of 68 subjects, 9328 hours of recording, and 704 ictal events. To our knowledge, the SWEC iEEG dataset is the largest publicly available iEEG dataset, fully curated and labelled by experienced clinicians. Due to institutional data privacy concerns, the dataset does not contain information about the location of the channels in the brain. Appendix D reports more details and illustrates two example recordings.

## 5 EXPERIMENTS

### 5.1 SETUP

**Pre-training** We pre-train MVPFormer on 18 subjects, leaving the remaining 50 subjects for testing. MVPFormerM is pre-trained for 1.2M steps on a single node with 8 NVIDIA A100-80GB GPUs for two weeks. The chosen optimizer is FusedAdam with 0.1 weight decay, from the Deepspeed library compiled on the specific machine. The training used bf16-mixed DeepSpeed stage 2 without activation checkpointing. The learning rate is fixed to $10^{-4}$. The training environment includes PyTorch 2.0, PyTorch Lightning 2.0, and Triton 2.1.0.

**Fine-tuning** After pre-training, we further fine-tune the MVPFormer for each task. For the seizure detection task, we fine-tune on the same 18 subjects of the SWEC iEEG dataset, and then test in a zero-shot manner on nearly 7,000 hours of iEEG data from 50 unseen subjects, all suffering from epilepsy. To keep computational cost moderate, we use a subset of the channels of each subject for testing: we select them based on a combination of variance and kurtosis, excluding the noisier ones (see Appendix E.1). In a real-world clinical scenario selection would comprise a minimal additional burden for the expert. The number of channels chosen is fixed to 32 to simplify comparisons with the other baselines, but we also provide ablations using manual channel selection and no channel selection at all (see Appendix G.8). For the four tasks of the Brain TreeBank dataset, we follow the same procedure as BrainBERT (Wang et al., 2023) and PopT (Chau et al., 2025) by first fine-tuning on the specific subject on a subset of the data and then testing on the remaining data. As before, we also evaluate the robustness of MVPFormer with respect to the channel selection in Appendix G.16.

## 5.2 iEEG tasks

### 5.2.1 Seizure detection task

We begin by evaluating MVPFormer on the seizure detection task on iEEG data. First, we consider a clinically realistic setup that compares model predictions to a board-certified neurologist annotations using Cohen's Kappa. To do so, the predictions are post-processed to yield episodic results (see Appendix E). Cohen's Kappa (Danker-Hopfe et al., 2004; Schlögl et al., 2005; McHugh, 2012) is widely used to quantify inter-rater reliability in seizure classification. The Landis and Koch criteria (Landis & Koch, 1977) (see Appendix E) are often used in practice to evaluate human performance. Expert-level performance in the seizure classification task varies considerably, from 0.58 (Halford et al., 2015) to 0.53 (Grant et al., 2014) in EEG, to 0.57 (Quigg et al., 2015) in iEEG. We consider Kappa values above 0.53 to be expert-level. We must also consider that when comparing the decisions of human raters, only few (in the order of tens) curated episodes are evaluated. In contrast, our setup involves many more subjects and ictal events (in the order of thousands), making this task more challenging for MVPFormer. MVPFormer achieves an average Kappa of 0.61 across 50 unseen subjects from the SWEC iEEG dataset, matching human expert performance (see Figure 13). Importantly, agreement varies by subject, reflecting the clinical reality that seizure presentation complexity strongly affects classification (see Appendix G.10).

Overall, MVPFormer demonstrates expert-level seizure classification across a large, heterogeneous cohort. This performance, combined with its low false positive rate (0.15 fp/h), positions MVPFormer as a promising clinical assistant for real-world iEEG analysis.

Second, we consider a conventional evaluation (see Appendix G.3) based on F1-score, sensitivity, specificity, and number of false positives per hour (fp/h). We compare MVPFormer against three strong baselines: (1) Brant-2 (Yuan et al., 2024a), a SOTA Transformer for iEEG, fine-tuned here starting from the published weights. Brant-2 requires all subjects to have the same number of channels for classification; hence, we were not able to test all subjects with it. (2) BrainBERT (Wang et al., 2023), another SOTA iEEG model with public weights. (3) MV-Llama, an ablation of MVPFormer-S that replaces MVPA with vanilla attention, is trained identically to MVPFormer. After undergoing the same task-specific finetuning of MVPFormer, we test the considered models zero-shot across the unseen subjection of our SWEC iEEG dataset, and also apply them on the MAYO and FNUSA datasets (Nejedly et al., 2020) (see Appendix G.12).

As shown in Table 1, all baselines fail to generalize on our SWEC iEEG dataset, achieving Kappa scores of just 0.11, 0.05, and 0.00, while MVPFormer achieves 0.61 and 0.57 in medium and small configurations, respectively. Specifically, BrainBERT always fails to detect a seizure, while Brant-2's behaviour is more nuanced. We provide in Appendix G the complete per-subject statistics, including 95% confidence intervals. Moreover, MVPFormer outperforms the baselines on MAYO (highest f1-score of 0.36) and is on par on FNUSA. As a further baseline — to validate our choice of pre-training — we also compare MVPFormer-S with an equivalent model built without the generative base task (i.e., without the initial contrastive loss-based training, see Appendix G.13). With this setup, the purely discriminative model only reaches a Kappa score of 0.52, inferior to the equivalent MVPFormer-S which reached 0.54. The full set of results can be found in Appendix G.1.

Table 1: **Results on the iEEG seizure detection tasks.** We compare MVPFormer with multiple baselines across 3 iEEG datasets. The best results are bolded.

|  |  | SWEC | | MAYO | FNUSA |
| --- | --- | --- | --- | --- | --- |
| Model | Attention | Kappa | f1 | f1 | f1 |
| MVPFormer | MVPA | **0.61** | **0.59** | **0.36** | **0.46** |
| MVPFormer-S | MVPA | 0.57 | 0.53 | 0.35 | 0.46 |
| MV-Llama | Vanilla | 0.11 | 0.01 | / | / |
| Brant-2 | Vanilla | 0.06 | 0.01 | 0.19 | 0.46 |
| BrainBERT | Vanilla | 0.00 | 0.00 | / | / |

### 5.2.2 BRAIN TREEBANK DECODING TASKS

We validate the generalization of MVPFormer by testing it on the four tasks of the Brain TreeBank dataset (Wang et al., 2024a), as described in Wang et al. (2023); Chau et al. (2025). The four tasks are: 1) discrimination of volume level (volume), 2) discrimination of pitch (pitch), 3) classification of sentence onset (onset), and 4) classification of speech (speech). All four tasks involve the discrimination of high-level cognitive behaviors from iEEG recordings. As such, they represent a significant testbed for MVPFormer outside of its design environment of seizure detection. Table 2 shows the results of MVPFormer against the SOTA baselines represented by PopT (Chau et al., 2025), BrainBERT (Wang et al., 2023), and Brant (Zhang et al., 2023), as reported by PopT (Chau et al., 2025). The full results can be found in Appendix G.2.

Table 2: **Results on the Brain TreeBank tasks.** We compare MVPFormer with multiple baselines the 4 tasks of the Brain TreeBank dataset. The models requiring the electrodes' position are indicated by †. The best results without the electrodes' position are bolded, while the results where the electrodes' position is beneficial are underlined.

|  |  | Pitch | Volume | Onset | Speech |
| --- | --- | --- | --- | --- | --- |
| Model | Attention | acc | acc | acc | acc |
| MVPFormer-S | MVPA | **0.83** | **0.88** | **0.87** | **0.90** |
| MV-Llama | Vanilla | 0.63 | 0.77 | 0.80 | 0.81 |
| Brant | Vanilla | 0.61 | 0.74 | 0.80 | 0.80 |
| BrainBERT | Vanilla | 0.59 | 0.66 | 0.70 | 0.71 |
| PopT † | Vanilla | 0.74 | 0.87 | 0.90 | 0.93 |
| PopT | Vanilla | 0.62 | 0.76 | 0.81 | 0.83 |

The Brain TreeBank dataset contains information about the 3D location of the electrodes. On the one hand, this information is often unavailable in datasets, so we explicitly design MVPA not to require it, by autonomously building an implicit channel map (see Appendix G.11). On the other hand, PopT was specifically developed for the Brain TreeBank dataset, taking into account the electrodes' physical location. Nonetheless, MVPFormer surpasses all baselines on the pitch and volume tasks, providing further evidence that MVPA is well-suited to iEEG tasks and generalizing the model's ability beyond its original task. On the remaining two tasks, MVPFormer is second behind PopT, but is still superior to PopT without the electrodes' location. These results indicate that, while the electrodes' position might be beneficial in some instances, MVPA's implicit channel map produces superior results overall by being more flexible and adaptable to a wider variety of existing datasets.

## 6 EVALUATION ON GENERAL TIME-SERIES

We have shown that MVPFormer achieves SOTA results in its native modality of iEEG. However, the design of MVPA should allow it to make use of the tiem and channel information intrinsic to any multi-channel time-series. To this end, we also provide a baseline evaluation of MVPA against

established alternatives in the time-series domain. We compare MVPFormer with existing SOTA architectures on classical long-term forecasting and classification tasks.

## 6.1 TIME-SERIES FORECASTING

Table 3 reports the results of MVPFormer, the vanilla Transformer (Vaswani et al., 2017), PatchTST (Nie et al., 2023), TimesFM (Das et al., 2024), TimeMixer (Wang et al., 2024b), and WPMixer (Murad et al., 2025) on the ETTh1, ETTh2, and Weather datasets (Zhou et al., 2021; Wu et al., 2021). MVPFormer always equals or surpasses the baselines (see Appendix G.17 for the full results).

Table 3: **Results on the time-series forecasting task.** We report the mean-squared error (MSE) and mean-absolute error (MAE) averaged over all forecasting lengths.

| Model | | **MVPFormer** | | Transformer | | PatchTST | | TimesFM | | TimeMixer | | WPMixer | |
|---|---|---|---|---|---|---|---|---|---|---|---|---|---|
| Dataset | Length | MSE | MAE | MSE | MAE | MSE | MAE | MSE | MAE | MSE | MAE | MSE | MAE |
| ETTh1 | Avg. | **0.45** | 0.45 | 1.00 | 0.80 | **0.45** | 0.45 | **0.45** | 0.45 | **0.45** | 0.44 | **0.45** | 0.44 |
| ETTh2 | Avg. | **0.38** | 0.41 | 3.37 | 1.48 | 0.39 | 0.41 | **0.38** | 0.41 | 0.39 | 0.41 | **0.38** | 0.41 |
| Weather | Avg. | **0.25** | 0.28 | 0.59 | 0.53 | 0.26 | 0.28 | 0.26 | 0.28 | **0.25** | 0.28 | **0.25** | 0.28 |

## 6.2 TIME-SERIES CLASSIFICATION

Moreover, we evaluate MVPA on common classification tasks, against the vanilla Transformer and PatchTST on the EthanolConcentration (EtCo), FaceDetection (FaDe), HandWriting (HaWr), Heartbeat (HaBe), JapaneseVowels (JaVo), PEMS-SF (PEMS), SCP1, SCP2, SpokenArabic (SpAr), and Uwave datasets (Liu & Wang, 2024).

Table 4: **Accuracy on time-series classification tasks.** We report the accuracy per task.

| | EtCo | FaDe | HaWr | HaBe | JaVo | PEMS | SCP1 | SCP2 | SpAr | Uwave |
|---|---|---|---|---|---|---|---|---|---|---|
| **MVPFormer** | **0.33** | 0.66 | 0.21 | 0.70 | **0.95** | **0.86** | **0.86** | **0.54** | **0.97** | 0.80 |
| Transformer | 0.29 | 0.64 | 0.20 | 0.70 | 0.91 | 0.84 | 0.83 | **0.54** | 0.95 | 0.80 |
| PatchTST | 0.29 | **0.67** | **0.23** | **0.72** | **0.95** | 0.85 | 0.83 | 0.51 | **0.97** | **0.82** |
| TimesFM | 0.29 | **0.68** | **0.23** | 0.71 | 0.93 | 0.84 | 0.83 | 0.52 | **0.99** | **0.82** |

TimeMixer and WPMixer are forecasting-only architectures, so we could not test them. Table 4 shows that MVPFormer achieves SOTA results on general classification tasks as well. At the same time, these results highlight the generalization capability of MVPFormer that, in contrast to other models, is effective in both forecasting and classification.

## 6.3 ABLATION OF THE THREE COMPONENTS

Finally, we make use of the general time-series setting to validate the contribution of the three different components of MVPA: content, time, and channel. In Sections 5.2.1 and 5.2.2 we confirmed that MVPA provides a notable increase in performance to MVPFormer in its native iEEG environment.

In Table 5 we report the results of MVPA, content-only MVPA, time-only MVPA, channel-only MVPA, and no-component MVPA on the ETTh1, ETTh2, and Weather datasets. MVPA obtains the best performance over all datasets, providing further evidence that the three components jointly learn the different aspects of the signal. Interestingly, the performance gap on the Weather dataset is larger, as is its number of channels: 21 instead of 7 of ETTh1 and ETTh2. This result supports our design of MVPA, which is able to take advantage of the information content in the strongly multi-variate time-series better than its simpler counterparts. The full results can be found in Appendix G.18.

Table 5: **Ablation of the components of MVPA on the time-series forecasting task.** We report the mean-squared error (MSE) and mean-absolute error (MAE) averaged over all forecasting lengths.

| Model | | MVPA | | Content-only | | Time-only | | Channel-only | | None | |
|---|---|---|---|---|---|---|---|---|---|---|---|
| Dataset | Length | MSE | MAE | MSE | MAE | MSE | MAE | MSE | MAE | MSE | MAE |
| ETTh1 | Avg. | **0.45** | 0.45 | 0.46 | 0.45 | 0.46 | 0.45 | 0.46 | 0.45 | 0.47 | 0.46 |
| ETTh2 | Avg. | **0.38** | 0.41 | **0.38** | 0.40 | **0.38** | 0.41 | 0.38 | 0.41 | 0.40 | 0.41 |
| Weather | Avg. | **0.25** | 0.28 | 0.27 | 0.29 | 0.27 | 0.29 | 0.26 | 0.28 | 0.27 | 0.29 |

## 7 RELATED WORKS

Single-channel data has been treated as 1D sequences for tasks like speech recognition, where the signal is divided into patches that serve as tokens (Schneider et al., 2019; Gulati et al., 2020). Extending vanilla attention to multi-dimensional data, such as images, is more complex. The Vision Transformer (Dosovitskiy et al., 2021) processes images by flattening 2D patches into a 1D sequence, losing spatial structure in the process. However, this approach is inflexible and unsuitable for generalizing to images with different heights and widths. Other mechanisms (Ho et al., 2019; Huang et al., 2019; Child et al., 2019; Bulat et al., 2021; Arnab et al., 2021) alternative to vanilla attention have been developed to speed up computation in the 1D case or to extend it to 2D. We compare MVPA against such alternatives in more detail in Appendix A.2.

For multi-variate time-series, such as EEG, Transformers face challenges due to the need to preserve both time and channel information (Wen et al., 2022; Cui & Lv, 2024). Channel-independent approaches (Nie et al., 2023) reuse vanilla self-attention and discard all information content in the time dimension, while channel-mixing promises to preserve it by either fusing the channels (Zhou et al., 2022) or processing them sequentially (Zhang & Yan, 2023). For neural spike data, a fusion of channel and time aspects has been proposed (Le & Shlizerman, 2022), albeit without complete integration at the attention level. Specifically for iEEG and EEG, there exist few Transformer-based solutions (Zhang et al., 2023; Yuan et al., 2024a). Since electrode placements vary widely across subjects, these models struggle with the heterogeneous nature of the data.

Some models (Chau et al., 2025) tackle this issues by requiring the absolute position of the electrodes, limiting their applicability to datasets that do have such information. Due to practical and ethical concerns, the publicly available datasets without the absolute position of the electrodes (Burrello et al., 2018; 2019; Nejedly et al., 2020; Li et al., 2021), including ours, notably outsize the ones that do (Wang et al., 2024a; Keles et al., 2024; Zada et al., 2025). Overall, the complex interplay between time and space, where distant brain regions may be more strongly connected than nearby ones, makes it difficult for conventional attention mechanisms to effectively process iEEG signals.

## 8 CONCLUSION

We introduce MVPA, a novel attention mechanism designed to effectively process multi-variate time-series data, exemplified by its application to iEEG signal analysis. MVPA enables MVP-Former, a foundation model trained on our novel SWEC iEEG dataset, to capture complex interactions between time and spatial dimensions in multi-variate time-series. We also contribute the SWEC iEEG dataset itself, as the largest iEEG dataset currently publicly available. MVPFormer is trained following the foundational paradigm to predict the next brain states, and then further fine-tuned on multiple tasks. MVPA ensures robust performance across several iEEG tasks and dataset. It reaches high inter-rater agreement (0.61 Kappa score) on our large scale and challenging SWEC iEEG dataset, notably surpassing the SOTA Brant-2 (0.08). It also achieves SOTA results on the four tasks of the Brain TreeBank dataset, even surpassing models specifically designed for them. Moreover, MVPA equals or surpasses the SOTA also in classical time-series forecasting and classification tasks. Overall, our results show that MVPA affords MVPFormer superior generalization capabilities while maintaining computational efficiency and scalability, marking a significant advancement in the analysis of time-series data and iEEG in particular.

ACKNOWLEDGMENTS

This work is supported by the Swiss National Science foundation (SNF), grant no. 200800.

ETHICS STATEMENT

During the collection of the SWEC iEEG dataset, all the subjects gave written informed consent that their iEEG data might be used for research and teaching purposes. The decision on the necessity for iEEG recordings, the electrode implantation scheme, and the decision about surgical therapy were made entirely on clinical grounds. These decisions were taken prior to and completely independently from the compilation of this dataset.

REPRODUCIBILITY STATEMENT

This paper describes the MVPA algorithm in Section 2 and Appendix A, and the architecture of MVPFormer in Section 3 and Appendix B. All the hyperparameters are in Appendix B.

The setup used for training and evaluating our model are in Section 5.1.

The SWEC iEEG dataset is publicly available at `https://huggingface.co/datasets/NeuroTec/SWEC_iEEG_Dataset`.

The code and the weights of MVPFormer are publicly available at `https://github.com/IBM/multi-variate-parallel-transformer`.

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

# A  DETAILS ON MULTI-VARIATE PARALLEL ATTENTION (MVPA)

Algorithm 1 illustrates the multi-variate parallel attention algorithm.

---

**Algorithm 1:** Computation of MVPA

---

**Input:** $\boldsymbol{x}_{c,t} \in \mathbb{R}^{n_{\text{embed}}}$ output token of Encoder; $n_{\text{embed}} = 768$
**Output:** $\boldsymbol{o}_{c,t} \in \mathbb{R}^{n_{\text{embed}}}$ output attention
**Data:** $\boldsymbol{t}$ time encoding; $\boldsymbol{c}$ channel encoding; $\boldsymbol{u}, \boldsymbol{y}, \boldsymbol{w}$ biases; $h \in [1, \ldots, n_{\text{heads}}]$;
    $\boldsymbol{h}_{k,v} \in [1, \ldots, n_{\text{gqa}}]$

1  **def** MVPAttention($\boldsymbol{x}_{c,t}$):
       # Compute query separately from key and value due to GQA
2      $\boldsymbol{q}_{c,t}^h \leftarrow \text{LINEARNOBIAS}(\boldsymbol{x}_{c,t})$
3      $\boldsymbol{k}_{c,t}^{h_{k,v}}, \boldsymbol{v}_{c,t}^{h_{k,v}} \leftarrow \text{LINEARNOBIAS}(\boldsymbol{x}_{c,t})$
       # Compute the three components of MVPA
4      $\boldsymbol{g}_{(c,t),(c',t')}^h \leftarrow (\boldsymbol{q}_{c,t}^h + \boldsymbol{u}^{h_{k,v}})^T \boldsymbol{q}_{c',t'}^h$
       # Time and channel components are independent of the key
          content, so they do not need to be recomputed
5      $\boldsymbol{s}_t^h \leftarrow (\boldsymbol{q}_{c,t}^h + \boldsymbol{y}^{h_{k,v}})^T \boldsymbol{t}^{h_{k,v}}$
6      $\boldsymbol{l}_c^h \leftarrow (\boldsymbol{q}_{c,t}^h + \boldsymbol{w}^{h_{k,v}})^T \boldsymbol{c}^{h_{k,v}}$
       # Shift the time and channel components to avoid
          recomputation, from Transformer-XL
7      $\boldsymbol{s}_{t,t'}^h \leftarrow \text{SHIFTTIME}_{t'}(\boldsymbol{s}_t^h)$
8      $\boldsymbol{l}_{c,c'}^h \leftarrow \text{SHIFTCHANNEL}_{c'}(\boldsymbol{l}_c^h)$
       # Apply window and causal mask
9      $\boldsymbol{m}_{(c,t),(c',t')}^h \leftarrow \text{CAUSALMASK}(\boldsymbol{g}_{(c,t),(c',t')}^h + \boldsymbol{s}_{t,t'}^h + \boldsymbol{l}_{c,c'}^h)$
10     $\boldsymbol{n}_{(c,t),(c',t')}^h \leftarrow \text{WINDOWMASK}_{10}(\boldsymbol{m}_{(c,t),(c',t')}^h)$
       # Apply structured dropout
11     $\boldsymbol{d}_{(c,t),(c',t')}^h \leftarrow \text{STRUCTUREDDROPOUT}_{0.1}(\boldsymbol{n}_{(c,t),(c',t')}^h)$
       # Compute final attention value
12     $\boldsymbol{a}_{(c,t),(c',t')}^h \leftarrow \text{SIGMOID}(\frac{1}{\sqrt{n_{\text{embed}}}} \boldsymbol{d}_{(c,t),(c',t')}^h)$
13     $\boldsymbol{o}_{c,t}^h \leftarrow \sum_{c',t'} \boldsymbol{a}_{(c,t),(c',t')}^h \cdot \boldsymbol{v}_{(c,t),(c',t')}^{h_{k,v}}$
14     **return** $\boldsymbol{o}_{c,t}^h$

---

## A.1  FURTHER MOTIVATION OF MVPA

Single-channel data can be treated equivalently to sentences, by dividing the signal into 1D patches, which form the tokens. This modality has attracted considerable interest (Schneider et al., 2019; Gulati et al., 2020), frequently for speech recognition tasks that are once again related to the natural language domain.

There is no straightforward extension of vanilla attention to the 2D case. The Vision Transformer (Dosovitskiy et al., 2021) processes images by extending the notion of the patches to the 2D case. It carves images into a collection of patches, which it then flattens into a 1D sequence. Each patch has 2D coordinates $(i, j)$ which get flattened by an arbitrary function $f : \mathbb{N} \times \mathbb{N} \to \mathbb{N}$ into a 1D index ($l$). This is a simple way to recover the 1D case, but it has several drawbacks. First, by flattening the patches we lose any notion of spatial structure, as nearby patches in space are no longer necessarily close in the sequence. Any information about the structure of the patches is lost. However, if the size of the images, the number of patches, and the flattening direction are kept constant, then the Transformer might autonomously learn it. If it learns the structure, then it cannot be exposed to different images as it would completely misinterpret them; if it does not learn the structure, then it is missing critical information. This leads to an inflexible model which cannot easily generalize to different images. One possible solution is to choose a bijective $f$, such as the Cantor pairing function, to have a one-to-one correspondence between the position of the patch in

the image and in the sequence. This solution is, however, quite unintuitive. Second, the Vision Transformer does not distinguish between the two dimensions of height and width, i.e., it does not distinguish between up, down, left, and right. For images this limitation is not too impactful, as most of the information is conveyed in the closeness of two patches and not their relative position in any dimension.

The patching schema of the Vision Transformer is unsuitable to multi-variate time-series, as the two dimensions of time and channels require delicate handling. Transformers for time-series are a well-known problem in the field (Wen et al., 2022). Channel-independent approaches (Nie et al., 2023) reuse vanilla self-attention and discard all information content in the time dimension, while channel-mixing promises to preserve it by either fusing the channels (Zhou et al., 2022) or processing them sequentially (Zhang & Yan, 2023). The second family of solutions is more promising in addressing the issue but is still limited either with respect to computational expense or expressiveness.

EEG signals are multi-variate recordings of the brain. Transformer-based approaches to EEG are sparse (Si et al., 2023; Cui et al., 2023), due to the often unmanageable complexity of the data. In iEEG recordings, the subjects are implanted with electrodes directly in multiple areas of the brain for the purpose of clinical diagnosis. There is no standardized location, or even number of electrodes, for intracranial implants. This makes iEEG an extremely heterogeneous data modality, intractable for conventional attention approaches. The channels present a fundamental source of information, as electric fields spread in different areas of the brain on different time-scales and with different intensities depending on the strength of the connection between the areas. Moreover, the relationship between brain regions is not always proportional to their spatial closeness, as distant areas might be more strongly connected than close ones. There is a tremendously intricate interplay between space and time, which the Transformer must exploit.

## A.2 COMPARISON WITH ALTERNATIVE ATTENTION MECHANISMS

We further compare MVPA with other existing alternatives to better characterize the features of MVPA. In particular, we draw our main inspiration for the disentanglement and relative positional encoding from Transformer-XL (Dai et al., 2019) and DeBERTa (He et al., 2021), which were the first to introduce this concept. We now compare MVPA against a selection of relevant alternative attention mechanisms.

Table 6: **Summary of the differences between MVPA and existing attention mechanisms.**

|  | Domain | Complexity | Disentangled | Relative position | Simultaneous time and space | Receptive field |
|---|---|---|---|---|---|---|
| Vanilla | 1D | Quadratic | No | No | No | Global |
| Transformer-XL | 1D | Quadratic | Yes | Yes | No | Global |
| DeBERTa | 1D | Quadratic | Yes | No | No | Global |
| Axial | 2D | Subquadratic | No | No | No | Local |
| Criss-cross | 2D | Subquadratic | No | No | Yes | Local |
| Localized | 2D | Quadratic | No | No | Yes | Local |
| Space-time | 2D | Subquadratic | Yes | No | No | Global |
| ViViT | 2D | Subquadratic | Yes | No | No | Global |
| **Ours** | **2D** | **Subquadratic** | **Yes** | **Yes** | **Yes** | **Global** |

Axial attention (Ho et al., 2019) consists of two separate attention mechanisms, *RowAttention* and *ColumnAttention*, each of which attends to one row (one channel) or one column (one timepoint) only. The layers are then stacked sequentially to recover the full receptive field. MVPA, in contrast, attends to both time and space simultaneously, and has a global receptive field built-in at every layer.

Criss-cross attention (Huang et al., 2019) computes the attention between each point and all the other points in its row or column via the *affinity* operation. Once again, the layers are applied recursively to obtain the full receptive field. One of the most significant differences between MVPA and criss-cross attention is in the encoding of the position. In fact, MVPA treats rows and columns differently through two independent positional codebooks, while in criss-cross attention distance in rows and heights is equivalent. This is a natural consequence of the design choices, as criss-cross attention is designed for images, where the two dimensions are indeed equivalent. Moreover, MVPA again has a global receptive field.

Localized sparse transformers (Child et al., 2019) use separate heads with separate connectivity patterns to improve on the computational requirements of the full attention. As before, the full receptive field is only recovered with multiple applications. MVPA, on the other hand, computes the full 2D attention over the entire input in every head. Moreover, the separate positional codebooks allow MVPA to treat the dimensions differently, which localized sparse transformers cannot do.

Space-time mixing (Bulat et al., 2021) reduces the computational complexity of the full quadratic self-attention by only computing attention across the channels, and then performing a simple averaging in time. On the other hand, MVPA fully integrates both dimensions of the signal in the computation at the attention level.

Finally, ViViT (Arnab et al., 2021) disentangles space and time through the use of sequential encoders, similar to the channels mixing approaches mentioned in Section 7.

Table 6 summarizes the main features of MVPA with respect to the considered alternatives.

### A.3 EFFICIENT COMPUTATION OF MVPA

Vanilla attention is quadratic in the number of input elements, and this often represents a significant computational roadblock (Kitaev et al., 2020). The input becomes intractable as the number of channels increases, especially for multi-variate time-series. At the same time, more channels imply more sources of information, and we cannot simply discard them.

MVPA is also quadratic, but we employ a number of techniques to significantly reduce the computational complexity and make the processing of very large signals feasible. Letting $T$ be the number of time segments and $C$ be the number of channels, the context length of the Transformer becomes $T \times C$ and number of terms necessary to compute for vanilla attention $O(T^2 \times C^2)$. Given a reasonable estimation of 100 segments and 50 channels the context length would be 5,000, until recently intractable even for language models.

By dividing MVPA into three components we gain considerable advantages (see Table 7 for the complexity of each term). Efficiently computing the time- and channel-based terms requires two main techniques. First, we recognize that it is not necessary to compute the full square matrix, which would be quadratic in the context length (i.e., both time and space). By design, all elements of the time-based attention are the same for each channel, and all elements of the channel-based attention are the same for each time point. Hence, complexity is quadratic in one dimension and constant in the other. We then simply repeat the elements along the right dimension at no additional cost. Second, we employ the shifting operation described in Supplementary Section B of Transformer-XL (Dai et al., 2019) to compute all relative embeddings in one pass.

|  | Time-based | Channel-based | Content-based (w/ window) | Content-based (vanilla) |
|---|---|---|---|---|
| Complexity | $O(T^2 \times C)$ | $O(T \times C^2)$ | $O(L^2 \times C^2)$ | $O(T^2 \times C^2)$ |

Table 7: **Complexity of each component of MVPA.** $T$ is the number of time segments in the signal, $C$ is the number of channels, and $L$ is the size of the local window. Content-based attention without window has the same complexity as vanilla attention.

Content attention, though stripped of positional encoding, remains the most expensive component. To further reduce computational cost, with little impact to performance, we make use of a local attention window (Child et al., 2019) which focuses on the most recent L time points discarding ones that have little information content. Since time-based attention is not limited, the lookup window still spans the entire context (though it is affected, see Figure 19b). Thus, for $L \ll T$, the total complexity of MVPA is $O(T^2 \times C + T \times C^2)$, quadratic in each dimension but subquadratic in the context length. Combining all techniques, MVPA pushes the effective total context length to over 10,000.

Given the three components are independent of each other, it is possible to exclude any one and reduce computations even more. As an additional cost-saving measure, we use grouped query attention (Ainslie et al., 2023) to reduce the number of heads without loss of performance. In summary, MVPA correctly treats time and space as unrelated dimensions, forcing the model to consider them separately, all with little computational overhead.

### A.4 TRITON IMPLEMENTATION OF MVPA

MVPFormer's training effectiveness is heavily affected by batch size, as its training routine draws the negative samples from the batch. The bigger the batch size, the more variety in the negative samples and the better the model generalizes. Given the large context size of MVPFormer, up to 10k, a pure Python implementation of scaled dot product attention would consume too much VRAM to be useful. FlashAttention (Dao et al., 2022) and FlashAttention-2 (Dao, 2024) provide the blueprint to solve this problem, though they only apply to vanilla attention. Using tiling, FlashAttention makes VRAM consumption linear instead of quadratic in the context length, enabling training on much longer context.

We develop FlashMVPA using the same technique in the OpenAI Triton language, which gives lower-level access to CUDA primitives. While a CUDA implementation could likely deliver better raw performance, the choice of Triton is dictated by the much lower coding time, though Triton is less robust and more prone to unexpected behaviors at this point. The time-based and channel-based components of MVPA are computed using PyTorch's own matrix-multiply, but are then shifted (Transformer-XL trick) and added using Triton, while the content-based component is fully implemented in Triton. This is due to limitations in Triton. FlashMVPA reaches 20 TFlops on an A100.

---

**Algorithm 2:** Computation of FlashMVPA

**Input:** $\boldsymbol{x}_{c,t} \in \mathbb{R}^{n_{\text{embed}}}$ output token of Encoder; $n_{\text{embed}} = 768$
**Output:** $\boldsymbol{o}_{c,t} \in \mathbb{R}^{n_{\text{embed}}}$ output attention
**Data:** $\boldsymbol{t}$ time encoding; $\boldsymbol{c}$ channel encoding; $\boldsymbol{u}, \boldsymbol{y}, \boldsymbol{w}$ biases; $h \in [1, \ldots, n_{\text{heads}}]$;
     $h_{k,v} \in [1, \ldots, n_{\text{gqa}}]$

1 **def** FlashMVPAttention($\boldsymbol{x}_{c,t}$):
    # Compute query separately from key and value due to GQA
2     $\boldsymbol{q}_{c,t}^h \leftarrow$ LINEARNOBIAS($\boldsymbol{x}_{c,t}$)
3     $\boldsymbol{k}_{c,t}^{h_{k,v}}, \boldsymbol{v}_{c,t}^{h_{k,v}} \leftarrow$ LINEARNOBIAS($\boldsymbol{x}_{c,t}$)
    # Need to compute time and channel components outside Triton
4     $\boldsymbol{s}_t^h \leftarrow (\boldsymbol{q}_{c,t}^h + v^{h_{k,v}})^T \boldsymbol{t}^{h_{k,v}}$
5     $\boldsymbol{l}_c^h \leftarrow (\boldsymbol{q}_{c,t}^h + w^{h_{k,v}})^T \boldsymbol{c}^{h_{k,v}}$
    # Triton MVPA combines all computations into one kernel
6     $\boldsymbol{o}_{c,t}^h \leftarrow$ TRITONMVPA($\boldsymbol{q}_{c,t}^h, \boldsymbol{s}_t^h, \boldsymbol{l}_t^h, \boldsymbol{v}, \boldsymbol{u}, \boldsymbol{y}, \boldsymbol{w}$)
7     **return** $\boldsymbol{o}_{c,t}^h$

---

### A.5 RELATIVE SHIFTING

By design, MVPA requires the computation of relative time and channel encodings, which can notably slow down the overall operation. While this does not affect vanilla attention, other relative attentions provide us with an elegant solution to this problem. In particular, the shifting operation from Transformer-XL provides us with an efficient alternative to recomputing the time- and channel-based attention components. To keep notation simple, let $\boldsymbol{q}_t = \boldsymbol{x}_{\cdot,t}^T W_q^T$, $\boldsymbol{p}_c = \boldsymbol{x}_{c,\cdot}^T W_q^T$, $\boldsymbol{T}_i = W_{k_t} \mathcal{T}_{T-1-i}$, and $\boldsymbol{C}_i = W_{k_C} \mathcal{C}_{C-1-i}$. The shift in time can be performed as in the original implementation

$$\begin{pmatrix} \boldsymbol{q}_0 \boldsymbol{T}_0 & \boldsymbol{q}_0 \boldsymbol{T}_1 & \cdots & \cdots & \boldsymbol{q}_0 \boldsymbol{T}_{T-1} \\ \boldsymbol{q}_1 \boldsymbol{T}_0 & \boldsymbol{q}_1 \boldsymbol{T}_1 & \cdots & \cdots & \boldsymbol{q}_1 \boldsymbol{T}_{T-1} \\ \vdots & \vdots & \vdots & \vdots & \vdots \\ \boldsymbol{q}_{T-1} \boldsymbol{T}_0 & \boldsymbol{q}_{T-1} \boldsymbol{T}_1 & \cdots & \cdots & \boldsymbol{q}_{T-1} \boldsymbol{T}_{T-1} \end{pmatrix} \xrightarrow{\text{SHIFTTIME}} \begin{pmatrix} \boldsymbol{q}_0 \boldsymbol{T}_{T-1} & 0 & \cdots & \cdots & 0 \\ \boldsymbol{q}_1 \boldsymbol{T}_{T-2} & \boldsymbol{q}_1 \boldsymbol{T}_{T-1} & 0 & \cdots & 0 \\ \vdots & \vdots & \vdots & \vdots & \vdots \\ \boldsymbol{q}_{T-1} \boldsymbol{T}_0 & \cdots & \cdots & \cdots & \boldsymbol{q}_T \boldsymbol{T}_{T-1} \end{pmatrix}$$
(6)

The right triangular matrix is zeroed out as a requisite of autoregressive training, i.e., we cannot attend to keys in the future. The entire time shifting operation can be performed efficiently and quickly using tensor manipulation in PyTorch.

Thus, the time attention component does not require recomputation for each time position, i.e. each row in the matrix of the time component.

The shift in channels is more involved

$$\begin{pmatrix} \boldsymbol{p}_0\boldsymbol{C}_0 & \boldsymbol{p}_0\boldsymbol{C}_1 & \cdots & \cdots & \boldsymbol{p}_0\boldsymbol{C}_{C-1} \\ \boldsymbol{p}_1\boldsymbol{C}_0 & \boldsymbol{p}_1\boldsymbol{C}_1 & \cdots & \cdots & \boldsymbol{p}_1\boldsymbol{C}_{C-1} \\ \vdots & \vdots & \vdots & \vdots & \vdots \\ \boldsymbol{p}_{C-1}\boldsymbol{C}_0 & \boldsymbol{p}_{C-1}\boldsymbol{C}_1 & \cdots & \cdots & \boldsymbol{p}_{C-1}\boldsymbol{C}_{C-1} \end{pmatrix} \xrightarrow{\text{\small SHIFTCHANNEL}} \begin{pmatrix} \boldsymbol{p}_0\boldsymbol{C}_{C-1} & \boldsymbol{p}_0\boldsymbol{C}_{C-2} & \cdots & \cdots & \boldsymbol{p}_0\boldsymbol{C}_0 \\ \boldsymbol{p}_1\boldsymbol{C}_{C-2} & \boldsymbol{p}_1\boldsymbol{C}_{C-1} & \boldsymbol{p}_1\boldsymbol{C}_{C-2} & \cdots & \boldsymbol{p}_1\boldsymbol{C}_1 \\ \vdots & \vdots & \vdots & \vdots & \vdots \\ \boldsymbol{p}_{C-1}\boldsymbol{C}_0 & \cdots & \cdots & \cdots & \boldsymbol{p}_C\boldsymbol{C}_{C-1} \end{pmatrix} \tag{7}$$

Here, no element is zeroed out, as all channels can attend to all other channels. The channel shifting operation does not (to our knowledge) enjoy an implementation which as efficient as the time shifting one in PyTorch, but requires relatively complex index manipulation which cannot be streamlined.

As before, thanks to this shifting operation the channel attention component does not require recomputation for each channel position.

We provide a Triton implementation for both operations which is much more efficient and must be preferred when training a model.

## A.6 STRUCTURED ATTENTION DROPOUT

Dropout is a common technique to improve the generalization performance of neural networks. In Transformers, it is often applied inside the attention block to randomly zero-out some query-key attentions, to avoid over-reliance of the model on specific connections.

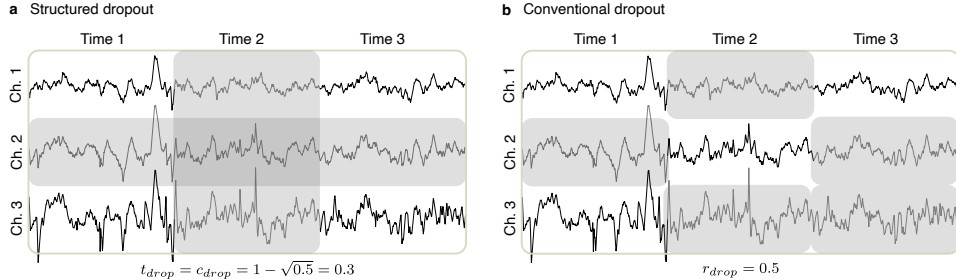

Figure 3: **Structured dropout. (a)** Our structured dropout blanks entire channels and time steps, to reduce the number of correlated segments. The dropout rate is computed to maintain the same number of dropped out segments as conventional dropout. **(b)** Conventional dropout blanks segments randomly. This is less effective with time-series because adjacent segments in time or space contain much of the same information.

Dropout usually applies to all elements with equal probability and creates uniform holes in the attention matrix. This is not efficient in the case of multi-variate time-series, as for each hole the neighboring segments are likely to carry very similar information, reducing dropout's effectiveness. We introduce a structured dropout technique which blanks entire channels and time points instead of individual segments. This technique is in principle much more effective by removing all segments which are more likely to be strongly correlated. We keep the same parameters as in conventional Dropout and compute the channel-specific and time-specific dropout rates as

$$t_{\text{drop}} = c_{\text{drop}} = 1 - \sqrt{1 - r_{\text{drop}}} \tag{8}$$

This ensures that approximately the same overall number of elements are zeroed (see Figure 3).

For the specific dropout rates and the location of the structured dropout layers refer to the description of the architecture in Appendix B.

## A.7 PERFORMANCE COMPARISON

To provide a clearer evaluation of the computational performance benefits of MVPA, and in particular FlashMVPA, we compare the inference speed and VRAM usage of multiple attention implementations. Specifically, we test:

- Naïve self-attention

- FlashAttention 2
- Linear attention (Nyströmformer; Xiong et al. 2021)
- MVPA
- FlashMVPA

We test all implementations with a batch size of 64 and a size of 768, with bfloat16 numeric type, to maintain a realistic scenario. We use 12 heads and no GQA, as it is not natively implemented for Nyströmformer. We vary both the number of time windows (T) and the number of channels (C) from 1 to 50, and report the runtime and memory consumption for a forward pass. We test the attention module in isolation to avoid introducing confounding variables, and we follow the best practices in GPU benchmarking.

Tables 8 and 9 show the full quadratic scaling of MVPA and the full self-attention. However, the tricks we employ for efficient computation still results in lesser memory usage for MVPA, albeit at a higher computational cost.

On the other hand, Tables 10 and 11 report the much more favorable scaling of the two dedicated implementations, with FlashMVPA having the same memory consumption as the more optimized FlashAttention 2.

Finally, Table 12 indicates that while Nyströmformer attention has theoretically favorable linear complexity, its implementation is much less performance than either FlashAttention 2 or Flash-MVPA.

Table 8: **Computational performance of MVPA**. Runtime and VRAM consumption of the naïve implementation of MVPA.

| T \ C | 1 | 10 | 20 | 30 | 40 | 50 |
|---|---|---|---|---|---|---|
| 1 | 1.56 us / 0.02 GB | 1.75 us / 0.05 GB | 1.90 us / 0.06 GB | 1.96 us / 0.07 GB | 2.03 us / 0.08 GB | 2.12 us / 0.10 GB |
| 10 | 1.72 us / 0.05 GB | 1.81 us / 0.18 GB | 2.59 us / 0.43 GB | 5.11 us / 0.81 GB | 7.53 us / 1.31 GB | 11.83 us / 1.92 GB |
| 20 | 1.69 us / 0.06 GB | 2.64 us / 0.43 GB | 7.59 us / 1.30 GB | 15.29 us / 2.64 GB | 25.30 us / 4.47 GB | 38.15 us / 6.75 GB |
| 30 | 1.77 us / 0.07 GB | 5.20 us / 0.81 GB | 15.27 us / 2.64 GB | 34.52 us / 5.54 GB | 67.64 us / 9.50 GB | 111.33 us / 14.52 GB |
| 40 | 1.84 us / 0.08 GB | 7.68 us / 1.31 GB | 25.15 us / 4.47 GB | 67.48 us / 9.50 GB | 108.71 us / 16.43 GB | 161.07 us / 25.25 GB |
| 50 | 1.74 us / 0.10 GB | 11.86 us / 1.92 GB | 37.77 us / 6.75 GB | 110.74 us / 14.52 GB | 160.72 us / 25.25 GB | 278.60 us / 38.90 GB |

Table 9: **Computational performance of self-attention**. Runtime and VRAM consumption of the naïve implementation of vanilla self-attention.

| T \ C | 1 | 10 | 20 | 30 | 40 | 50 |
|---|---|---|---|---|---|---|
| 1 | 1.08 us / 0.03 GB | 1.21 us / 0.05 GB | 1.15 us / 0.06 GB | 1.18 us / 0.06 GB | 1.18 us / 0.07 GB | 1.19 us / 0.08 GB |
| 10 | 1.17 us / 0.05 GB | 1.36 us / 0.16 GB | 2.17 us / 0.43 GB | 3.72 us / 0.84 GB | 4.93 us / 1.41 GB | 7.63 us / 2.11 GB |
| 20 | 1.16 us / 0.06 GB | 2.17 us / 0.43 GB | 4.93 us / 1.41 GB | 8.99 us / 2.97 GB | 14.37 us / 5.12 GB | 20.97 us / 7.85 GB |
| 30 | 1.21 us / 0.06 GB | 3.72 us / 0.84 GB | 8.99 us / 2.97 GB | 20.79 us / 6.41 GB | 35.93 us / 11.17 GB | 59.22 us / 17.25 GB |
| 40 | 1.24 us / 0.07 GB | 4.95 us / 1.41 GB | 14.35 us / 5.12 GB | 35.93 us / 11.17 GB | 56.17 us / 19.56 GB | 81.37 us / 30.31 GB |
| 50 | 1.21 us / 0.08 GB | 7.63 us / 2.11 GB | 20.98 us / 7.85 GB | 59.21 us / 17.25 GB | 81.38 us / 30.31 GB | 143.18 us / 47.03 GB |

Table 10: **Computational performance of FlashMVPA**. Runtime and VRAM consumption of the Triton implementation of MVPA.

| T \ C | 1 | 10 | 20 | 30 | 40 | 50 |
|---|---|---|---|---|---|---|
| 1 | 1.34 us / 0.02 GB | 1.40 us / 0.05 GB | 1.39 us / 0.06 GB | 1.42 us / 0.07 GB | 1.41 us / 0.08 GB | 1.49 us / 0.09 GB |
| 10 | 1.41 us / 0.05 GB | 1.45 us / 0.14 GB | 1.78 us / 0.24 GB | 2.83 us / 0.34 GB | 4.10 us / 0.45 GB | 4.53 us / 0.56 GB |
| 20 | 1.42 us / 0.06 GB | 1.69 us / 0.24 GB | 4.09 us / 0.44 GB | 5.66 us / 0.65 GB | 10.04 us / 0.87 GB | 12.10 us / 1.09 GB |
| 30 | 1.42 us / 0.07 GB | 2.92 us / 0.34 GB | 6.00 us / 0.65 GB | 12.09 us / 0.96 GB | 18.93 us / 1.29 GB | 23.98 us / 1.63 GB |
| 40 | 1.47 us / 0.08 GB | 3.98 us / 0.45 GB | 12.43 us / 0.87 GB | 16.17 us / 1.29 GB | 27.29 us / 1.73 GB | 37.68 us / 2.19 GB |
| 50 | 1.44 us / 0.09 GB | 4.79 us / 0.55 GB | 11.69 us / 1.08 GB | 23.45 us / 1.62 GB | 36.28 us / 2.19 GB | 60.36 us / 2.75 GB |

Table 11: **Computational performance of FlashAttention 2**. Runtime and VRAM consumption of FlashAttention 2, a custom CUDA implementation of self-attention.

| T \ C | 1 | 10 | 20 | 30 | 40 | 50 |
|---|---|---|---|---|---|---|
| 1 | 1.09 us / 0.04 GB | 1.08 us / 0.05 GB | 1.08 us / 0.06 GB | 1.11 us / 0.06 GB | 1.11 us / 0.07 GB | 1.12 us / 0.08 GB |
| 10 | 1.10 us / 0.05 GB | 1.15 us / 0.12 GB | 1.29 us / 0.21 GB | 1.68 us / 0.30 GB | 2.09 us / 0.39 GB | 2.40 us / 0.47 GB |
| 20 | 1.10 us / 0.06 GB | 1.30 us / 0.21 GB | 2.07 us / 0.39 GB | 4.56 us / 0.55 GB | 4.70 us / 0.73 GB | 5.16 us / 0.90 GB |
| 30 | 1.12 us / 0.06 GB | 1.66 us / 0.30 GB | 5.27 us / 0.55 GB | 4.73 us / 0.81 GB | 6.17 us / 1.07 GB | 8.11 us / 1.33 GB |
| 40 | 1.17 us / 0.07 GB | 2.14 us / 0.39 GB | 4.71 us / 0.73 GB | 6.19 us / 1.07 GB | 8.46 us / 1.41 GB | 10.69 us / 1.76 GB |
| 50 | 1.15 us / 0.08 GB | 2.41 us / 0.47 GB | 5.07 us / 0.90 GB | 8.16 us / 1.33 GB | 10.64 us / 1.76 GB | 13.99 us / 2.19 GB |

Table 12: **Computational performance of Nyströmformer**. Runtime and VRAM consumption of the linear attention from Nyströmformer Xiong et al. (2021).

| T \ C | 1 | 10 | 20 | 30 | 40 | 50 |
|---|---|---|---|---|---|---|
| 1 | 0.62 us / 0.03 GB | 0.72 us / 0.05 GB | 0.70 us / 0.06 GB | 0.72 us / 0.07 GB | 0.73 us / 0.08 GB | 0.73 us / 0.09 GB |
| 10 | 0.73 us / 0.05 GB | 1.56 us / 0.15 GB | 2.62 us / 0.32 GB | 4.44 us / 0.55 GB | 5.63 us / 0.85 GB | 8.00 us / 1.19 GB |
| 20 | 0.72 us / 0.06 GB | 2.62 us / 0.32 GB | 5.63 us / 0.85 GB | 8.95 us / 1.60 GB | 12.52 us / 2.60 GB | 16.61 us / 3.81 GB |
| 30 | 0.71 us / 0.07 GB | 4.44 us / 0.55 GB | 8.95 us / 1.60 GB | 17.73 us / 3.17 GB | 31.13 us / 5.27 GB | 57.89 us / 7.90 GB |
| 40 | 0.76 us / 0.08 GB | 5.63 us / 0.85 GB | 12.51 us / 2.60 GB | 31.16 us / 5.27 GB | 43.59 us / 8.89 GB | 57.37 us / 13.45 GB |
| 50 | 0.74 us / 0.09 GB | 7.99 us / 1.19 GB | 16.63 us / 3.81 GB | 57.85 us / 7.90 GB | 57.35 us / 13.45 GB | 122.60 us / 20.46 GB |

## B  MVPFORMER ARCHITECTURE

MVPFormer is part of a family of predictive deep learning models with 74 million (MVPFormer-S) to 1.2 billion (MVPFormer-M, or simply MVPFormer for brevity) parameters based on the Transformer (Vaswani et al., 2017) architecture, capable of generating iEEG signals.

**Wavelet encoder**  The first processing step maps the raw iEEG signal to continuous embeddings. We begin by partitioning the raw data into segments of five seconds. Each segment passes independently through a db4 wavelet decomposition, which has been shown to be highly effective for biosignals (Adeli et al., 2003; Shen et al., 2022). Depending on the model's overall size, it is then linearly projected onto a smaller latent space. This projection, or feature vector, is the embedding. Our method is inspired by wav2vec (Schneider et al., 2019), though we use learnable embeddings. We apply the encoding channel-wise, meaning each segment remains one-dimensional.

**Decoder**  MVPFormer is based on the Llama2 architecture (Touvron et al., 2023) with parallel attention and MLP blocks inspired by Megatron-LM (Shoeybi et al., 2019). This choice was informed by the selection of a generative model powerful enough to process brain iEEG signals and computationally light enough to enable extensive testing. We provide two models to evaluate the scaling of our foundational model: MVPFormer-S with 75M parameters and MVPFormer-M (or simply MVPFormer) with 1.2B parameters.

See Tables 13 and 14 for a breakdown of the models sizes and hyperparameters.

Table 13: **Breakdown of the parameters of MVPFormer-S.** The dimensions are indicated for each of the components of MVPFormer-S.

| Transformer | Encoder | Signal |
|---|---|---|
| $n_{\text{layers}} \leftarrow 12$ | $n_{\text{input}} \leftarrow 2560$ | $w_{\text{length}} \leftarrow 500\,\text{s}$ |
| $n_{\text{heads}} \leftarrow 12$ | $n_{\text{embed}} \leftarrow 768$ | $n_{\text{segments}} \leftarrow 100$ |
| $n_{\text{gqa}} \leftarrow 4$ | | $w_{\text{segment}} \leftarrow 5\text{s}$ |
| $n_{\text{embed}} \leftarrow 768$ | | $n_{\text{negatives}} \leftarrow 30$ |
| $n_{\text{inner}} \leftarrow 1728$ | | $n_{\text{local}} \leftarrow 10$ |
| $r_{\text{drop}} \leftarrow 0.1$ | | |

Table 14: **Breakdown of the parameters of MVPFormer-M.** The dimensions are indicated for each of the components of MVPFormer-M.

| Transformer | Encoder | Signal |
|---|---|---|
| $n_{\text{layers}} \leftarrow 24$ | $n_{\text{input}} \leftarrow 2560$ | $w_{\text{length}} \leftarrow 500\,\text{s}$ |
| $n_{\text{heads}} \leftarrow 16$ | $n_{\text{embed}} \leftarrow 1024$ | $n_{\text{segments}} \leftarrow 100$ |
| $n_{\text{gqa}} \leftarrow 8$ | | $w_{\text{segment}} \leftarrow 5\text{s}$ |
| $n_{\text{embed}} \leftarrow 2048$ | | $n_{\text{negatives}} \leftarrow 30$ |
| $n_{\text{inner}} \leftarrow 5362$ | | $n_{\text{local}} \leftarrow 10$ |
| $r_{\text{drop}} \leftarrow 0.1$ | | |

### B.1  INFERENCE

The full end-to-end inference procedure is reported in Algorithm 3.

### B.2  ENCODER

The Encoder block is detailed above. The algorithmic overview is presented in Algorithm 4.

---

**Algorithm 3:** Full inference with MVPFormer

---

**Input:** $x \in \mathbb{R}^{C \times T}$ raw input; $C, T$ number of channels and length resp.; $n_{\text{layers}} = 12$
**Data:** $c \in [1, \ldots, C]; t \in [1, \ldots, T//n_{\text{segments}} + 1]$
**Output:** $o_{c,(t-1)} \in \mathbb{R}^{n_{\text{embed}}}$ generated embedding; $n_{\text{embed}} = 768$

1 **def** `Inference`$(z_{c,t})$:
2      $x_{c,t} \leftarrow \text{SEGMENT}(x)$
3      $e_{c,t} \leftarrow \text{ENCODER}(x_{c,t})$
4      **for** $l \leftarrow 1$ **to** $n_{layers}$ **do**
5          $e_{c,t} \leftarrow \text{DECODER}(e_{c,t})$
6      $o_{c,(t-1)} \leftarrow e_{c,t}$
7      **return** $o_{c,(t-1)}$

---

**Algorithm 4:** Encoder block of MVPFormer

---

**Input:** $x_{c,t} \in \mathbb{R}^{n_{\text{input}}}$ raw input segment; $n_{\text{input}} = 2560$; $c \in [1, \ldots, C]; t \in [1, \ldots, T]$
**Output:** $o_{c,t} \in \mathbb{R}^{n_{\text{embed}}}$ output token; $n_{\text{embed}} = 768$
**Data:** $l = 8$ maximum decomposition level given $n_{\text{input}}$

1 **def** `Encoder`$(x_{c,t})$:
2      $d_{c,t} \leftarrow \text{DISCRETEWAVELETDECOMPOSITION}_{db4}(x_{c,t}, l)$
3      $z_{c,t} \leftarrow \text{RMSNORM}(d_{c,t})$
4      $o_{c,t} \leftarrow \text{LINEAR}(z_{c,t})$
5      **return** $o_{c,t}$

---

### B.3 DECODER

The collection of vectors resulting from the Encoder block is flattened into a 1D sequence to provide a unified input interface to the Transformer decoder blocks, consistent with conventional Transformers. All the encoded segments corresponding to a window form the input to the Transformer module, which computes the MVPA among all the segments. The segments are sequentially processed by multiple Transformer layers, composed of attention and MLP blocks in a deep network configuration. The attention blocks are masked to guarantee that MVPFormer only has access to past segments to generate the target. The model produces one output embedding for each input segment. The algorithmic overview is presented in Algorithm 5, while the MLP block in Algorithm 6.

---

**Algorithm 5:** Decoder block of MVPFormer

---

**Input:** $o_{c,t} \in \mathbb{R}^{n_{\text{embed}}}$ input tokens; $n_{\text{embed}} = 768$
**Output:** $o_{c,t} \in \mathbb{R}^{n_{\text{embed}}}$

1 **def** `Decoder`$(o_{c,t})$:
2      $z_{c,t} \leftarrow \text{RMSNORM}(o_{c,t})$
     `# Compute attention`
3      $a_{c,t} \leftarrow \text{MVPATTENTION}(z_{c,t})$
4      $d_{c,t} \leftarrow \text{DROPOUT}(\text{LINEARNOBIAS}((a_{c,t}))$
     `# Compute feedforward residuals in parallel with`
     `  attention (Wang, 2021)`
5      $s_{c,t} \leftarrow \text{MLP}(z_{c,t})$
     `# Sum residuals and attention`
6      $o_{c,t} \leftarrow o_{c,t} + d_{c,t} + s_{c,t}$
7      **return** $o_{c,t}$

---

---

**Algorithm 6:** MLP block of MVPFormer

---

**Input:** $\boldsymbol{z}_{c,t} \in \mathbb{R}^{n_{\text{embed}}}$ normalised Decoder output; $n_{\text{embed}} = 768$
**Data:** $\boldsymbol{u}_{c,t}, \boldsymbol{g}_{c,t} \in \mathbb{R}^{n_{\text{inner}}}$; $n_{\text{inner}} = 1728$
**Output:** $\boldsymbol{s}_{c,t} \in \mathbb{R}^{n_{\text{embed}}}$

1 **def** MLP $(\boldsymbol{z}_{c,t})$:
2     $\boldsymbol{u}_{c,t} \leftarrow \text{LINEARNOBIAS}(\boldsymbol{z}_{c,t})$
3     $\boldsymbol{g}_{c,t} \leftarrow \text{SILU}(\text{LINEARNOBIAS}(\boldsymbol{z}_{c,t}))$
4     $\boldsymbol{s}_{c,t} \leftarrow \text{LINEARNOBIAS}(\boldsymbol{u}_{c,t} + \boldsymbol{g}_{c,t})$
5     **return** $\boldsymbol{s}_{c,t}$

---

## C    DETAILS ON TRAINING

### C.1    GENERATIVE PRE-TRAINING

MVPFormer is used to generate neuronal activity while in the base prediction task. During training, the target for each output is the successive input segment in time, not in space. First, we divide each recording into windows of 500 seconds each, with a stride of five seconds. Then, each window is divided into 100 segments (each five seconds long), yielding a total of 39B total training segments.

For each target, we sample random input segments from the rest of the batched windows to create the confounding targets $Z = \{z_1, ..., z_n\}$. These segments still represent actual iEEG signals, so they are plausible, but they are expected to be very different from the true target.

This scheme strikes the correct balance between too much similarity and too little. The objective of MVPFormer is to generate future iEEG signals, so we choose a contrastive loss to increase the cosine similarity of its output with the true target, while decreasing it with the confounding targets. As training progresses, MVPFormer starts to produce outputs that look like encoded segments, i.e., its inputs. MVPFormer becomes more and more capable of choosing the right target and thus is able to predict the future signal.

**Extraction of positive and negative examples**    Out of the entire dataset, $B$ windows are chosen at random to form a batch. Each window $W_{i \in [1..B]}$ has an arbitrary sample rate and $C_i$ channels. First, the sampling rate is normalized to 512 Hz, then the windows are divided into $T$ non-overlapping segments per-channel, leaving us with $C_i \times T$ segments per window. Each segment is passed in parallel through the encoder. For the sake of simplicity, suppose one window $W^*$ (with $C^*$ channels) is selected at random as the positive window, and all the others as the confounding windows. The embeddings of $W^*$ form the input context $E$ with length $C^* \times T$.

For each segment, $n$ embeddings are selected at random from the confounding windows to form the negative samples $Z$. Each $Z_{c,t}$ has $n$ elements, thus $Z$ has size $C^* \times T \times n$. $Z$ is excluded from backpropagation.

MVPFormer processes the entire $E$ at once and produces an output $O$ also of size $C^* \times T$. We then compute the losses and iteratively optimize to train the model.

**Contrastive loss**    We train MVPFormer using a contrastive loss (Oord et al., 2018) and an auxiliary loss. To compute the contrastive loss, we rely on having other windows in the batch, so a larger batch size leads in general to a more stable training and better generalization performance. Let $e^i, \ i \in [1, \ldots, B]$ be the outputs of the signal Encoder and $o^i, \ i \in [1, \ldots, B]$ the outputs of the Decoder stack, for $B$ the batch size. For each $i^*$, we select at random $n_{\text{negatives}}$ elements from $e^i, i \neq i^*$ to act as our negative samples $n^{i*}$. Clearly, the bigger the batch the greater the entropy. We compute the contrastive loss for each $i^*$ as follows:

$$\mathcal{L}_{c,t}^i = -\log \frac{\exp(\text{sim}(\boldsymbol{o}_{c,t}, \boldsymbol{e}_{c(t+1)})/\tau)}{\sum_{\boldsymbol{z}_k \in Z} \exp(\text{sim}(\boldsymbol{o}_{c,t}, \boldsymbol{z}_k)/\tau)} \tag{9}$$

Finally summing over every $i$, $c$, $t$ gives us the optimization target for the generative task.

The loss is invariant to the channel $c$, which encourages all the outputs to be the same regardless of channel. The temperature $\tau$ is 0.1.

### C.2    GENERATION OF NEURONAL ACTIVITY

The generation of brain signals during inference proceeds analogously as during training. However, we do not have access to the same source of entropy as in training since the batches are limited to one subject at a time. This limitation implies that the evaluation scores of MVPFormer must be more punishing than the training objective, since we cannot reliably estimate the accuracy with which MVPFormer chooses the right target. For this reason, we measure the cosine similarity directly in a three-way reference scheme. First, we consider the cosine similarity of the output with the true target. Second, we consider the similarity with the maximally correlated target. Third, we measure the cosine similarity with the highest form of entropy available, random segments in the batch that

are still close by in time. This measurement ensures that the difference in similarity between the true and confounding targets remains significant.

# D    SWEC iEEG DATASET

The SWEC iEEG dataset is presented in Section 4. The iEEG signals were recorded intracranially with a sampling rate of either 512 Hz or 1024 Hz, which was then normalized to 512 Hz before training MVPFormer. The signals were median-referenced and digitally band-pass filtered between 0.5 and 120 Hz using a fourth-order Butterworth filter, both in a forward and backward pass to minimize phase distortions. All the recordings were inspected by an expert for identification of seizure onsets and offsets, and to remove channels corrupted by artifacts.

This dataset may only be used for research. For other applications any liability is denied. In particular, the dataset must not be used for diagnostic purposes.

Here, Table 15 shows the full details of the dataset in a subject-by-subject breakdown. Finally, Figure 4 shows two annotated seizures in the dataset.

Table 15: **Per-subject details of our SWEC iEEG dataset.** Ch. is the number of electrodes, $f_s$ is the sampling frequency in Hz, Rec. [h] is the length of the recording in hours, and Ev. is the number of seizures. The entire dataset contains 68 subjects, 9328 hours of recording and 704 ictal events.

| Subject | Ch. | $f_s$ [Hz] | Rec [h] | Ev. | Subject | Ch. | $f_s$ [Hz] | Rec [h] | Ev. | Subject | Ch. | $f_s$ [Hz] | Rec [h] | Ev. |
|---|---|---|---|---|---|---|---|---|---|---|---|---|---|---|
| ID01 | 88 | 512 | 293.4 | 2 | ID24 | 32 | 1024 | 40.7 | 14 | ID47 | 32 | 1024 | 330.4 | 3 |
| ID02 | 66 | 512 | 235.2 | 2 | ID25 | 128 | 512 | 109.4 | 4 | ID48 | 57 | 1024 | 28.4 | 6 |
| ID03 | 64 | 512 | 158.4 | 4 | ID26 | 34 | 1024 | 87.6 | 1 | ID49 | 60 | 512 | 140.4 | 6 |
| ID04 | 32 | 1024 | 40.7 | 14 | ID27 | 32 | 1024 | 146 | 8 | ID50 | 64 | 1024 | 177.2 | 2 |
| ID05 | 128 | 512 | 109.4 | 4 | ID28 | 75 | 512 | 69 | 4 | ID51 | 89 | 512 | 161.5 | 1 |
| ID06 | 32 | 1024 | 146 | 8 | ID29 | 61 | 1024 | 143.8 | 70 | ID52 | 69 | 512 | 112.6 | 2 |
| ID07 | 75 | 512 | 69 | 4 | ID30 | 48 | 1024 | 40.9 | 27 | ID53 | 22 | 1024 | 134.9 | 1 |
| ID08 | 61 | 1024 | 143.8 | 70 | ID31 | 32 | 1024 | 42.4 | 17 | ID54 | 54 | 1024 | 202 | 3 |
| ID09 | 48 | 1024 | 40.9 | 27 | ID32 | 32 | 1024 | 212.2 | 2 | ID55 | 24 | 1024 | 152.1 | 2 |
| ID10 | 32 | 1024 | 42.4 | 17 | ID33 | 104 | 512 | 53.6 | 1 | ID56 | 62 | 1024 | 130.5 | 3 |
| ID11 | 32 | 1024 | 212.2 | 2 | ID34 | 56 | 1024 | 191.4 | 9 | ID57 | 40 | 1024 | 90.7 | 12 |
| ID12 | 56 | 1024 | 191.4 | 9 | ID35 | 64 | 1024 | 104 | 7 | ID58 | 92 | 512 | 138.2 | 7 |
| ID13 | 64 | 1024 | 104 | 7 | ID36 | 24 | 1024 | 161.4 | 60 | ID59 | 54 | 1024 | 107.3 | 15 |
| ID14 | 24 | 1024 | 161.4 | 60 | ID37 | 98 | 512 | 195.9 | 2 | ID60 | 74 | 512 | 50.7 | 8 |
| ID15 | 98 | 512 | 195.9 | 2 | ID38 | 34 | 1024 | 177.1 | 5 | ID61 | 76 | 512 | 89.6 | 6 |
| ID16 | 34 | 1024 | 177.1 | 5 | ID39 | 60 | 1024 | 129.6 | 2 | ID62 | 60 | 1024 | 235.1 | 7 |
| ID17 | 60 | 1024 | 129.6 | 2 | ID40 | 42 | 1024 | 205.1 | 5 | ID63 | 64 | 512 | 179.8 | 4 |
| ID18 | 42 | 1024 | 205.1 | 5 | ID41 | 33 | 1024 | 82.7 | 3 | ID64 | 56 | 1024 | 36.3 | 20 |
| ID19 | 29 | 1024 | 21.7 | 25 | ID42 | 63 | 1024 | 87.8 | 2 | ID65 | 49 | 1024 | 139.7 | 8 |
| ID20 | 88 | 512 | 293.4 | 2 | ID43 | 126 | 512 | 63.2 | 2 | ID66 | 39 | 1024 | 212.3 | 2 |
| ID21 | 66 | 512 | 235.2 | 2 | ID44 | 60 | 1024 | 150.3 | 2 | ID67 | 63 | 512 | 111.7 | 4 |
| ID22 | 64 | 512 | 158.4 | 4 | ID45 | 47 | 1024 | 157.3 | 1 | ID68 | 32 | 1024 | 167.8 | 3 |
| ID23 | 32 | 1024 | 42.4 | 33 | ID46 | 86 | 512 | 140.5 | 21 | | | | | |

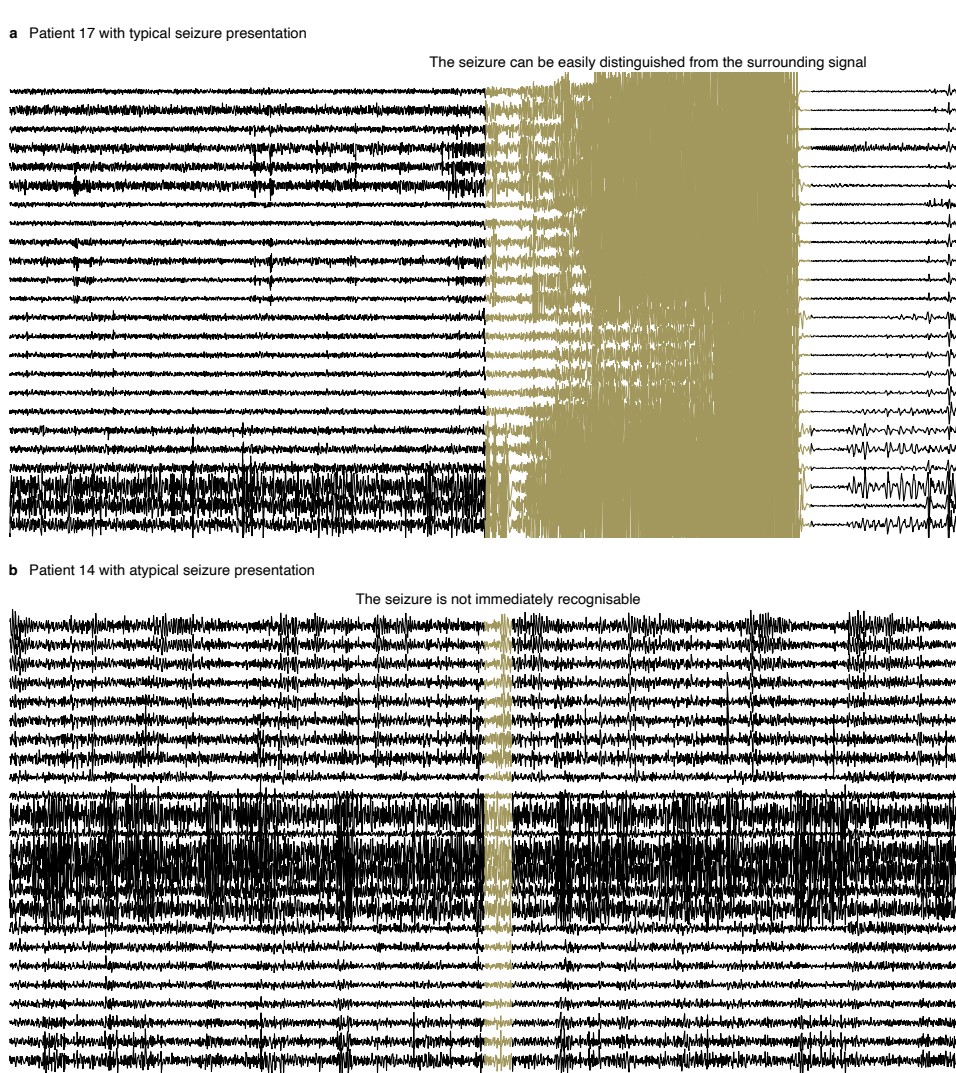

Figure 4: **iEEG activity of two patients with different ictal patterns. (a)** Patient 17 of the SWEC iEEG dataset presents typical ictal events. The seizure can be clearly distinguished even by a non-expert, and MVPFormer performs very well on this patient. The number of channels is reduced from the original recording to facilitate comparison with the more difficult presentation. **(b)** Patient 14 of the SWEC iEEG dataset does not have typical events. The neuronal activity during seizures for this patient cannot be clearly distinguished, and assessment by experts would diverge considerably. As expected, MVPFormer has a high level of disagreement on this patient. All the channels of the original recording are presented to exclude the chance of some channels carrying additional information.

# E    GENERATION OF RESULTS

During evaluation, the target and data are prepared according to each task and model's specification, analogously to the pre-training task. For MVPFormer, we first divide each recording into windows of 500 seconds each, with a stride of five seconds. Then, each window is divided into 100 segments (each five seconds long). In cases where datasets are too small for MVPFormer's context window, we shorten it accordingly. The classification decision is taken on the last window.

## E.1    CHANNEL SELECTION

In order to ensure a consistent setup across all baselines and to speed up evaluation we perform the seizure detection task on the SWEC iEEG dataset using a subset of the channels (see Appendix G.8 for an ablation of the selection mechanism). Specifically, we always choose 32 channels to comply with the fixed-channel models such as Brant-2.

First, we compute the variance and kurtosis for all channels within the first 30 minutes of each subject's recording. We exclude channels based on the following:

- Variance above the 99th percentile and below the 1st percentile
- Kurtosis above the 95th percentile

This results in a first quality filtering. Next, we rank the channels based on a simple combination of variance and kurtosis with the following:

$$r_C = \frac{\text{var}(C)}{1 + \text{kurt}(C)} \tag{10}$$

and choose the first 32 channels. In the cases where there not enough channels, we also include some of the channels excluded above to reach 32.

## E.2    EPISODIC SEIZURE POST-PROCESSING

For episodic evaluation we apply three post-processing steps to the model output:

- Merge events happening within 5 minutes of each other
- Remove events shorter than 20 seconds in length
- Remove events with less than 5 positive responses

Moreover, when the subject has multiple seizures in one minute we merge them into one.

## E.3    ONLINE SEIZURE THRESHOLDING

In the clinical evaluation setup we apply a simple thresholding to decide whether to report a seizure or not. We set 3 positive seconds out of 10 to be the lower limit for detecting a seizure, to deter false positives; events shorter than 3 seconds are thus not reported, and an additional latency of 3 seconds is to be considered. We find this trade-off has limited drawbacks in practice, as there is often large disagreement even among neurologists about very short events.

## E.4    KAPPA SCORE ESTIMATION

To estimate the Kappa score, we choose 300 random segments per subject to compare their classification from MVPFormer and the labels. We perform multiple iterations to ensure no bias in this computation. Figure 5 indicates that our choice of 250 iterations is sufficient for stable results.

E.5   LANDIS AND KOCH CRITERIA

Table 16 reports the commonly used Landis and Koch criteria for qualitative inter-rater agreement estimation from Kappa scores.

| Kappa | Agreement |
|---|---|
| 0 – 0.20 | Slight |
| 0.21 – 0.40 | Fair |
| 0.41 – 0.60 | Moderate |
| 0.61 – 0.80 | Substantial |
| 0.81 – 1.00 | Almost perfect |

Table 16: **Landis and Koch criteria.** Landis and Koch criteria (Landis & Koch, 1977) for evaluating Cohen's kappa in the context of inter-rater agreement between human experts on seizure classification.

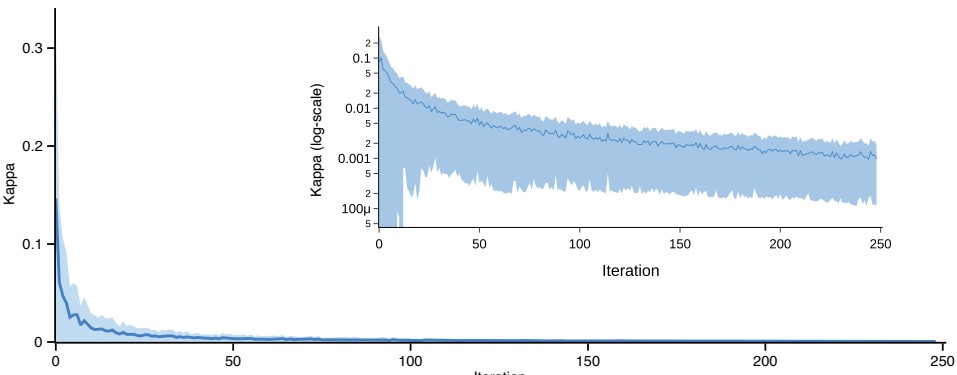

Figure 5: **Mean absolute error in Cohen's kappa estimation.** Our estimation scheme for Cohen's kappa converges after very few iterations. The error is computed per-subject as the absolute difference between the running averages at each two consecutive iterations; the running average is the average of all preceding steps. The average and standard deviation across all subjects is reported here. We compute up to 250 random iterations to ensure precise reporting.

# F PREDICTION OF iEEG SIGNALS

We evaluate the effects of the size of the model and the attention mechanism on the iEEG prediction task. Figure 6 shows that both MVPA and the vanilla attention are effective at predicting the next brain states. Scaling up the model size from MVPFormer-S to MVPFormer-M has the effect of shortening the tail of the true distribution, effectively increasing the concentration of the cosine similarity towards the maximum.

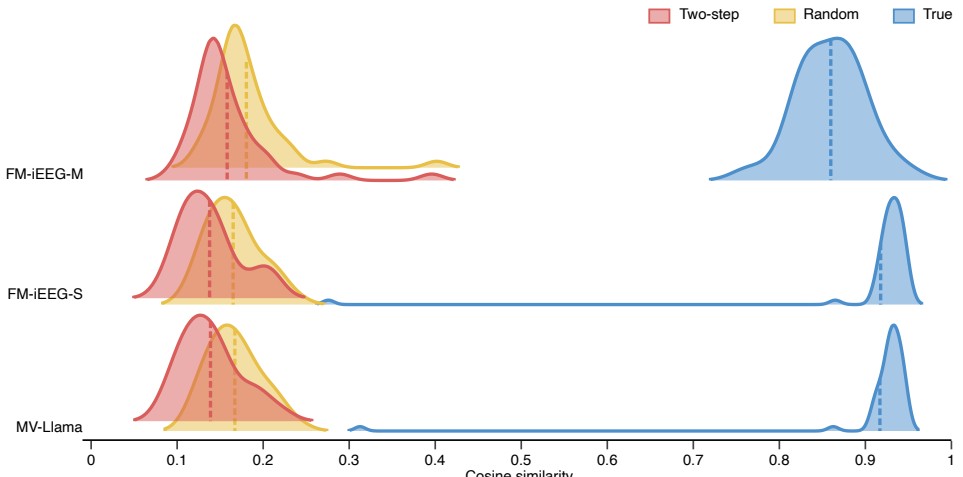

Figure 6: **Performance of MVPFormer-M, MVPFormer-S, and MV-Llama on the iEEG prediction task.** We report the average two-step, random, and true target cosine similarities for the three different models. All three are effective at predicting iEEG activity, while the larger model takes advantage of the increased embedding size by increasing the concentration of cosine similarities towards 1.

Complete details are available for the iEEG prediction performance of MVPFormer-M (see Appendix F.1, MVPFormer-S (see Appendix F.2), and MV-Llama (see Appendix F.3).

### F.1 PREDICTION OF iEEG AND ICTAL ACTIVITY

MVPFormer is primarily a neuronal prediction model, trained to generate neuronal activity regardless of whether such activity is pathological or physiological. To understand the behavior of MVPFormer with anomalous brain states, we evaluate its performance in generating ictal neuronal activity. The precise relationship between ictal and interictal states is a point of ongoing discussion (Beenhakker & Huguenard, 2009; Zaveri et al., 2020), but many consider an approach to seizures as anomalies (Martini et al., 2021) the most appropriate. The SWEC iEEG dataset contains many ictal events, so we are able to evaluate the performance of MVPFormer in generating anomalous activity. In particular, in this dataset the ratio between non-ictal and ictal states is approximately 500:1.

Figure 7 shows that ictal states are not anomalous for MVPFormer. In particular, the prediction similarity of MVPFormer does not degrade when generating ictal activity. Moreover, the prediction similarity in the ictal state is neither significantly different from the average similarity nor from the non-ictal similarity. This indicates that MVPFormer's understanding of the mechanisms of generation of neuronal activity encompasses the pathological ictal state as well. Therefore, MVPFormer must model patterns found both in physiological and pathological brain states. Finally, MVPFormer incorporates a model of seizure generation as a by-product of its predictive task, which is particularly noteworthy.

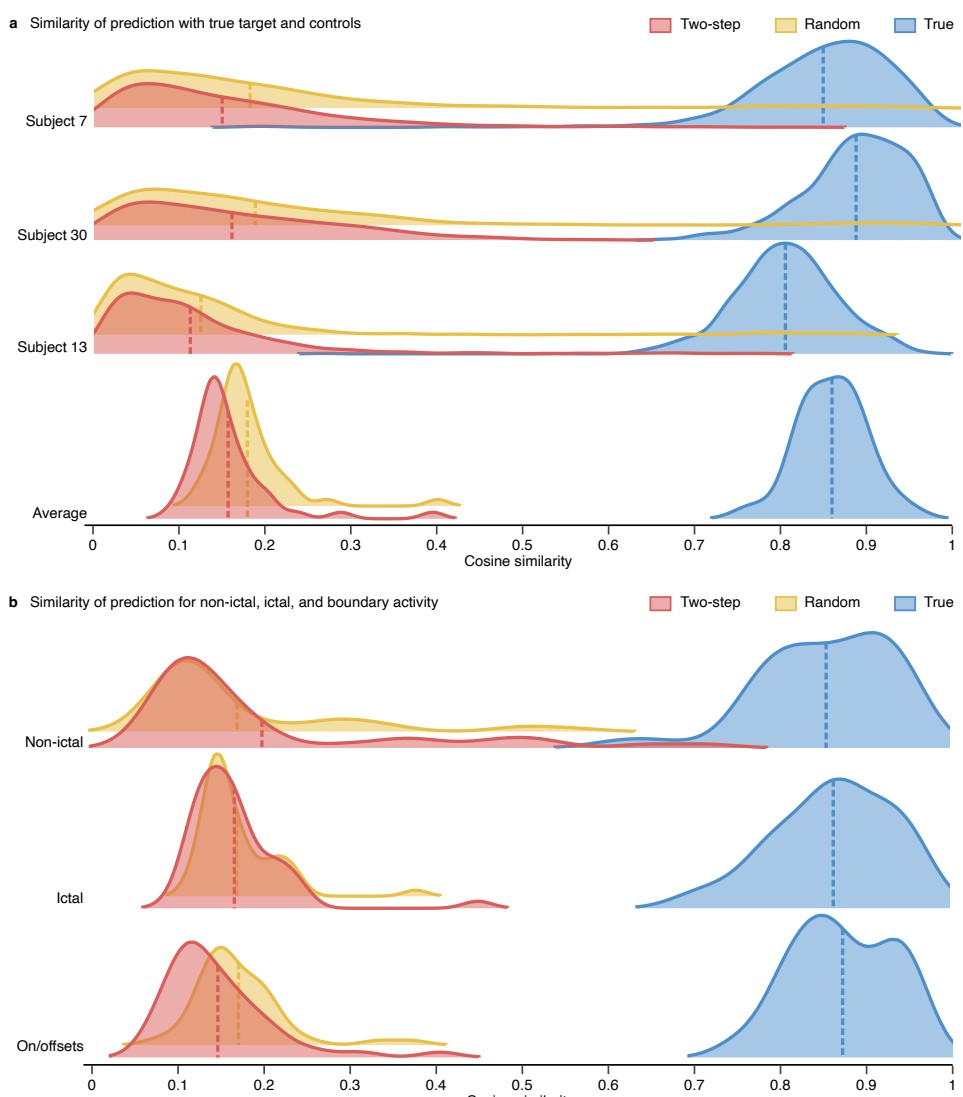

Figure 7: **Performance of MVPFormer on the prediction task.** (a) A three-reference evaluation scheme is used to assess MVPFormer's performance. The true target is the immediate future, i.e. the next five seconds of iEEG signal. The two-step target is the signal twice removed in the future, i.e. the five seconds of iEEG signal coming after the true target. Finally, the random target is chosen from iEEG signals which are close by in time with the true target. The distribution of the average similarity across the entire recording is shown together with the similarity within three representative subjects (with maximum, median, and minimum average similarity). (b) The prediction similarity is computed again for all three targets, distinguishing between targets which lie within an ictal event, without, or at the boundary. There is no significant difference in the performance of MVPFormer in predicting ictal or non-ictal activity, indicating that MVPFormer can encompass anomalous brain states as well, together with the transitions between physiological and anomalous.

## F.2 EFFECTS OF THE SCALE OF THE MODEL

Figure 8 shows the full details on the performance of MVPFormer-S on the SWEC iEEG dataset in the iEEG prediction task.

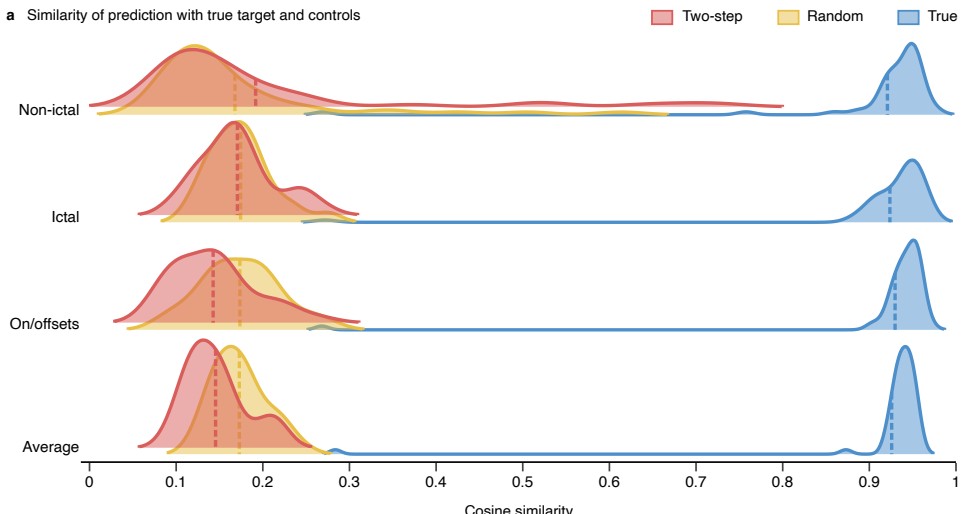

Figure 8: **Performance of MVPFormer-S on the prediction task.** Performance of MVPFormer-S in generating neuronal activity of unseen test subjects. The prediction similarity is computed for all three targets, distinguishing between targets which lie within an ictal event or without. There is no significant difference in the performance of MVPFormer-S in predicting ictal or non-ictal activity.

## F.3 EFFECTS OF THE ATTENTION MECHANISM

Figure 9 shows that vanilla attention is also effective in predicting the development of iEEG signal.

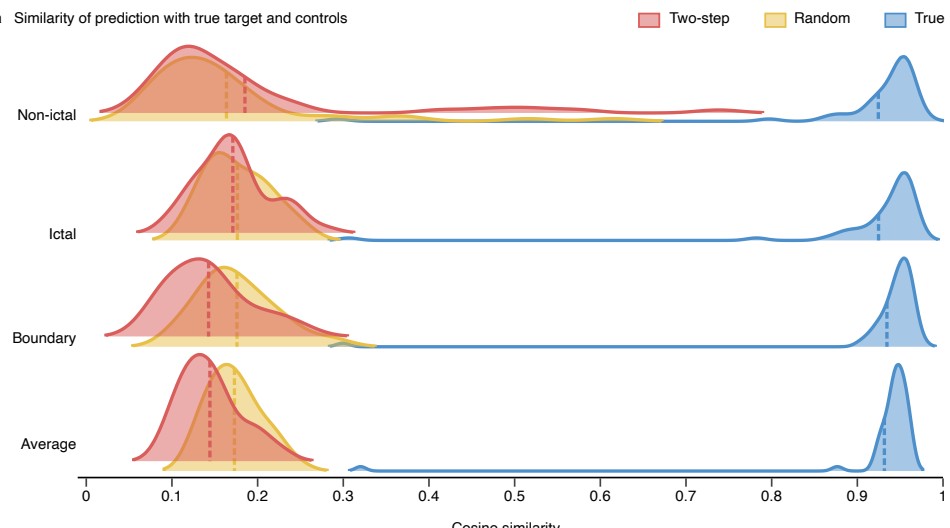

Figure 9: **Performance of MV-Llama on the prediction task.** The prediction similarity is computed for all three targets, distinguishing between targets which lie within an ictal event or without. There is no significant difference in the performance of MV-Llama in predicting ictal or non-ictal activity. This indicates that vanilla attention with a proper positional encoding scheme can effectively generate neuronal activity. However, this does not translate to improved performance in the seizure classification task (see Table 21 vs. Table 20)

### F.4 Per-subject cosine similarity

We provide a detailed per-subject breakdown of the maximum cosine similarity measure for MVP-Former. Figure 10 shows the per-patient global similarity. Figure 11 shows the per-patient similarity within an anomaly. Figure 12 shows the per-patient similarity at the boundary of an anomaly.

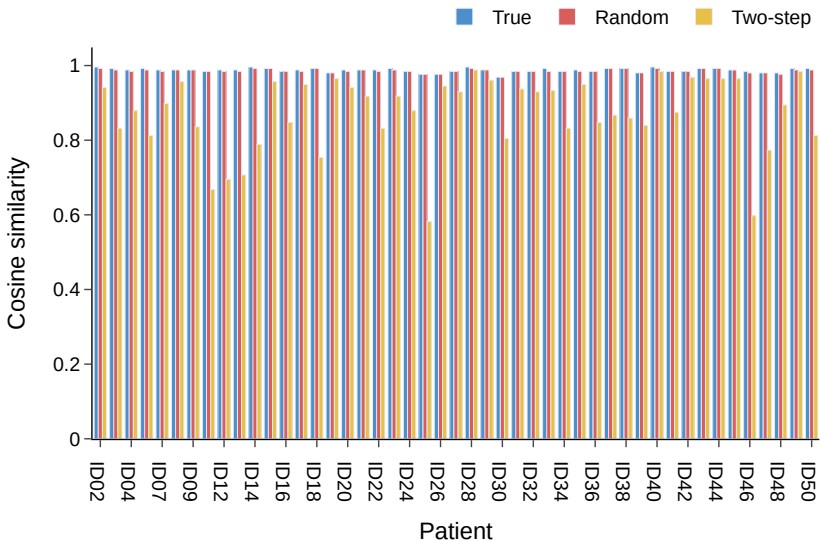

Figure 10: **Breakdown of total cosine similarity per-patient.** Maximum cosine similarity of MVP-Former's output with the true, random, and two-step targets over the entire SWEC iEEG dataset. The data is shown patient-by-patient.

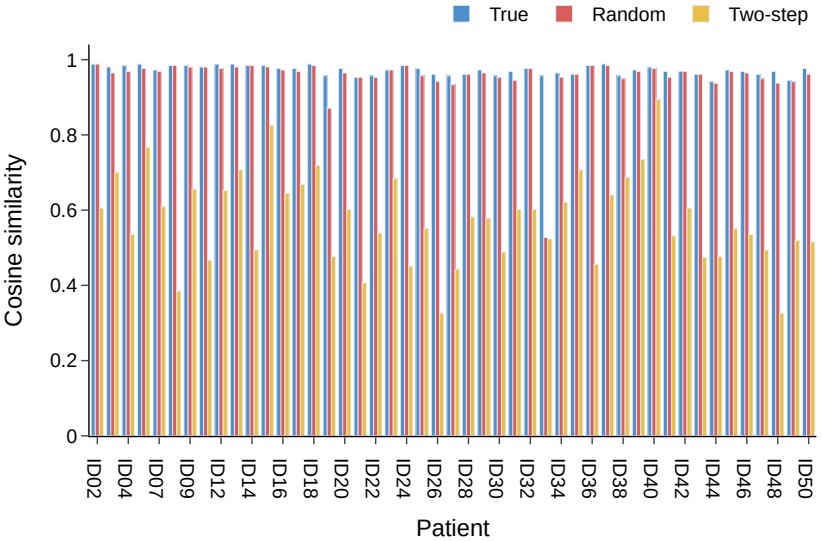

Figure 11: **Breakdown of anomaly cosine similarity per-patient.** Maximum cosine similarity of MVPFormer's output with the true, random, and two-step targets while within an anomaly (seizure) in the SWEC iEEG dataset. The data is shown patient-by-patient.

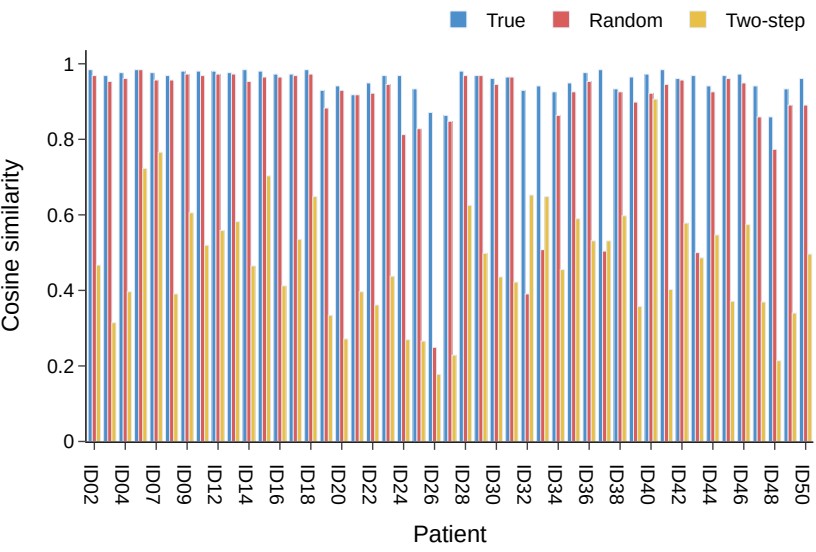

Figure 12: **Breakdown of boundary cosine similarity per-patient.** Maximum cosine similarity of MVPFormer's output with the true, random, and two-step targets while in the onset and offset zones of seizures in the SWEC iEEG dataset. The data is shown patient-by-patient.

# G ADDITIONAL RESULTS

## G.1 SEIZURE DETECTION

Table 17 reports a summary of the seizure detection results across all datasets and architectures.

Table 17: **Results on seizure detection.** We compare MVPFormer with Brant-2, the current SOTA Transformer model for iEEG, and MV-Llama, our vanilla attention-based baseline.

| | | SWEC | | | | MAYO | | | FNUSA | | |
|---|---|---|---|---|---|---|---|---|---|---|---|
| | | Episodic | | | Raw | Raw | | | Raw | | |
| Model | Attention | **Kappa** | f1 | sens | fp/h | f1 | f1 | sens | spec | f1 | sens | spec |
| **MVPFormer** | MVPA | **0.61** | **0.59** | 0.72 | 0.15 | **0.51** | **0.36** | 0.38 | 0.91 | **0.46** | 0.94 | 0.10 |
| MVPFormer-S | MVPA | 0.57 | 0.53 | 0.71 | 0.12 | 0.49 | 0.35 | 0.41 | 0.88 | **0.46** | 0.99 | 0.03 |
| Brant-2 | Vanilla | 0.06 | 0.01 | 0.01 | 0.11 | 0.00 | 0.19 | 1.00 | 0.18 | **0.46** | 0.99 | 0.02 |
| BrainBERT | Vanilla | 0.00 | 0.00 | 0.00 | 0.00 | 0.00 | / | / | / | / | / | / |
| MV-Llama | Vanilla | 0.11 | 0.01 | 0.01 | 0.02 | 0.00 | / | / | / | / | / | / |

## G.2 BRAIN TREEBANK DECODING TASKS

Table 18 reports a summary of the decoding tasks of the Brain TreeBank dataset.

Table 18: **Results on Brain TreeBank iEEG tasks.** We compare MVPFormer with multiple Transformer-based architectures on the four tasks of the Brain TreeBank dataset (Wang et al., 2024a). The best results are bolded, while the results where the electrodess position is beneficial are underlined.

| Model | Attention | Electrode location | Pitch | Volume | Onset | Speech |
|---|---|---|---|---|---|---|
| **MVPFormer** | MVPA | No | **0.83** (0.02) | **0.88** (0.01) | **0.87** (0.02) | **0.90** (0.02) |
| MV-Llama | Vanilla | No | 0.62 (0.03) | 0.77 (0.02) | 0.80 (0.03) | 0.81 (0.02) |
| Brant | Vanilla | No | 0.61 (0.03) | 0.74 (0.03) | 0.80 (0.04) | 0.80 (0.03) |
| PopT w/o encoding | Vanilla | No | 0.62 (0.07) | 0.76 (0.07) | 0.81 (0.09) | 0.83 (0.10) |
| PopT (BrainBERT) | Vanilla | Yes | 0.74 (0.03) | 0.87 (0.03) | 0.90 (0.01) | 0.93 (0.02) |
| PopT (TOTEM) | Vanilla | Yes | 0.64 (0.03) | 0.79 (0.02) | 0.90 (0.02) | 0.88 (0.05) |

## G.3 CONVENTIONAL EVALUATION

In addition to our clinically motivated evaluation (see Section 5.2.1), we assess all our models using conventional machine learning metrics for seizure detection: F1-score, sensitivity, and false positive rate. These metrics are commonly used in benchmarking seizure detection models (Ziyabari et al., 2017; Shah et al., 2020), and allow comparison with prior work. The full seizure detection results of MVPFormer are shown in Table 17 and Figure 13.

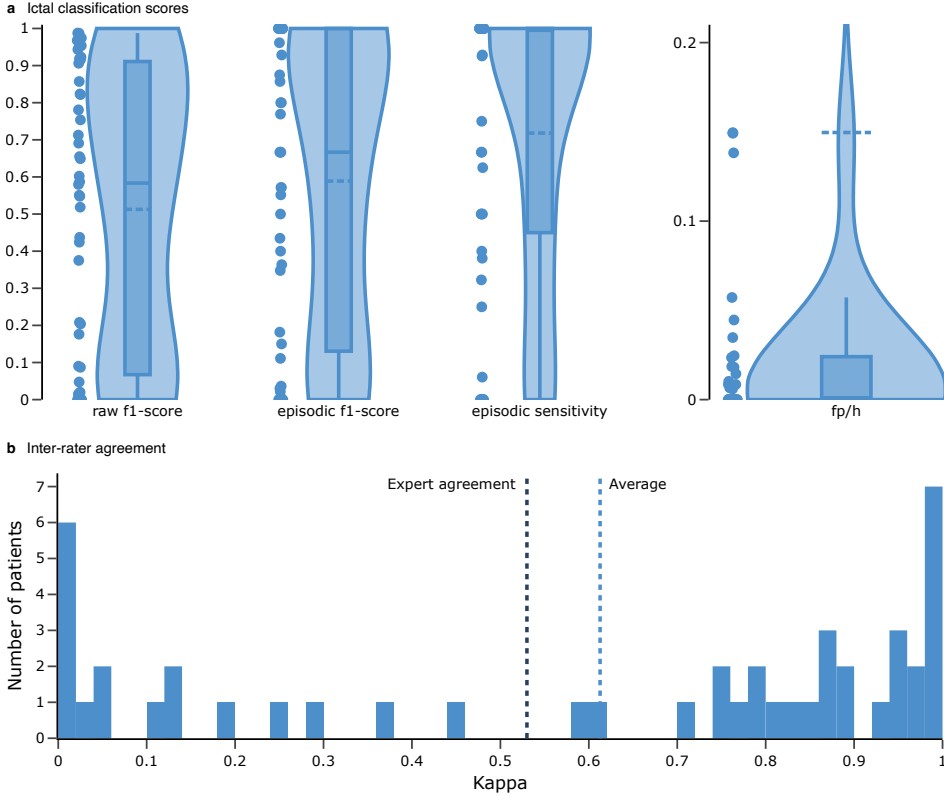

Figure 13: **Performance of MVPFormer on the classification task.** (a) The F1-score, sensitivity, and fp/h are reported. Raw results are computed without any post-processing of MVPFormer's output, while episodic results follow a common post-processing procedure which merge close ictal classifications. (b) Cohen's Kappa is used to measure the agreement between the artificial assistant and the human expert. The average kappa is 0.61, competitive with the values obtained between human experts. The distribution of kappa values clearly indicates that a minority of subjects are the source of most disagreement, consistent with the variability of inter-rater agreement among human experts.

We evaluate against two baselines: Brant-2 (Yuan et al., 2024a), a SOTA Transformer model for iEEG, and MV-Llama, an ablation of MVPFormer-S that uses standard attention instead of MVPA (see Appendix G.5). Brant-2 is fine-tuned with its published pre-trained weights and protocol. MV-Llama is trained identically to MVPFormer-S.

We report both raw and episodic metrics. Episodic metrics reflect clinically meaningful detections by grouping predictions into events (Ziyabari et al., 2017). The detailed results are provided in Tables 25 and 19.

The similarity between raw and episodic F1-scores suggests that MVPFormer naturally learns to detect seizure episodes of realistic length and frequency. On the 50-subject SWEC test set, the false positive rate is 0.15 fp/h (0.12 for MVPFormer-S), comparable to commercial EEG devices used in clinical practice (Van de Vel et al., 2014; Bruno et al., 2020). As expected in a real-world dataset, false positive rates vary considerably across subjects, with 81% having fewer than 0.05 fp/h.

Table 19: **Details of seizure detection results of MVPFormer with 18 subject pre-training.** Kappa is the inter-rater agreement. The classification metrics report the raw and episodic metrics relevant for the seizure classification task. The similarity reports the breakdown of the cosine similarity in each of the considered scenarios.

| | | | Classification metrics | | | | Similarity | | | | | | | | |
| | | | Raw | Episodic | | | True | | | Random | | | Two-step | | |
| Subject | Kappa | 95% CI | f1-score | f1-score | sensitivity | fp/h | average | ictal | non-ictal | average | ictal | non-ictal | average | ictal | non-ictal |
|---|---|---|---|---|---|---|---|---|---|---|---|---|---|---|---|
| ID19 | 0.00 | 0.00 | 0.01 | 0.35 | 0.25 | 0.14 | 0.30 | 0.27 | 0.25 | 0.14 | 0.15 | 0.10 | 0.16 | 0.14 | 0.10 |
| ID20 | 0.87 | 0.05 | 0.92 | 0.57 | 1.00 | 0.01 | 0.89 | 0.95 | 0.91 | 0.17 | 0.19 | 0.12 | 0.14 | 0.17 | 0.12 |
| ID21 | 0.82 | 0.09 | 0.59 | 0.67 | 1.00 | 0.01 | 0.92 | 0.92 | 0.93 | 0.16 | 0.15 | 0.16 | 0.13 | 0.22 | 0.18 |
| ID22 | 0.99 | 0.02 | 0.94 | 1.00 | 1.00 | 0.00 | 0.89 | 0.92 | 0.90 | 0.15 | 0.15 | 0.10 | 0.12 | 0.15 | 0.08 |
| ID23 | 0.12 | 0.03 | 0.21 | 0.11 | 0.06 | 0.02 | 0.88 | 0.95 | 0.94 | 0.22 | 0.22 | 0.20 | 0.20 | 0.21 | 0.20 |
| ID24 | 0.92 | 0.01 | 0.91 | 0.93 | 0.93 | 0.02 | 0.80 | 0.93 | 0.89 | 0.13 | 0.13 | 0.09 | 0.14 | 0.11 | 0.14 |
| ID25 | 0.00 | 0.00 | 0.00 | 0.00 | 0.00 | 0.00 | 0.84 | 0.90 | 0.90 | 0.17 | 0.24 | 0.14 | 0.15 | 0.20 | 0.13 |
| ID26 | 0.03 | 0.03 | 0.02 | 0.04 | 1.00 | 0.61 | 0.88 | 0.96 | 0.94 | 0.16 | 0.21 | 0.09 | 0.13 | 0.16 | 0.10 |
| ID27 | 0.99 | 0.01 | 0.95 | 1.00 | 1.00 | 0.00 | 0.88 | 0.95 | 0.95 | 0.20 | 0.23 | 0.20 | 0.17 | 0.23 | 0.21 |
| ID28 | 0.83 | 0.04 | 0.82 | 0.86 | 0.75 | 0.00 | 0.85 | 0.89 | 0.61 | 0.26 | 0.14 | 0.07 | 0.16 | 0.15 | 0.10 |
| ID29 | 0.71 | 0.02 | 0.58 | 0.43 | 0.32 | 0.03 | 0.86 | 0.92 | 0.92 | 0.14 | 0.14 | 0.11 | 0.11 | 0.12 | 0.13 |
| ID30 | 0.95 | 0.01 | 0.92 | 0.96 | 0.93 | 0.00 | 0.89 | 0.95 | 0.94 | 0.17 | 0.17 | 0.13 | 0.14 | 0.16 | 0.14 |
| ID31 | 0.99 | 0.00 | 0.97 | 1.00 | 1.00 | 0.00 | 0.88 | 0.95 | 0.95 | 0.18 | 0.21 | 0.13 | 0.15 | 0.22 | 0.17 |
| ID32 | 1.00 | 0.00 | 0.97 | 1.00 | 1.00 | 0.00 | 0.93 | 0.93 | 0.94 | 0.17 | 0.15 | 0.12 | 0.13 | 0.16 | 0.11 |
| ID33 | 0.06 | 0.07 | 0.00 | 0.00 | 0.00 | 0.15 | 0.95 | 0.96 | 0.95 | 0.39 | 0.38 | 0.31 | 0.39 | 0.45 | 0.46 |
| ID34 | 0.99 | 0.01 | 0.97 | 1.00 | 1.00 | 0.00 | 0.89 | 0.94 | 0.92 | 0.14 | 0.14 | 0.09 | 0.11 | 0.14 | 0.12 |
| ID35 | 0.98 | 0.01 | 0.96 | 1.00 | 1.00 | 0.00 | 0.86 | 0.93 | 0.94 | 0.18 | 0.15 | 0.25 | 0.17 | 0.17 | 0.35 |
| ID36 | 0.60 | 0.03 | 0.78 | 0.57 | 0.40 | 0.00 | 0.91 | 0.95 | 0.94 | 0.19 | 0.20 | 0.13 | 0.16 | 0.18 | 0.14 |
| ID37 | 1.00 | 0.00 | 0.97 | 1.00 | 1.00 | 0.00 | 0.77 | 0.74 | 0.66 | 0.16 | 0.12 | 0.29 | 0.14 | 0.15 | 0.49 |
| ID38 | 0.99 | 0.01 | 0.99 | 1.00 | 1.00 | 0.00 | 0.82 | 0.85 | 0.77 | 0.15 | 0.13 | 0.13 | 0.13 | 0.14 | 0.09 |
| ID39 | 0.99 | 0.02 | 0.92 | 1.00 | 1.00 | 0.00 | 0.83 | 0.70 | 0.83 | 0.16 | 0.12 | 0.10 | 0.14 | 0.12 | 0.10 |
| ID40 | 1.00 | 0.01 | 0.99 | 1.00 | 1.00 | 0.00 | 0.87 | 0.88 | 0.90 | 0.15 | 0.13 | 0.09 | 0.13 | 0.13 | 0.11 |
| ID41 | 0.13 | 0.03 | 0.18 | 0.15 | 1.00 | 0.41 | 0.86 | 0.79 | 0.86 | 0.16 | 0.15 | 0.10 | 0.13 | 0.15 | 0.10 |
| ID42 | 0.96 | 0.03 | 0.86 | 1.00 | 1.00 | 0.00 | 0.82 | 0.80 | 0.79 | 0.15 | 0.14 | 0.07 | 0.13 | 0.15 | 0.07 |
| ID43 | 0.89 | 0.04 | 0.69 | 1.00 | 1.00 | 0.00 | 0.80 | 0.77 | 0.84 | 0.13 | 0.14 | 0.12 | 0.10 | 0.15 | 0.11 |
| ID44 | 0.90 | 0.12 | 0.42 | 0.67 | 0.50 | 0.00 | 0.82 | 0.89 | 0.89 | 0.11 | 0.14 | 0.49 | 0.08 | 0.09 | 0.65 |
| ID45 | 0.37 | 0.09 | 0.09 | 0.18 | 1.00 | 0.06 | 0.82 | 0.87 | 0.77 | 0.14 | 0.16 | 0.09 | 0.12 | 0.14 | 0.16 |
| ID46 | 0.45 | 0.04 | 0.44 | 0.55 | 0.38 | 0.00 | 0.85 | 0.87 | 0.61 | 0.18 | 0.14 | 0.28 | 0.16 | 0.11 | 0.51 |
| ID47 | 0.77 | 0.09 | 0.60 | 0.50 | 1.00 | 0.02 | 0.84 | 0.81 | 0.83 | 0.16 | 0.15 | 0.12 | 0.14 | 0.16 | 0.09 |
| ID48 | 0.87 | 0.01 | 0.71 | 1.00 | 1.00 | 0.00 | 0.80 | 0.83 | 0.79 | 0.12 | 0.12 | 0.08 | 0.10 | 0.11 | 0.09 |
| ID49 | 0.25 | 0.04 | 0.20 | 0.36 | 1.00 | 0.15 | 0.89 | 0.79 | 0.86 | 0.23 | 0.14 | 0.35 | 0.23 | 0.17 | 0.53 |
| ID50 | 0.97 | 0.05 | 0.82 | 1.00 | 1.00 | 0.00 | 0.86 | 0.90 | 0.84 | 0.16 | 0.24 | 0.11 | 0.14 | 0.21 | 0.10 |
| ID51 | 0.75 | 0.11 | 0.38 | 0.40 | 1.00 | 0.02 | 0.87 | 0.78 | 0.89 | 0.26 | 0.14 | 0.28 | 0.28 | 0.24 | 0.39 |
| ID52 | 0.85 | 0.05 | 0.65 | 1.00 | 1.00 | 0.00 | 0.82 | 0.84 | 0.79 | 0.21 | 0.18 | 0.11 | 0.19 | 0.16 | 0.14 |
| ID53 | 0.00 | 0.00 | 0.00 | 0.00 | 1.00 | 3.43 | 0.82 | 0.77 | 0.79 | 0.17 | 0.17 | 0.09 | 0.14 | 0.15 | 0.11 |
| ID54 | 0.01 | 0.02 | 0.02 | 0.02 | 0.67 | 0.89 | 0.82 | 0.71 | 0.83 | 0.16 | 0.14 | 0.12 | 0.13 | 0.14 | 0.09 |
| ID55 | 0.86 | 0.07 | 0.75 | 0.80 | 1.00 | 0.01 | 0.84 | 0.87 | 0.80 | 0.18 | 0.18 | 0.11 | 0.16 | 0.19 | 0.14 |
| ID56 | 0.79 | 0.06 | 0.55 | 0.80 | 0.67 | 0.00 | 0.87 | 0.81 | 0.79 | 0.19 | 0.18 | 0.12 | 0.17 | 0.20 | 0.18 |
| ID57 | 0.00 | 0.00 | 0.00 | 0.00 | 0.00 | 0.00 | 0.87 | 0.87 | 0.81 | 0.16 | 0.21 | 0.14 | 0.13 | 0.17 | 0.17 |
| ID58 | 0.30 | 0.11 | 0.00 | 0.00 | 0.00 | 0.01 | 0.86 | 0.84 | 0.77 | 0.21 | 0.22 | 0.33 | 0.20 | 0.24 | 0.30 |
| ID59 | 0.00 | 0.00 | 0.00 | 0.00 | 0.00 | 0.00 | 0.82 | 0.85 | 0.80 | 0.17 | 0.14 | 0.09 | 0.16 | 0.12 | 0.11 |
| ID60 | 0.20 | 0.03 | 0.09 | 0.67 | 0.50 | 0.00 | 0.87 | 0.88 | 0.93 | 0.15 | 0.14 | 0.56 | 0.12 | 0.11 | 0.37 |
| ID61 | 0.14 | 0.09 | 0.00 | 0.00 | 0.00 | 0.04 | 0.89 | 0.87 | 0.89 | 0.18 | 0.16 | 0.11 | 0.15 | 0.13 | 0.11 |
| ID62 | 0.79 | 0.04 | 0.65 | 0.88 | 1.00 | 0.01 | 0.83 | 0.86 | 0.92 | 0.14 | 0.13 | 0.52 | 0.13 | 0.12 | 0.72 |
| ID63 | 0.76 | 0.05 | 0.52 | 0.77 | 0.62 | 0.00 | 0.82 | 0.86 | 0.86 | 0.15 | 0.16 | 0.09 | 0.13 | 0.15 | 0.06 |
| ID64 | 0.95 | 0.07 | 0.55 | 1.00 | 1.00 | 0.00 | 0.86 | 0.87 | 0.85 | 0.15 | 0.15 | 0.16 | 0.12 | 0.14 | 0.10 |
| ID65 | 0.04 | 0.02 | 0.05 | 0.03 | 0.50 | 1.14 | 0.83 | 0.87 | 0.85 | 0.13 | 0.15 | 0.09 | 0.10 | 0.13 | 0.08 |
| ID66 | 0.61 | 0.19 | 0.00 | 0.00 | 0.00 | 0.01 | 0.75 | 0.80 | 0.76 | 0.12 | 0.13 | 0.06 | 0.10 | 0.11 | 0.08 |
| ID67 | 0.04 | 0.01 | 0.05 | 0.03 | 0.50 | 1.06 | 0.83 | 0.82 | 0.81 | 0.20 | 0.15 | 0.07 | 0.19 | 0.12 | 0.10 |
| ID68 | 0.66 | 0.18 | 0.00 | 0.00 | 0.00 | 0.01 | 0.84 | 0.80 | 0.82 | 0.17 | 0.22 | 0.15 | 0.14 | 0.19 | 0.16 |

These results confirm that MVPFormer performs competitively on conventional seizure detection benchmarks, while also offering robust generalization to clinically realistic evaluation settings.

### G.4 EFFECTS OF THE SCALE OF THE MODEL

The performance improvements of LLMs as a function of their model sizes have also been widely reported (Hoffmann et al., 2022; Kaplan et al., 2020). According to Chinchilla's scaling law the training dataset is already not large enough to fully train MVPFormer-S (75M parameters), so we investigate whether a larger model (MVPFormer-M, 1.2B parameters) can provide any improvement in performance.

Figure 14 shows the seizure detection performance of MVPFormer-S on the SWEC iEEG dataset (see Table 20). As noted in the main results, MVPFormer-M marginally improves seizure detection results over MVPFormer-S. In particular, it reaches higher F1-score but higher fp/h rate as well, with small net improvement. Therefore, we have shown that the amount of iEEG data currently available is not sufficient to fully take advantage of the increase in model size of Transformers. We hope that making the SWEC iEEG dataset publicly available will increase overall availability and unlock further model scaling potential.

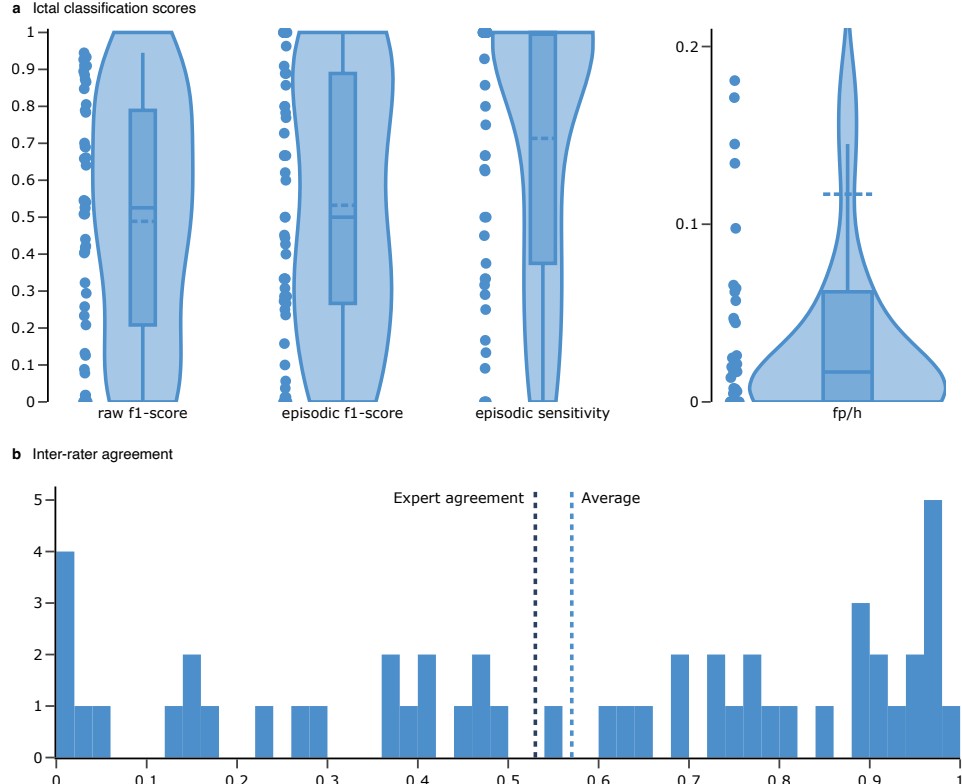

Figure 14: **Seizure detection with 18 patient pre-training.** **(a)** Seizure detection results of MVPFormer-S on unseen subjects: the F1-score, sensitivity, and fp/h are reported. Raw results are computed without any post-processing of MVPFormer's output, while episodic results follow a common post-processing procedure which merge close ictal classifications. **(b)** Cohen's Kappa is used to measure the agreement between MVPFormer-S and the human expert. The average kappa is 0.57, competitive with the values obtained between human experts. The distribution of kappa values clearly indicates that a minority of subjects are the source of most disagreement, consistent with the variability of inter-rater agreement among human experts.

Table 20: **Details of seizure detection results of MVPFormer-S.** Kappa is the inter-rater agreement. The classification metrics report the raw and episodic metrics relevant for the seizure classification task. The similarity reports the breakdown of the cosine similarity in each of the considered scenarios.

| | | | Classification metrics | | | | Similarity | | | | | | | | |
| | | | Raw | Episodic | | | True | | | Random | | | Two-step | | |
| Subject | Kappa | 95% CI | f1-score | f1-score | sensitivity | fp/h | average | ictal | non-ictal | average | ictal | non-ictal | average | ictal | non-ictal |
|---|---|---|---|---|---|---|---|---|---|---|---|---|---|---|---|
| ID19 | 0.01 | 0.00 | 0.02 | 0.33 | 0.50 | 1.11 | 0.28 | 0.27 | 0.27 | 0.21 | 0.20 | 0.25 | 0.21 | 0.20 | 0.23 |
| ID20 | 0.79 | 0.06 | 0.87 | 0.44 | 1.00 | 0.02 | 0.95 | 0.96 | 0.95 | 0.18 | 0.21 | 0.12 | 0.15 | 0.17 | 0.12 |
| ID21 | 0.73 | 0.11 | 0.51 | 0.50 | 1.00 | 0.02 | 0.94 | 0.93 | 0.95 | 0.16 | 0.18 | 0.18 | 0.13 | 0.24 | 0.20 |
| ID22 | 0.97 | 0.02 | 0.91 | 1.00 | 1.00 | 0.00 | 0.94 | 0.89 | 0.95 | 0.15 | 0.16 | 0.09 | 0.12 | 0.17 | 0.07 |
| ID23 | 0.15 | 0.03 | 0.21 | 0.16 | 0.09 | 0.05 | 0.95 | 0.95 | 0.95 | 0.25 | 0.28 | 0.23 | 0.22 | 0.24 | 0.24 |
| ID24 | 0.94 | 0.01 | 0.91 | 0.96 | 0.93 | 0.00 | 0.93 | 0.94 | 0.93 | 0.19 | 0.19 | 0.12 | 0.16 | 0.17 | 0.17 |
| ID25 | 0.13 | 0.09 | 0.00 | 0.00 | 0.00 | 0.06 | 0.94 | 0.96 | 0.96 | 0.17 | 0.19 | 0.15 | 0.14 | 0.18 | 0.15 |
| ID26 | 0.04 | 0.03 | 0.02 | 0.04 | 1.00 | 0.59 | 0.95 | 0.97 | 0.96 | 0.17 | 0.22 | 0.12 | 0.14 | 0.18 | 0.11 |
| ID27 | 0.88 | 0.03 | 0.78 | 0.89 | 1.00 | 0.01 | 0.95 | 0.96 | 0.96 | 0.22 | 0.24 | 0.21 | 0.19 | 0.25 | 0.24 |
| ID28 | 0.89 | 0.03 | 0.87 | 0.86 | 0.75 | 0.00 | 0.95 | 0.96 | 0.95 | 0.17 | 0.19 | 0.09 | 0.13 | 0.17 | 0.12 |
| ID29 | 0.45 | 0.03 | 0.42 | 0.27 | 0.29 | 0.18 | 0.95 | 0.95 | 0.95 | 0.16 | 0.15 | 0.14 | 0.12 | 0.13 | 0.13 |
| ID30 | 0.63 | 0.02 | 0.66 | 0.67 | 0.63 | 0.17 | 0.95 | 0.96 | 0.96 | 0.19 | 0.19 | 0.15 | 0.16 | 0.18 | 0.16 |
| ID31 | 0.98 | 0.00 | 0.93 | 1.00 | 1.00 | 0.00 | 0.95 | 0.96 | 0.96 | 0.19 | 0.24 | 0.11 | 0.16 | 0.24 | 0.18 |
| ID32 | 0.99 | 0.03 | 0.93 | 1.00 | 1.00 | 0.00 | 0.95 | 0.91 | 0.96 | 0.18 | 0.13 | 0.16 | 0.15 | 0.18 | 0.12 |
| ID33 | 0.48 | 0.14 | 0.00 | 0.00 | 0.00 | 0.02 | 0.93 | 0.95 | 0.93 | 0.23 | 0.28 | 0.19 | 0.21 | 0.26 | 0.21 |
| ID34 | 0.78 | 0.03 | 0.85 | 0.78 | 1.00 | 0.03 | 0.95 | 0.95 | 0.95 | 0.15 | 0.15 | 0.10 | 0.11 | 0.15 | 0.13 |
| ID35 | 0.77 | 0.03 | 0.79 | 0.80 | 0.86 | 0.02 | 0.92 | 0.94 | 0.92 | 0.17 | 0.19 | 0.19 | 0.15 | 0.17 | 0.22 |
| ID36 | 0.41 | 0.03 | 0.54 | 0.43 | 0.32 | 0.06 | 0.96 | 0.96 | 0.95 | 0.22 | 0.23 | 0.14 | 0.19 | 0.21 | 0.17 |
| ID37 | 0.50 | 0.09 | 0.44 | 0.31 | 1.00 | 0.05 | 0.92 | 0.91 | 0.91 | 0.16 | 0.15 | 0.33 | 0.14 | 0.17 | 0.69 |
| ID38 | 0.94 | 0.02 | 0.94 | 0.91 | 1.00 | 0.01 | 0.93 | 0.95 | 0.91 | 0.14 | 0.15 | 0.11 | 0.11 | 0.16 | 0.08 |
| ID39 | 0.97 | 0.03 | 0.89 | 1.00 | 1.00 | 0.00 | 0.93 | 0.90 | 0.92 | 0.16 | 0.13 | 0.11 | 0.14 | 0.13 | 0.11 |
| ID40 | 0.91 | 0.04 | 0.91 | 0.89 | 0.80 | 0.00 | 0.94 | 0.95 | 0.95 | 0.15 | 0.16 | 0.10 | 0.13 | 0.14 | 0.12 |
| ID41 | 0.28 | 0.05 | 0.41 | 0.33 | 1.00 | 0.15 | 0.95 | 0.91 | 0.94 | 0.16 | 0.16 | 0.14 | 0.13 | 0.16 | 0.11 |
| ID42 | 0.98 | 0.03 | 0.89 | 1.00 | 1.00 | 0.00 | 0.93 | 0.91 | 0.92 | 0.15 | 0.17 | 0.08 | 0.12 | 0.16 | 0.08 |
| ID43 | 0.92 | 0.03 | 0.69 | 1.00 | 1.00 | 0.00 | 0.93 | 0.90 | 0.95 | 0.14 | 0.14 | 0.10 | 0.11 | 0.16 | 0.12 |
| ID44 | 0.90 | 0.12 | 0.32 | 0.67 | 0.50 | 0.00 | 0.93 | 0.95 | 0.92 | 0.12 | 0.12 | 0.51 | 0.09 | 0.10 | 0.64 |
| ID45 | 0.01 | 0.02 | 0.00 | 0.01 | 1.00 | 0.99 | 0.94 | 0.95 | 0.93 | 0.15 | 0.18 | 0.13 | 0.12 | 0.14 | 0.14 |
| ID46 | 0.39 | 0.04 | 0.42 | 0.45 | 0.33 | 0.02 | 0.94 | 0.95 | 0.76 | 0.18 | 0.15 | 0.36 | 0.16 | 0.12 | 0.53 |
| ID47 | 0.06 | 0.05 | 0.09 | 0.06 | 1.00 | 0.30 | 0.94 | 0.93 | 0.94 | 0.18 | 0.20 | 0.12 | 0.15 | 0.18 | 0.12 |
| ID48 | 0.85 | 0.01 | 0.70 | 1.00 | 1.00 | 0.00 | 0.93 | 0.93 | 0.92 | 0.12 | 0.11 | 0.06 | 0.10 | 0.12 | 0.09 |
| ID49 | 0.54 | 0.05 | 0.54 | 0.60 | 1.00 | 0.06 | 0.95 | 0.92 | 0.93 | 0.22 | 0.14 | 0.26 | 0.21 | 0.19 | 0.51 |
| ID50 | 0.97 | 0.05 | 0.81 | 1.00 | 1.00 | 0.00 | 0.93 | 0.95 | 0.94 | 0.17 | 0.19 | 0.12 | 0.14 | 0.23 | 0.11 |
| ID51 | 0.70 | 0.12 | 0.55 | 0.33 | 1.00 | 0.02 | 0.93 | 0.91 | 0.93 | 0.23 | 0.15 | 0.20 | 0.23 | 0.28 | 0.26 |
| ID52 | 0.37 | 0.05 | 0.54 | 0.27 | 1.00 | 0.10 | 0.93 | 0.94 | 0.95 | 0.16 | 0.18 | 0.10 | 0.13 | 0.16 | 0.10 |
| ID53 | 0.47 | 0.08 | 0.40 | 0.25 | 1.00 | 0.04 | 0.93 | 0.90 | 0.93 | 0.19 | 0.17 | 0.13 | 0.15 | 0.16 | 0.12 |
| ID54 | 0.01 | 0.01 | 0.01 | 0.01 | 0.67 | 1.51 | 0.94 | 0.88 | 0.94 | 0.18 | 0.19 | 0.14 | 0.14 | 0.16 | 0.10 |
| ID55 | 0.41 | 0.09 | 0.29 | 0.29 | 1.00 | 0.07 | 0.94 | 0.94 | 0.94 | 0.20 | 0.18 | 0.14 | 0.17 | 0.20 | 0.17 |
| ID56 | 0.72 | 0.08 | 0.52 | 0.67 | 0.67 | 0.01 | 0.94 | 0.91 | 0.88 | 0.19 | 0.18 | 0.13 | 0.17 | 0.19 | 0.16 |
| ID57 | 0.00 | 0.00 | 0.00 | 0.00 | 0.00 | 0.00 | 0.94 | 0.94 | 0.89 | 0.14 | 0.13 | 0.14 | 0.11 | 0.09 | 0.16 |
| ID58 | 0.29 | 0.11 | 0.00 | 0.00 | 0.00 | 0.01 | 0.94 | 0.94 | 0.86 | 0.21 | 0.22 | 0.18 | 0.20 | 0.24 | 0.15 |
| ID59 | 0.16 | 0.06 | 0.13 | 0.24 | 0.13 | 0.00 | 0.92 | 0.94 | 0.91 | 0.14 | 0.14 | 0.11 | 0.12 | 0.12 | 0.09 |
| ID60 | 0.23 | 0.03 | 0.13 | 0.50 | 0.38 | 0.02 | 0.95 | 0.96 | 0.98 | 0.14 | 0.14 | 0.61 | 0.11 | 0.12 | 0.37 |
| ID61 | 0.36 | 0.09 | 0.08 | 0.29 | 0.17 | 0.00 | 0.96 | 0.96 | 0.96 | 0.19 | 0.17 | 0.16 | 0.16 | 0.16 | 0.14 |
| ID62 | 0.81 | 0.05 | 0.69 | 1.00 | 1.00 | 0.00 | 0.92 | 0.93 | 0.97 | 0.14 | 0.14 | 0.43 | 0.11 | 0.11 | 0.73 |
| ID63 | 0.64 | 0.05 | 0.64 | 0.73 | 1.00 | 0.02 | 0.93 | 0.94 | 0.95 | 0.15 | 0.17 | 0.09 | 0.12 | 0.16 | 0.07 |
| ID64 | 0.60 | 0.02 | 0.66 | 0.62 | 0.45 | 0.00 | 0.95 | 0.95 | 0.95 | 0.16 | 0.17 | 0.13 | 0.13 | 0.14 | 0.11 |
| ID65 | 0.75 | 0.05 | 0.51 | 0.77 | 0.62 | 0.00 | 0.94 | 0.95 | 0.94 | 0.14 | 0.17 | 0.08 | 0.11 | 0.14 | 0.09 |
| ID66 | 0.90 | 0.10 | 0.53 | 0.80 | 1.00 | 0.00 | 0.87 | 0.92 | 0.92 | 0.13 | 0.13 | 0.08 | 0.10 | 0.13 | 0.08 |
| ID67 | 0.15 | 0.05 | 0.26 | 0.10 | 0.25 | 0.13 | 0.94 | 0.94 | 0.94 | 0.22 | 0.17 | 0.11 | 0.21 | 0.13 | 0.10 |
| ID68 | 0.69 | 0.13 | 0.23 | 0.40 | 0.33 | 0.01 | 0.94 | 0.92 | 0.95 | 0.15 | 0.13 | 0.31 | 0.10 | 0.10 | 0.10 |

## G.5 EFFECTS OF THE ATTENTION MECHANISM

To assess the validity of our MVPA scheme, we train MV-Llama, a model almost equivalent to MVPFormer-S that uses vanilla attention instead of MVPA. While MV-Llama uses vanilla attention, it is still based on the SOTA Llama2 architecture. We also re-use the vanilla positional encoding, with a simple adjustment to recover a one-to-one correspondence between the positional encoding and the position of the patch in the time-series (the Cantor pairing function, see App. A.1).

Table 17 in the main text indicates that vanilla attention does not perform seizure detection at a level comparable to MVPA. In particular, the performance of MV-Llama is poor, indicating that it cannot generalize to this task. We argue this is due to higher flexibility of the internal representations generated by MVPA, which better lend themselves to further tasks, such as seizure classification.

Table 21: **Details of seizure detection results of MV-Llama with 18 subject pre-training.** Kappa is the inter-rater agreement. The classification metrics report the raw and episodic metrics relevant for the seizure classification task. The similarity reports the breakdown of the cosine similarity in each of the considered scenarios.

| | | | Classification metrics | | | | Similarity | | | | | | | | |
| | | | Raw | Episodic | | | True | | | Random | | | Two-step | | |
| Subject | Kappa | 95% CI | f1-score | f1-score | sensitivity | fp/h | average | ictal | non-ictal | average | ictal | non-ictal | average | ictal | non-ictal |
|---|---|---|---|---|---|---|---|---|---|---|---|---|---|---|---|
| ID19 | 0.00 | 0.00 | 0.01 | 0.12 | 0.06 | 0.00 | 0.32 | 0.31 | 0.29 | 0.23 | 0.23 | 0.23 | 0.23 | 0.22 | 0.22 |
| ID20 | 0.00 | 0.00 | 0.00 | 0.00 | 0.00 | 0.00 | 0.96 | 0.97 | 0.95 | 0.18 | 0.21 | 0.14 | 0.15 | 0.18 | 0.13 |
| ID21 | 0.00 | 0.00 | 0.00 | 0.00 | 0.00 | 0.00 | 0.95 | 0.90 | 0.96 | 0.16 | 0.15 | 0.15 | 0.13 | 0.23 | 0.20 |
| ID22 | 0.34 | 0.12 | 0.00 | 0.00 | 0.00 | 0.03 | 0.95 | 0.78 | 0.95 | 0.15 | 0.13 | 0.08 | 0.12 | 0.16 | 0.07 |
| ID23 | 0.00 | 0.00 | 0.00 | 0.00 | 0.00 | 0.00 | 0.96 | 0.96 | 0.96 | 0.25 | 0.26 | 0.20 | 0.22 | 0.24 | 0.24 |
| ID24 | 0.00 | 0.00 | 0.00 | 0.00 | 0.00 | 0.00 | 0.94 | 0.94 | 0.93 | 0.19 | 0.20 | 0.19 | 0.16 | 0.18 | 0.17 |
| ID25 | 0.02 | 0.06 | 0.00 | 0.00 | 0.00 | 0.17 | 0.95 | 0.96 | 0.96 | 0.17 | 0.19 | 0.16 | 0.14 | 0.18 | 0.15 |
| ID26 | 0.00 | 0.00 | 0.00 | 0.00 | 0.00 | 0.00 | 0.96 | 0.97 | 0.96 | 0.18 | 0.22 | 0.10 | 0.14 | 0.18 | 0.12 |
| ID27 | 0.27 | 0.11 | 0.00 | 0.00 | 0.00 | 0.01 | 0.96 | 0.96 | 0.96 | 0.22 | 0.25 | 0.19 | 0.19 | 0.25 | 0.24 |
| ID28 | 0.00 | 0.00 | 0.00 | 0.00 | 0.00 | 0.00 | 0.96 | 0.96 | 0.96 | 0.17 | 0.19 | 0.08 | 0.13 | 0.17 | 0.13 |
| ID29 | 0.00 | 0.00 | 0.00 | 0.00 | 0.00 | 0.00 | 0.95 | 0.95 | 0.96 | 0.16 | 0.15 | 0.09 | 0.12 | 0.13 | 0.13 |
| ID30 | 0.01 | 0.06 | 0.00 | 0.00 | 0.00 | 0.22 | 0.96 | 0.96 | 0.96 | 0.19 | 0.20 | 0.15 | 0.16 | 0.19 | 0.17 |
| ID31 | 0.00 | 0.00 | 0.00 | 0.00 | 0.00 | 0.00 | 0.96 | 0.96 | 0.96 | 0.20 | 0.24 | 0.16 | 0.16 | 0.24 | 0.18 |
| ID32 | 0.12 | 0.09 | 0.00 | 0.00 | 0.00 | 0.05 | 0.95 | 0.88 | 0.96 | 0.18 | 0.16 | 0.16 | 0.15 | 0.18 | 0.12 |
| ID33 | 0.00 | 0.00 | 0.00 | 0.00 | 0.00 | 0.00 | 0.93 | 0.95 | 0.92 | 0.23 | 0.24 | 0.19 | 0.21 | 0.25 | 0.20 |
| ID34 | 0.00 | 0.00 | 0.00 | 0.00 | 0.00 | 0.00 | 0.95 | 0.95 | 0.96 | 0.15 | 0.16 | 0.12 | 0.11 | 0.15 | 0.12 |
| ID35 | 0.38 | 0.12 | 0.00 | 0.00 | 0.00 | 0.01 | 0.93 | 0.94 | 0.92 | 0.17 | 0.18 | 0.16 | 0.14 | 0.17 | 0.18 |
| ID36 | 0.17 | 0.09 | 0.00 | 0.00 | 0.00 | 0.08 | 0.96 | 0.96 | 0.96 | 0.22 | 0.24 | 0.16 | 0.19 | 0.21 | 0.17 |
| ID37 | 0.00 | 0.00 | 0.00 | 0.00 | 0.00 | 0.00 | 0.92 | 0.88 | 0.87 | 0.16 | 0.15 | 0.36 | 0.14 | 0.16 | 0.57 |
| ID38 | 0.00 | 0.00 | 0.00 | 0.00 | 0.00 | 0.00 | 0.93 | 0.95 | 0.91 | 0.14 | 0.16 | 0.10 | 0.11 | 0.16 | 0.07 |
| ID39 | 0.00 | 0.00 | 0.00 | 0.00 | 0.00 | 0.00 | 0.93 | 0.87 | 0.94 | 0.16 | 0.14 | 0.08 | 0.13 | 0.13 | 0.11 |
| ID40 | 0.02 | 0.06 | 0.00 | 0.00 | 0.00 | 0.04 | 0.94 | 0.95 | 0.95 | 0.15 | 0.17 | 0.10 | 0.12 | 0.15 | 0.12 |
| ID41 | 0.72 | 0.22 | 0.00 | 0.00 | 0.00 | 0.00 | 0.96 | 0.92 | 0.95 | 0.16 | 0.15 | 0.13 | 0.13 | 0.15 | 0.11 |
| ID42 | 0.00 | 0.00 | 0.00 | 0.00 | 0.00 | 0.00 | 0.94 | 0.92 | 0.94 | 0.15 | 0.18 | 0.09 | 0.12 | 0.17 | 0.08 |
| ID43 | 0.31 | 0.11 | 0.00 | 0.00 | 0.00 | 0.01 | 0.94 | 0.91 | 0.96 | 0.15 | 0.11 | 0.12 | 0.11 | 0.17 | 0.12 |
| ID44 | 0.00 | 0.00 | 0.00 | 0.00 | 0.00 | 0.00 | 0.94 | 0.96 | 0.86 | 0.12 | 0.12 | 0.37 | 0.09 | 0.10 | 0.50 |
| ID45 | 0.42 | 0.14 | 0.00 | 0.00 | 0.00 | 0.04 | 0.94 | 0.95 | 0.93 | 0.15 | 0.15 | 0.12 | 0.12 | 0.14 | 0.15 |
| ID46 | 0.00 | 0.00 | 0.00 | 0.00 | 0.00 | 0.00 | 0.94 | 0.96 | 0.80 | 0.18 | 0.14 | 0.28 | 0.16 | 0.12 | 0.53 |
| ID47 | 0.29 | 0.12 | 0.00 | 0.00 | 0.00 | 0.04 | 0.95 | 0.94 | 0.95 | 0.18 | 0.20 | 0.14 | 0.15 | 0.18 | 0.13 |
| ID48 | 0.00 | 0.00 | 0.00 | 0.00 | 0.00 | 0.00 | 0.93 | 0.94 | 0.93 | 0.12 | 0.12 | 0.07 | 0.09 | 0.12 | 0.09 |
| ID49 | 0.00 | 0.03 | 0.00 | 0.00 | 0.00 | 0.01 | 0.95 | 0.93 | 0.92 | 0.21 | 0.14 | 0.29 | 0.20 | 0.19 | 0.46 |
| ID50 | 0.06 | 0.07 | 0.00 | 0.00 | 0.00 | 0.01 | 0.94 | 0.96 | 0.94 | 0.17 | 0.21 | 0.12 | 0.14 | 0.23 | 0.11 |
| ID51 | 0.00 | 0.00 | 0.00 | 0.00 | 0.00 | 0.00 | 0.93 | 0.90 | 0.92 | 0.22 | 0.15 | 0.18 | 0.21 | 0.28 | 0.21 |
| ID52 | 0.00 | 0.00 | 0.00 | 0.00 | 0.00 | 0.00 | 0.94 | 0.95 | 0.95 | 0.16 | 0.17 | 0.12 | 0.13 | 0.17 | 0.10 |
| ID53 | 0.00 | 0.00 | 0.00 | 0.00 | 0.00 | 0.00 | 0.95 | 0.93 | 0.94 | 0.19 | 0.20 | 0.11 | 0.16 | 0.17 | 0.12 |
| ID54 | 0.52 | 0.10 | 0.17 | 0.33 | 0.25 | 0.01 | 0.94 | 0.89 | 0.95 | 0.18 | 0.16 | 0.12 | 0.15 | 0.16 | 0.10 |
| ID55 | 0.00 | 0.00 | 0.00 | 0.00 | 0.00 | 0.00 | 0.95 | 0.95 | 0.95 | 0.20 | 0.21 | 0.13 | 0.17 | 0.20 | 0.17 |
| ID56 | 0.68 | 0.21 | 0.00 | 0.00 | 0.00 | 0.00 | 0.94 | 0.91 | 0.89 | 0.19 | 0.18 | 0.15 | 0.16 | 0.18 | 0.15 |
| ID57 | 0.09 | 0.08 | 0.00 | 0.00 | 0.00 | 0.04 | 0.95 | 0.95 | 0.90 | 0.14 | 0.13 | 0.10 | 0.10 | 0.09 | 0.14 |
| ID58 | 0.00 | 0.00 | 0.00 | 0.00 | 0.00 | 0.00 | 0.94 | 0.94 | 0.88 | 0.21 | 0.22 | 0.15 | 0.19 | 0.23 | 0.17 |
| ID59 | 0.16 | 0.06 | 0.13 | 0.24 | 0.13 | 0.00 | 0.93 | 0.94 | 0.93 | 0.14 | 0.15 | 0.06 | 0.11 | 0.12 | 0.10 |
| ID60 | 0.23 | 0.03 | 0.13 | 0.50 | 0.38 | 0.02 | 0.95 | 0.96 | 0.98 | 0.14 | 0.15 | 0.62 | 0.11 | 0.12 | 0.41 |
| ID61 | 0.36 | 0.09 | 0.08 | 0.29 | 0.17 | 0.00 | 0.96 | 0.96 | 0.97 | 0.19 | 0.20 | 0.18 | 0.16 | 0.16 | 0.15 |
| —ID62 | 0.81 | 0.05 | 0.69 | 1.00 | 1.00 | 0.00 | 0.92 | 0.93 | 0.97 | 0.13 | 0.13 | 0.52 | 0.10 | 0.10 | 0.74 |
| ID63 | 0.64 | 0.05 | 0.64 | 0.73 | 1.00 | 0.02 | 0.94 | 0.95 | 0.95 | 0.16 | 0.18 | 0.11 | 0.12 | 0.16 | 0.07 |
| ID64 | 0.60 | 0.02 | 0.66 | 0.62 | 0.45 | 0.00 | 0.95 | 0.95 | 0.96 | 0.16 | 0.18 | 0.12 | 0.13 | 0.15 | 0.11 |
| ID65 | 0.75 | 0.05 | 0.51 | 0.77 | 0.62 | 0.00 | 0.95 | 0.95 | 0.95 | 0.14 | 0.15 | 0.11 | 0.11 | 0.14 | 0.09 |
| ID66 | 0.90 | 0.10 | 0.53 | 0.80 | 1.00 | 0.00 | 0.88 | 0.93 | 0.92 | 0.12 | 0.14 | 0.08 | 0.09 | 0.13 | 0.08 |
| ID67 | 0.15 | 0.05 | 0.26 | 0.10 | 0.25 | 0.13 | 0.95 | 0.95 | 0.94 | 0.22 | 0.17 | 0.09 | 0.20 | 0.14 | 0.10 |
| ID68 | 0.69 | 0.13 | 0.23 | 0.40 | 0.33 | 0.01 | 0.86 | 0.91 | 0.89 | 0.16 | 0.17 | 0.10 | 0.20 | 0.16 | 0.08 |

## G.6 SEIZURE DETECTION WITH BRANT-2

Table 22 presents the detailed subject-by-subject breakdown of the performance of Brant-2.

Table 22: **Details of seizure detection results of Brant-2 with 18 subject pre-training.** Kappa is the inter-rater agreement. The classification metrics report the raw and episodic metrics relevant for the seizure classification task.

| | | | Classification metrics | | | |
| | | | Raw | Episodic | | |
| Subject | Kappa | 95% CI | f1-score | f1-score | sensitivity | fp/h |
|---|---|---|---|---|---|---|
| ID19 | N.A. | N.A. | N.A. | N.A. | N.A. | N.A. |
| ID20 | 0.00 | 0.00 | 0.00 | 0.00 | 0.00 | 0.00 |
| ID21 | 0.00 | 0.00 | 0.00 | 0.00 | 0.00 | 0.01 |
| ID22 | 0.00 | 0.00 | 0.00 | 0.00 | 0.00 | 0.02 |
| ID23 | 0.00 | 0.00 | 0.00 | 0.00 | 0.00 | 0.00 |
| ID24 | -0.02 | 0.03 | 0.00 | 0.00 | 0.00 | 0.29 |
| ID25 | 0.00 | 0.00 | 0.00 | 0.00 | 0.00 | 0.00 |
| ID26 | 0.00 | 0.00 | 0.00 | 0.00 | 0.00 | 0.00 |
| ID27 | 0.00 | 0.06 | 0.00 | 0.00 | 0.00 | 0.11 |
| ID28 | -0.01 | 0.02 | 0.00 | 0.00 | 0.00 | 0.52 |
| ID29 | 0.00 | 0.00 | 0.00 | 0.00 | 0.00 | 0.00 |
| ID30 | 0.00 | 0.00 | 0.00 | 0.00 | 0.00 | 0.00 |
| ID31 | 0.00 | 0.00 | 0.00 | 0.00 | 0.00 | 0.00 |
| ID32 | 0.00 | 0.00 | 0.00 | 0.00 | 0.00 | 0.00 |
| ID33 | 0.48 | 0.14 | 0.00 | 0.00 | 0.00 | 0.02 |
| ID34 | 0.17 | 0.09 | 0.00 | 0.00 | 0.00 | 0.05 |
| ID35 | 0.03 | 0.07 | 0.00 | 0.00 | 0.00 | 0.07 |
| ID36 | N.A. | N.A. | N.A. | N.A. | N.A. | N.A. |
| ID37 | 0.00 | 0.00 | 0.00 | 0.00 | 0.00 | 0.00 |
| ID38 | 0.48 | 0.15 | 0.00 | 0.00 | 0.00 | 0.01 |
| ID39 | 0.07 | 0.08 | 0.00 | 0.00 | 0.00 | 0.07 |
| ID40 | 0.32 | 0.12 | 0.00 | 0.00 | 0.00 | 0.03 |
| ID41 | 0.00 | 0.00 | 0.00 | 0.00 | 0.00 | 0.16 |
| ID42 | 0.00 | 0.06 | 0.00 | 0.00 | 0.00 | 0.14 |
| ID43 | 0.00 | 0.00 | 0.00 | 0.00 | 0.00 | 0.00 |
| ID44 | 0.00 | 0.00 | 0.00 | 0.00 | 0.00 | 0.00 |
| ID45 | 0.00 | 0.00 | 0.00 | 0.00 | 0.00 | 0.00 |
| ID46 | 0.00 | 0.06 | 0.00 | 0.00 | 0.00 | 0.02 |
| ID47 | 0.00 | 0.00 | 0.00 | 0.00 | 0.00 | 0.14 |
| ID48 | 0.00 | 0.00 | 0.00 | 0.00 | 0.00 | 0.00 |
| ID49 | 0.16 | 0.09 | 0.00 | 0.00 | 0.00 | 0.02 |
| ID50 | 0.00 | 0.00 | 0.00 | 0.00 | 0.00 | 0.02 |
| ID51 | 0.00 | 0.00 | 0.00 | 0.00 | 0.00 | 0.87 |
| ID52 | 0.00 | 0.04 | 0.00 | 0.00 | 0.00 | 0.20 |
| ID53 | N.A. | N.A. | N.A. | N.A. | N.A. | N.A. |
| ID54 | 0.00 | 0.00 | 0.00 | 0.00 | 0.00 | 0.00 |
| ID55 | N.A. | N.A. | N.A. | N.A. | N.A. | N.A. |
| ID56 | 0.01 | 0.06 | 0.00 | 0.00 | 0.00 | 0.16 |
| ID57 | 0.00 | 0.03 | 0.00 | 0.00 | 0.00 | 0.01 |
| ID58 | -0.01 | 0.05 | 0.00 | 0.00 | 0.00 | 0.12 |
| ID59 | 0.00 | 0.05 | 0.00 | 0.00 | 0.00 | 0.10 |
| ID60 | 0.08 | 0.03 | 0.03 | 0.29 | 0.29 | 0.10 |
| ID61 | 0.69 | 0.20 | 0.00 | 0.00 | 0.00 | 0.01 |
| ID62 | -0.01 | 0.03 | 0.00 | 0.00 | 0.00 | 0.43 |
| ID63 | 0.00 | 0.00 | 0.00 | 0.00 | 0.00 | 0.00 |
| ID64 | 0.01 | 0.02 | 0.01 | 0.07 | 0.15 | 1.10 |
| ID65 | 0.00 | 0.00 | 0.00 | 0.00 | 0.00 | 0.00 |
| ID66 | 0.03 | 0.07 | 0.00 | 0.00 | 0.00 | 0.10 |
| ID67 | -0.01 | 0.04 | 0.00 | 0.00 | 0.00 | 0.12 |
| ID68 | 0.34 | 0.12 | 0.00 | 0.00 | 0.00 | 0.02 |

## G.7 SEIZURE DETECTION WITH BRAINBERT

Table 23 presents the detailed subject-by-subject breakdown of the performance of BrainBERT.

Table 23: **Details of seizure detection results of BrainBERT with 18 subject pre-training.** Kappa is the inter-rater agreement. The classification metrics report the raw and episodic metrics relevant for the seizure classification task.

| | | | Classification metrics | | | |
| | | | Raw | Episodic | | |
| Subject | Kappa | 95% CI | f1-score | f1-score | sensitivity | fp/h |
|---|---|---|---|---|---|---|
| ID19 | 0.00 | 0.00 | 0.00 | 0.00 | 0.00 | 0.00 |
| ID20 | 0.00 | 0.00 | 0.00 | 0.00 | 0.00 | 0.00 |
| ID21 | 0.00 | 0.00 | 0.00 | 0.00 | 0.00 | 0.00 |
| ID22 | 0.00 | 0.00 | 0.00 | 0.00 | 0.00 | 0.00 |
| ID23 | 0.00 | 0.00 | 0.00 | 0.00 | 0.00 | 0.00 |
| ID24 | 0.00 | 0.00 | 0.00 | 0.00 | 0.00 | 0.00 |
| ID25 | 0.00 | 0.00 | 0.00 | 0.00 | 0.00 | 0.00 |
| ID26 | 0.00 | 0.00 | 0.00 | 0.00 | 0.00 | 0.00 |
| ID27 | 0.00 | 0.00 | 0.00 | 0.00 | 0.00 | 0.00 |
| ID28 | 0.00 | 0.00 | 0.00 | 0.00 | 0.00 | 0.00 |
| ID29 | 0.00 | 0.00 | 0.00 | 0.00 | 0.00 | 0.00 |
| ID30 | 0.00 | 0.00 | 0.00 | 0.00 | 0.00 | 0.00 |
| ID31 | 0.00 | 0.00 | 0.00 | 0.00 | 0.00 | 0.00 |
| ID32 | 0.00 | 0.00 | 0.00 | 0.00 | 0.00 | 0.00 |
| ID33 | 0.00 | 0.00 | 0.00 | 0.00 | 0.00 | 0.00 |
| ID34 | 0.00 | 0.00 | 0.00 | 0.00 | 0.00 | 0.00 |
| ID35 | 0.00 | 0.00 | 0.00 | 0.00 | 0.00 | 0.00 |
| ID36 | 0.00 | 0.00 | 0.00 | 0.00 | 0.00 | 0.00 |
| ID37 | 0.00 | 0.00 | 0.00 | 0.00 | 0.00 | 0.00 |
| ID38 | 0.00 | 0.00 | 0.00 | 0.00 | 0.00 | 0.00 |
| ID39 | 0.00 | 0.00 | 0.00 | 0.00 | 0.00 | 0.00 |
| ID40 | 0.00 | 0.00 | 0.00 | 0.00 | 0.00 | 0.00 |
| ID41 | 0.00 | 0.00 | 0.00 | 0.00 | 0.00 | 0.00 |
| ID42 | 0.00 | 0.00 | 0.00 | 0.00 | 0.00 | 0.00 |
| ID43 | 0.00 | 0.00 | 0.00 | 0.00 | 0.00 | 0.00 |
| ID44 | 0.00 | 0.00 | 0.00 | 0.00 | 0.00 | 0.00 |
| ID45 | 0.00 | 0.00 | 0.00 | 0.00 | 0.00 | 0.00 |
| ID46 | 0.00 | 0.00 | 0.00 | 0.00 | 0.00 | 0.00 |
| ID47 | 0.00 | 0.00 | 0.00 | 0.00 | 0.00 | 0.00 |
| ID48 | 0.00 | 0.00 | 0.00 | 0.00 | 0.00 | 0.00 |
| ID49 | 0.00 | 0.00 | 0.00 | 0.00 | 0.00 | 0.00 |
| ID50 | 0.00 | 0.00 | 0.00 | 0.00 | 0.00 | 0.00 |
| ID51 | 0.00 | 0.00 | 0.00 | 0.00 | 0.00 | 0.00 |
| ID52 | 0.00 | 0.00 | 0.00 | 0.00 | 0.00 | 0.00 |
| ID53 | 0.00 | 0.00 | 0.00 | 0.00 | 0.00 | 0.00 |
| ID54 | 0.00 | 0.00 | 0.00 | 0.00 | 0.00 | 0.00 |
| ID55 | 0.00 | 0.00 | 0.00 | 0.00 | 0.00 | 0.00 |
| ID56 | 0.00 | 0.00 | 0.00 | 0.00 | 0.00 | 0.00 |
| ID57 | 0.00 | 0.00 | 0.00 | 0.00 | 0.00 | 0.00 |
| ID58 | 0.00 | 0.00 | 0.00 | 0.00 | 0.00 | 0.00 |
| ID59 | 0.00 | 0.00 | 0.00 | 0.00 | 0.00 | 0.00 |
| ID60 | 0.00 | 0.00 | 0.00 | 0.00 | 0.00 | 0.00 |
| ID61 | 0.00 | 0.00 | 0.00 | 0.00 | 0.00 | 0.00 |
| ID62 | 0.00 | 0.00 | 0.00 | 0.00 | 0.00 | 0.00 |
| ID63 | 0.00 | 0.00 | 0.00 | 0.00 | 0.00 | 0.00 |
| ID64 | 0.00 | 0.00 | 0.00 | 0.00 | 0.00 | 0.00 |
| ID65 | 0.00 | 0.00 | 0.00 | 0.00 | 0.00 | 0.00 |
| ID66 | 0.00 | 0.00 | 0.00 | 0.00 | 0.00 | 0.00 |
| ID67 | 0.00 | 0.00 | 0.00 | 0.00 | 0.00 | 0.00 |
| ID68 | 0.00 | 0.00 | 0.00 | 0.00 | 0.00 | 0.00 |

### G.8 Effects of the selection of channels

As discussed in previous sections, the number of channels can vary considerably across subjects.

We test three different scenarios:

- Automatic channel selection (Appendix G.3)
- Manual channel selection
- Evaluation with all channels

Beyond the automatic channel selection we use for the main results, we also select a subset of the channels (up to 50) which visually appear least noisy and most relevant. We also test the effect of including all channels, expecting it to decrease both the speed and performance due to the decrease of the overall signal-to-noise ratio.

Figure 15 shows that the performance decreases when we use a manual channel selection (for a detailed breakdown see Table 24). While this non-expert selection only has a minor impact on the overall performance, it still is notable that a standardized procedure produces better results.

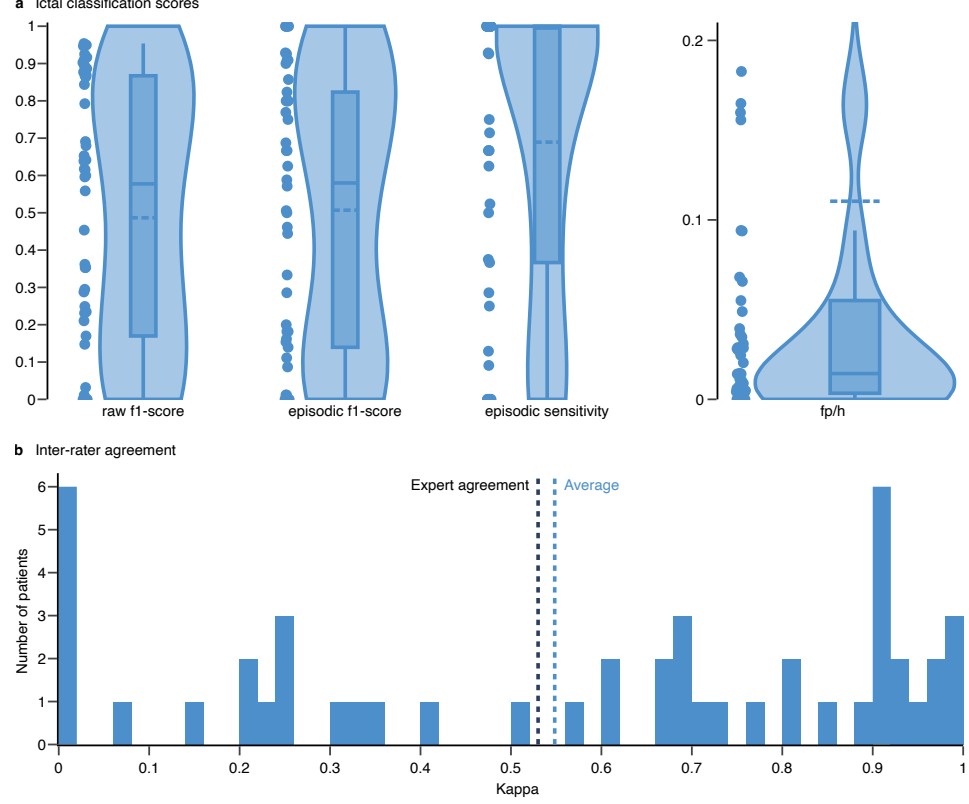

Figure 15: **Seizure detection with 18 patient pre-training and evaluation on a manual subset of channels.** **(a)** Seizure detection results of MVPFormer-S on unseen subjects evaluated on a manual subset of channels: the F1-score, sensitivity, and fp/h are reported. Raw results are computed without any post-processing of MVPFormer's output, while episodic results follow a common post-processing procedure which merge close ictal classifications. **(b)** Cohen's Kappa is used to measure the agreement between MVPFormer-S and the human expert. The average kappa is 0.54, lower than with the automatic channel selection routine. The distribution of kappa values clearly indicates that a minority of subjects are the source of most disagreement, consistent with the variability of inter-rater agreement among human experts.

Figure 16 shows that the performance decreases when we use all channels (for a detailed breakdown see Table 25). This is expected, as the noise contained in the entire recording increases together

Table 24: **Details of seizure detection results of MVPFormer-S with evaluation on a manual subset of channels.** Kappa is the inter-rater agreement. The classification metrics report the raw and episodic metrics relevant for the seizure classification task. The similarity reports the breakdown of the cosine similarity in each of the considered scenarios.

| | | | Classification metrics | | | |
| | | | Raw | Episodic | | |
| Subject | Kappa | 95% CI | f1-score | f1-score | sensitivity | fp/h |
|---------|-------|--------|----------|----------|-------------|------|
| ID19 | -0.05 | 0.02 | 0.00 | 0.00 | 0.00 | 0.97 |
| ID20 | 0.98 | 0.03 | 0.87 | 0.80 | 1.00 | 0.00 |
| ID21 | 0.92 | 0.07 | 0.86 | 0.80 | 1.00 | 0.00 |
| ID22 | 0.98 | 0.02 | 0.89 | 1.00 | 1.00 | 0.00 |
| ID23 | 0.26 | 0.03 | 0.17 | 0.14 | 0.09 | 0.16 |
| ID24 | 0.91 | 0.01 | 0.90 | 0.93 | 0.93 | 0.02 |
| ID25 | 0.35 | 0.12 | 0.00 | 0.00 | 0.00 | 0.03 |
| ID26 | 0.08 | 0.06 | 0.03 | 0.11 | 1.00 | 0.18 |
| ID27 | 0.99 | 0.01 | 0.93 | 1.00 | 1.00 | 0.00 |
| ID28 | 0.71 | 0.04 | 0.68 | 0.75 | 0.75 | 0.01 |
| ID29 | 0.25 | 0.05 | 0.29 | 0.20 | 0.13 | 0.03 |
| ID30 | 0.92 | 0.01 | 0.88 | 0.93 | 0.93 | 0.05 |
| ID31 | 0.98 | 0.00 | 0.95 | 1.00 | 1.00 | 0.00 |
| ID32 | 1.00 | 0.02 | 0.92 | 1.00 | 1.00 | 0.00 |
| ID33 | 0.00 | 0.05 | 0.00 | 0.00 | 0.00 | 0.26 |
| ID34 | 0.89 | 0.03 | 0.91 | 0.90 | 1.00 | 0.01 |
| ID35 | 0.85 | 0.03 | 0.87 | 0.82 | 1.00 | 0.03 |
| ID36 | 0.50 | 0.03 | 0.62 | 0.51 | 0.37 | 0.03 |
| ID37 | 0.74 | 0.09 | 0.69 | 0.50 | 1.00 | 0.02 |
| ID38 | 0.94 | 0.02 | 0.95 | 0.91 | 1.00 | 0.01 |
| ID39 | 0.92 | 0.05 | 0.89 | 0.80 | 1.00 | 0.01 |
| ID40 | 0.95 | 0.03 | 0.95 | 0.91 | 1.00 | 0.00 |
| ID41 | 0.61 | 0.05 | 0.65 | 0.67 | 1.00 | 0.04 |
| ID42 | 0.86 | 0.05 | 0.84 | 0.80 | 1.00 | 0.01 |
| ID43 | 0.90 | 0.03 | 0.64 | 1.00 | 1.00 | 0.00 |
| ID44 | 0.00 | 0.00 | 0.00 | 0.00 | 0.00 | 0.00 |
| ID45 | 0.01 | 0.02 | 0.00 | 0.01 | 1.00 | 1.06 |
| ID46 | 0.60 | 0.04 | 0.56 | 0.69 | 0.52 | 0.00 |
| ID47 | 0.26 | 0.08 | 0.29 | 0.16 | 1.00 | 0.09 |
| ID48 | 0.68 | 0.02 | 0.62 | 0.80 | 0.67 | 0.00 |
| ID49 | 0.85 | 0.03 | 0.79 | 0.86 | 1.00 | 0.01 |
| ID50 | 0.69 | 0.09 | 0.60 | 0.44 | 1.00 | 0.03 |
| ID51 | 0.35 | 0.09 | 0.23 | 0.15 | 1.00 | 0.07 |
| ID52 | 0.23 | 0.05 | 0.35 | 0.18 | 1.00 | 0.16 |
| ID53 | 0.12 | 0.06 | 0.21 | 0.09 | 1.00 | 0.16 |
| ID54 | 0.01 | 0.02 | 0.01 | 0.01 | 0.67 | 1.72 |
| ID55 | 0.37 | 0.08 | 0.35 | 0.29 | 1.00 | 0.07 |
| ID56 | 0.72 | 0.08 | 0.60 | 0.67 | 0.67 | 0.01 |
| ID57 | 0.00 | 0.00 | 0.00 | 0.00 | 0.00 | 0.00 |
| ID58 | 0.22 | 0.07 | 0.25 | 0.18 | 0.29 | 0.09 |
| ID59 | 0.00 | 0.00 | 0.00 | 0.00 | 0.00 | 0.00 |
| ID60 | 0.33 | 0.02 | 0.23 | 0.46 | 0.38 | 0.04 |
| ID61 | 0.00 | 0.00 | 0.00 | 0.00 | 0.00 | 0.00 |
| ID62 | 0.69 | 0.06 | 0.45 | 0.77 | 0.71 | 0.00 |
| ID63 | 0.94 | 0.02 | 0.90 | 1.00 | 1.00 | 0.00 |
| ID64 | 0.60 | 0.02 | 0.64 | 0.62 | 0.50 | 0.06 |
| ID65 | 0.50 | 0.06 | 0.35 | 0.59 | 0.62 | 0.03 |
| ID66 | 0.70 | 0.11 | 0.36 | 0.57 | 1.00 | 0.01 |
| ID67 | 0.26 | 0.07 | 0.15 | 0.33 | 0.25 | 0.01 |
| ID68 | 0.65 | 0.19 | 0.00 | 0.00 | 0.00 | 0.01 |

with the information content. MVPFormer's ability to generalize is not affected by the number of channels, but the noise affects the performance. Therefore, the optimal real-world operation of MVPFormer is obtained by selecting a subset of channels for detection.

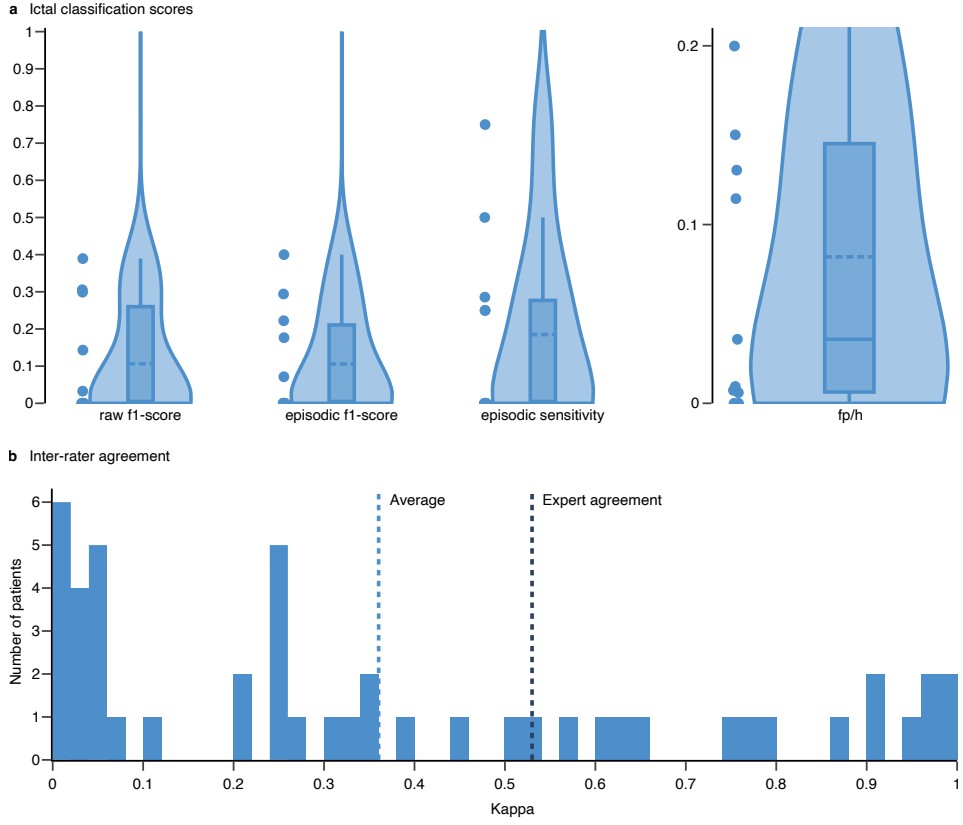

Figure 16: **Seizure detection with 18 patient pre-training and evaluation on all channels.** (a) Seizure detection results of MVPFormer-S on the unseen patients evaluated on all channels: the F1-score, sensitivity, and fp/h are reported. The performance metrics are reduced with respect to the results obtained when selecting a subset of the channels. MVPFormer is not affected by the number of channels, but the increase of noise emerging from all the channels contributes to a reduction in performance. **(b)** The average kappa is 0.36, reduced from the evaluation on a subset of channels.

Table 25: **Details of seizure detection results of MVPFormer-S with evaluation on all channels.** Kappa is the inter-rater agreement. The classification metrics report the raw and episodic metrics relevant for the seizure classification task.

| | | | Classification metrics | | | |
| | | | Raw | Episodic | | |
| **Subject** | **Kappa** | **95% CI** | f1-score | f1-score | sensitivity | fp/h |
|---|---|---|---|---|---|---|
| ID19 | -0.05 | 0.02 | 0.00 | 0.00 | 0.00 | 0.00 |
| ID20 | 0.84 | 0.09 | 0.47 | 0.50 | 0.50 | 0.50 |
| ID21 | 0.86 | 0.10 | 0.40 | 0.80 | 1.00 | 1.00 |
| ID22 | 0.91 | 0.04 | 0.81 | 0.89 | 1.00 | 1.00 |
| ID23 | 0.25 | 0.03 | 0.17 | 0.14 | 0.09 | 0.09 |
| ID24 | 0.91 | 0.01 | 0.90 | 0.93 | 0.93 | 0.93 |
| ID25 | 0.45 | 0.10 | 0.17 | 0.25 | 0.25 | 0.25 |
| ID26 | 0.25 | 0.08 | 0.04 | 0.22 | 1.00 | 1.00 |
| ID27 | 0.98 | 0.01 | 0.93 | 1.00 | 1.00 | 1.00 |
| ID28 | 0.64 | 0.07 | 0.42 | 0.67 | 0.50 | 0.50 |
| ID29 | 0.05 | 0.02 | 0.04 | 0.07 | 0.29 | 0.29 |
| ID30 | 0.20 | 0.03 | 0.31 | 0.19 | 0.11 | 0.11 |
| ID31 | 0.98 | 0.00 | 0.95 | 1.00 | 1.00 | 1.00 |
| ID32 | 0.98 | 0.04 | 0.92 | 1.00 | 1.00 | 1.00 |
| ID33 | 0.01 | 0.06 | 0.00 | 0.00 | 0.00 | 0.00 |
| ID34 | 0.04 | 0.02 | 0.06 | 0.04 | 1.00 | 1.00 |
| ID35 | 0.61 | 0.05 | 0.63 | 0.55 | 0.43 | 0.43 |
| ID36 | 0.50 | 0.03 | 0.62 | 0.51 | 0.37 | 0.37 |
| ID37 | 0.03 | 0.03 | 0.02 | 0.04 | 1.00 | 1.00 |
| ID38 | 0.97 | 0.01 | 0.97 | 1.00 | 1.00 | 1.00 |
| ID39 | 0.77 | 0.08 | 0.52 | 0.67 | 1.00 | 1.00 |
| ID40 | 0.37 | 0.06 | 0.50 | 0.31 | 0.80 | 0.80 |
| ID41 | 0.52 | 0.05 | 0.47 | 0.55 | 1.00 | 1.00 |
| ID42 | 0.63 | 0.07 | 0.54 | 0.57 | 1.00 | 1.00 |
| ID43 | 0.94 | 0.04 | 0.71 | 1.00 | 1.00 | 1.00 |
| ID44 | 0.00 | 0.02 | 0.00 | 0.00 | 0.00 | 0.00 |
| ID45 | 0.00 | 0.02 | 0.00 | 0.00 | 0.00 | 0.00 |
| ID46 | 0.06 | 0.03 | 0.03 | 0.06 | 0.19 | 0.19 |
| ID47 | 0.23 | 0.08 | 0.29 | 0.16 | 1.00 | 1.00 |
| ID48 | 0.25 | 0.03 | 0.18 | 0.31 | 0.33 | 0.33 |
| ID49 | 0.40 | 0.05 | 0.47 | 0.40 | 1.00 | 1.00 |
| ID50 | 0.05 | 0.04 | 0.05 | 0.06 | 1.00 | 1.00 |
| ID51 | 0.06 | 0.06 | 0.05 | 0.07 | 1.00 | 1.00 |
| ID52 | 0.00 | 0.01 | 0.01 | 0.01 | 1.00 | 1.00 |
| ID53 | 0.13 | 0.07 | 0.21 | 0.09 | 1.00 | 1.00 |
| ID54 | 0.01 | 0.02 | 0.01 | 0.01 | 0.67 | 0.67 |
| ID55 | 0.37 | 0.08 | 0.35 | 0.29 | 1.00 | 1.00 |
| ID56 | 0.33 | 0.07 | 0.34 | 0.31 | 0.67 | 0.67 |
| ID57 | 0.00 | 0.00 | 0.00 | 0.00 | 0.00 | 0.00 |
| ID58 | 0.29 | 0.12 | 0.00 | 0.00 | 0.00 | 0.00 |
| ID59 | 0.02 | 0.05 | 0.00 | 0.00 | 0.00 | 0.00 |
| ID60 | 0.00 | 0.00 | 0.00 | 0.00 | 0.00 | 0.00 |
| ID61 | 0.00 | 0.00 | 0.00 | 0.00 | 0.00 | 0.00 |
| ID62 | 0.04 | 0.05 | 0.03 | 0.07 | 0.29 | 0.29 |
| ID63 | 0.21 | 0.05 | 0.39 | 0.18 | 0.75 | 0.75 |
| ID64 | 0.25 | 0.03 | 0.30 | 0.29 | 0.25 | 0.25 |
| ID65 | 0.04 | 0.07 | 0.00 | 0.00 | 0.00 | 0.00 |
| ID66 | 0.76 | 0.16 | 0.31 | 0.40 | 0.50 | 0.50 |
| ID67 | 0.17 | 0.07 | 0.14 | 0.22 | 0.25 | 0.25 |
| ID68 | 0.61 | 0.18 | 0.00 | 0.00 | 0.00 | 0.01 |

## G.9 EFFECTS OF THE SCALE OF THE PRE-TRAINING DATASET

The performance of LLMs as the size of their training dataset increases has been investigated quite thoroughly (Hoffmann et al., 2022; Kaplan et al., 2020), giving rise to a variety of scaling laws. Following Chinchilla's scaling law, a model with 75 million parameters like MVPFormer-S should be trained with around 2 billion tokens, while we only have 400 million at our disposal.

The architecture of the model and the nature of the training data, however, make it unclear whether such laws can be adopted for MVPFormer as well. We investigate this behavior by continuing the training of MVPFormer-S on 40 more subjects, to bring the total to 58 pre-training subjects for almost 7,000 hours of iEEG recordings. In particular, MVPFormer is initially trained on 304 ictal events, and then further on 323 more. Therefore, we are left with 10 unseen subject to test the downstream seizure detection task.

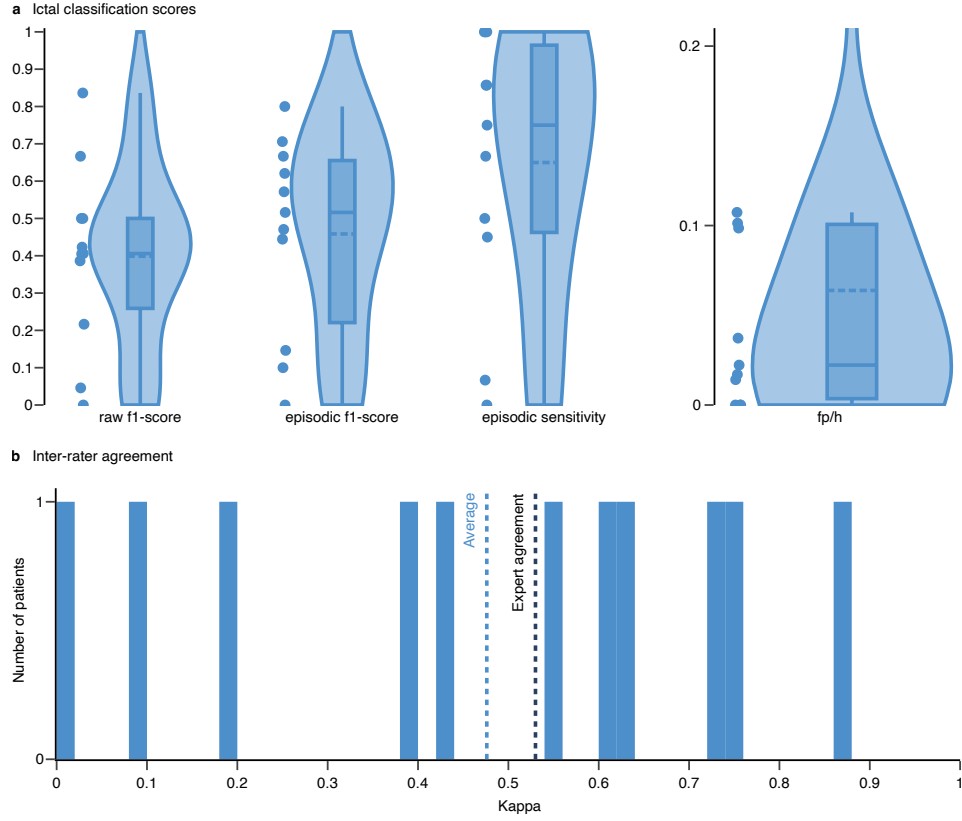

Figure 17: **Seizure detection with 58 patient pre-training and manual channel selection. (a)** Seizure detection results of MVPFormer-S on the 10 remaining unseen patients using manual channel selection: the F1-score, sensitivity, and fp/h are reported. All performance metrics are improved with respect to the original MVPFormer model. The raw and episodic F1-scores are significantly different here, indicating that MVPFormer benefits of the episode merging effect of post-processing on these patients. The false positive rate has decreased further with the scale of the pre-training dataset. **(b)** Inter-rater agreement of MVPFormer with the human expert: Cohen's kappa is used to measure the agreement between the artificial assistant and the human expert. The average kappa is increased to 0.48. The distribution of kappa values again indicates that there is considerable variability in the agreement.

Figure 17 shows the performance of the 58-subject MVPFormer-S on the 10 unseen subjects using a manual subset of channels (for a detailed breakdown see Table 26). On the other hand, Figure 18 shows the results of the original MVPFormer-S model on those same 10 subjects (for a detailed breakdown see Table 27). All performance metrics improve with a growing pre-training dataset size, although on a small test cohort, indicating that increasing the number of subjects in the pre-training dataset has a net positive effect on the downstream classification task.

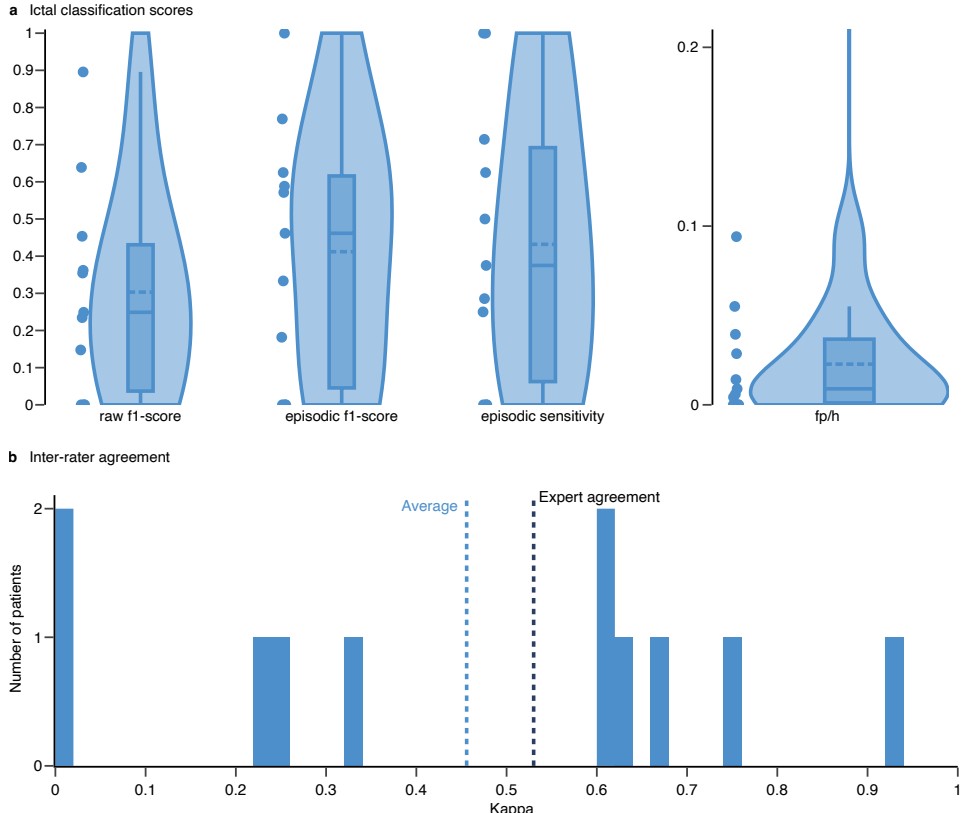

Figure 18: **Seizure detection with 18 subjects pre-training on a selection of 10 unseen subjects and manual channel selection.** (a) Seizure detection results of MVPFormer-S on the 10 subjects excluded from the 58-subjects model using manual channel selection: the F1-score, sensitivity, and fp/h are reported. The raw and episodic F1-scores are significantly different here, indicating that MVPFormer benefits of the episode merging effect of post-processing on these patients. These results are a subset of those presented in the Results section. (b) Inter-rater agreement of MVPFormer with the human expert: Cohen's kappa is used to measure the agreement between the artificial assistant and the human expert. The average kappa is 0.46, reduced from the overall results indicating that these subjects are more difficult than average. The distribution of kappa values again indicates that there is considerable variability in the agreement.

Table 26: **Details of seizure detection results of MVPFormer-S with 58 patient pre-training and manual channel selection.** Kappa is the inter-rater agreement. The classification metrics report the raw and episodic metrics relevant for the seizure classification task.

| | | | Classification metrics | | | |
| | | | Raw | Episodic | | |
| **Subject** | **Kappa** | 95% CI | f1-score | f1-score | sensitivity | fp/h |
|---|---|---|---|---|---|---|
| ID59 | 0.07 | 0.06 | 0.05 | 0.10 | 0.07 | 0.07 |
| ID60 | 0.54 | 0.02 | 0.39 | 0.47 | 0.50 | 0.50 |
| ID61 | 0.00 | 0.00 | 0.00 | 0.00 | 0.00 | 0.00 |
| ID62 | 0.65 | 0.05 | 0.50 | 0.71 | 0.86 | 0.86 |
| ID63 | 0.76 | 0.04 | 0.84 | 0.67 | 1.00 | 1.00 |
| ID64 | 0.60 | 0.02 | 0.67 | 0.62 | 0.45 | 0.45 |
| ID65 | 0.50 | 0.04 | 0.41 | 0.52 | 1.00 | 1.00 |
| ID66 | 0.79 | 0.12 | 0.50 | 0.57 | 1.00 | 1.00 |
| ID67 | 0.22 | 0.04 | 0.22 | 0.15 | 0.75 | 0.75 |
| ID68 | 0.73 | 0.09 | 0.42 | 0.80 | 0.67 | 0.00 |

Table 27: **Details of seizure detection results of MVPFormer-S with 18 patient pre-training on a selection of 10 subjects and manual channel selection.** Kappa is the inter-rater agreement. The classification metrics report the raw and episodic metrics relevant for the seizure classification task.

| | | | Classification metrics | | | |
| | | | Raw | Episodic | | |
| **Subject** | **Kappa** | 95% CI | f1-score | f1-score | sensitivity | fp/h |
|---|---|---|---|---|---|---|
| ID59 | 0.00 | 0.00 | 0.00 | 0.00 | 0.00 | 0.00 |
| ID60 | 0.33 | 0.02 | 0.23 | 0.46 | 0.38 | 0.04 |
| ID61 | 0.00 | 0.00 | 0.00 | 0.00 | 0.00 | 0.00 |
| ID62 | 0.69 | 0.06 | 0.45 | 0.77 | 0.71 | 0.00 |
| ID63 | 0.94 | 0.02 | 0.90 | 1.00 | 1.00 | 0.00 |
| ID64 | 0.60 | 0.02 | 0.64 | 0.62 | 0.50 | 0.06 |
| ID65 | 0.50 | 0.06 | 0.35 | 0.59 | 0.62 | 0.03 |
| ID66 | 0.70 | 0.11 | 0.36 | 0.57 | 1.00 | 0.01 |
| ID67 | 0.26 | 0.07 | 0.15 | 0.33 | 0.25 | 0.01 |
| ID68 | 0.65 | 0.19 | 0.00 | 0.00 | 0.00 | 0.01 |

## G.10 PATIENT CLASSIFICATION DIFFICULTY

In the medical practice each patient has unique seizure presentations, though they might be broadly grouped into different categories (Shokooh et al., 2021). As an effect, some patients have seizures which can be considered more typical (Figure 4a), and hence easier to detect, while others might have very atypical events (Figure 4b). There might be broad disagreement among neurologists over these atypical seizures, and at the same time no disagreement at all over the typical patients (Gotman, 2011).

This phenomenon intuitively creates a difficulty scale among the patients, which also affects MVP-Former and contributes to the spread of performance between the model and the human expert. To better assess the impact of this latent patient classification difficulty we performed a multiple correlation analysis using the total recording length, the number of seizures, and the frequency of seizures to predict the kappa score, yielding an $R^2$ of $0.054$. The model performance is thus independent of the three variables, and we believe the difficulty might help explain most of the variance. The literature supports this hypothesis, as the subjects themselves can account for up to 65% of the variance (Grant et al., 2014) while the clinical setup itself has no impact.

## G.11 CHANNEL CONNECTIVITY MAP

Generating future iEEG signal embedding implicitly places the greatest emphasis on the dimension of time, but to do so it is necessary to consider the interactions between channels as well. MVP-Former thus takes into consideration all electrodes concurrently, as electric potentials flow across different areas and circuits in the brain following their intrinsic connections and constraints (Betzel et al., 2019; Pang et al., 2023). The number of channels depends on the number of electrodes decided for a specific patient and the clinical setup. As iEEG implantations are decided on a case-by-case basis by a physician, there is no uniform standard on where to place the electrodes, in contrast with the 10-20 system (Jasper, 1958) for EEG. Therefore, we cannot give MVPFormer any a priori knowledge of how the channels will interact in space, and the model has to learn it on its own. MVPA enables our model to dynamically learn these connections to build an internal map of the flow, becoming independent from a specific electrode configuration.

It is well-known that two neighboring brain regions might not be as strongly connected as two faraway regions. The relationship between the electrodes (and hence the channels) mirrors this behavior. To truly understand the link between two channels the model must build a map of the connection strength between different brain areas and how these connections impact the diffusion of electric fields across channels. In MVPA, this understanding is the underpinning of the channel-based component. In particular, the complex interplay between the query, the channel codebook, and the channel attention, acts as the first level of processing. Further, the deep structure with multiple layers provides more representational power, as is typical of deep models.

Figure 19 shows that the channel-based MVPA component encodes a form of the brain connectivity map. Initially, the map is random as MVPA is randomly initialized. As training progresses, the attention magnitude among the channels starts to differentiate, building a map of the connection strength. The map is dominated by the diagonal component, which indicates that in general neighboring channels are more related than distant channels. However, it is possible to clearly distinguish clusters of strongly connected electrodes and also skipped connections, which possibly refer to strong connections between distant regions. Since the channel distance is relative, it can apply to arbitrary clinical setups and is not limited to already seen channels. Moreover, as the channel attention is a function of both the query content and the channel distance, the combination of the two can effectively modulate the attention even on unseen subjects.

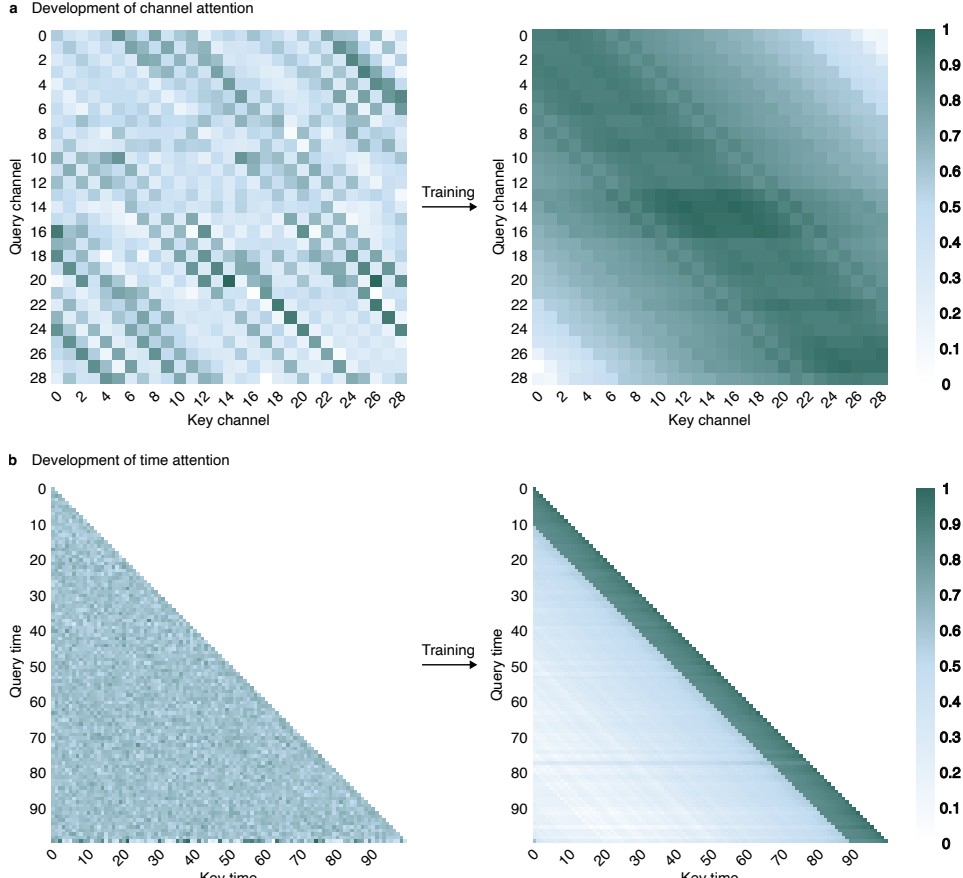

Figure 19: **Attention components before and after training.** **(a)** The channel-based attention component is randomly initialized. At the end of training, it shows the diagonal structure which indicates that the relationship between the channels is mainly one of proximity. This is expected, as nearby channels are expected to be more closely related, and showcases MVPA's learning outcome. **(b)** The time-based attention component is also randomly initialized. At the end of training, it shows that segments which are close in time are more related. Particularly, few closest segments are attended to more strongly, as the content-based attention's lookback windows is limited to a few segments.

## G.12 Results on the MAYO and FNUSA datasets

We test MVPFormer additionally against Brant-2 on the iEEG MAYO and FNUSA datasets. Both datasets are single-channel iEEG datasets containing both physiological and pathological activity. We use these datasets to evaluate MVPFormer's performance on extremely noisy data, and to compare its resilience with the SOTA.

The MAYO dataset contains 24 patients for a total of 130 hours of data. In particular, 36% of the data is non-ictal, 9% is ictal, and 53% is noise. Moreover, 18 subjects contain no ictal activity and 9 patients are fully noise, with 13 being majority noise. The FNUSA dataset contains 14 patients for a total of 160 hours. 48% of the dataset is non-ictal, 27% is ictal, and 23% is noise. Moreover, the data of 2 subjects is completely noise, and the data of 3 other subjects is fully ictal. Both datasets are much smaller in scale than the SWEC iEEG dataset, and are heavily dominated by noise, both artifact and powerline. In particular, the SWEC iEEG dataset is almost 2,000 times larger and is also carefully evaluated by an expert neurologist to remove channels which contain too much noise or artifacts.

As we wish to assess MVPFormer in a realistic, real-world scenario, we do not remove any noise from the dataset but test them as-is. In particular, we consider noise and physiological activity as one category, and pathological activity as another. However, kappa scores are not meaningful with such small datasets, therefore we provide the aggregate F1-score, and the average sensitivity and specificity. Specifically, given the fact that many patients do not contain ictal activity, we do not compute the average F1-score across subjects, but pool together all subject's results and compute the aggregate F1-score.

We use the same MVPFormer models pre-trained on our SWEC iEEG dataset, and train a specific classification head for either MAYO or FNUSA by fine-tuning on the first four patients. Then, we test on the remaining patients. We also use the Brant-2 model whose pre-trained weights are publicly available, and fine-tune in the same manner as MVPFormer using the fine-tuning code provided by the authors.

The results can be found in Tables 28 and 29. Given the very low signal-to-noise ratio of both datasets, overall performance is affected. On the FNUSA dataset, where the amount of noise is more moderate, all models perform similarly, with MVPFormer-M showing a higher specificity. However, MVPFormer has a clear advantage on the MAYO dataset, with almost double the F1-score with respect to Brant-2. The difference between MVPFormer-S and MVPFormer-M is minimal, as the sizes of the datasets involved are too small to fully train a very large model such as MVPFormer-M (see Appendix G.4 for more information).

Table 28: **Summary of seizure detection results of all models on the MAYO iEEG dataset.** Kappa is the inter-rater agreement. The classification metrics report the raw and episodic metrics relevant for the seizure classification task.

| Model | F1-score | Sensitivity | Specificity |
|---|---|---|---|
| MVPFormer-M | **0.36** | 0.38 | **0.91** |
| MVPFormer-S | 0.35 | 0.41 | 0.88 |
| Brant-2 | 0.19 | 1.00 | 0.18 |

Table 29: **Summary of seizure detection results of all models on the FNUSA iEEG dataset.** Kappa is the inter-rater agreement. The classification metrics report the raw and episodic metrics relevant for the seizure classification task.

| Model | F1-score | Sensitivity | Specificity |
|---|---|---|---|
| MVPFormer-M | **0.46** | 0.94 | **0.10** |
| MVPFormer-S | 0.46 | 0.99 | 0.03 |
| Brant-2 | 0.46 | 0.99 | 0.02 |

### G.13 ABLATION OF THE PREDICTION TASK

We design MVPFormer with a two-phase training regime. First, during the generative pre-training task MVPFormer learns to predict the neuronal activity. Second, during the classification task it needs to correctly classify ictal periods. To determine the significance of the generative task on the classification task, we train MVPFormer only on the classification task and compare its performance with the full architecture on the manual selection of channels. Figure 20 and Table 30 clearly indicate that the generative task is of fundamental importance to the overall architecture, with a Kappa score decrease to 0.52. This is below the original result of 0.54 and below the human agreement threshold. Moreover, the distribution of agreement has flattened, with an overall decrease of performance across the board and an increase of subjects with no agreement. This suggests that without pre-training the generalization capability of MVPFormer suffers. Therefore, the generative task is necessary and is a significant contributor to learning.

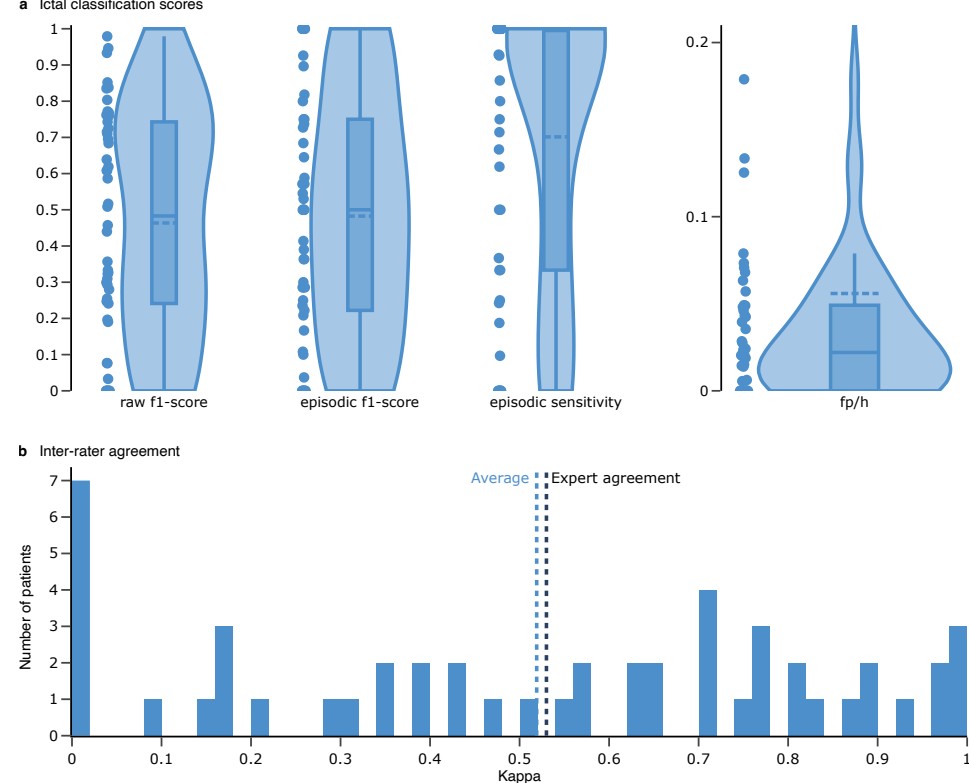

Figure 20: **Seizure detection with no generative pre-training and 18 subjects classification training on the manual selection of channels.** (a) Seizure detection results of MVPFormer-S on 40 unseen subjects which are part of the training set for the 58-subjects model and manual selection of channels: the F1-score, sensitivity, and fp/h are reported. The raw and episodic F1-scores are notably lower here with respect to the 58-subjects model. This is expected given the 58-subject model is pre-trained on these subjects. These results are a subset of those presented in the Results section. (b) Cohen's kappa is used to measure the agreement between the artificial assistant and the human expert. The average kappa is 0.56, competitive with expert agreement but, as expected, reduced from the 58-subjects pre-trained model.

Table 30: **Seizure detection with no generative pre-training and 18 subjects classification training on the manual selection of channels.** Kappa is the inter-rater agreement. The classification metrics report the raw and episodic metrics relevant for the seizure classification task.

| Subject | Kappa | 95% CI | Raw f1-score | Episodic f1-score | sensitivity | fp/h |
|---|---|---|---|---|---|---|
| ID19 | 0.37 | 0.02 | 0.46 | 0.30 | 0.19 | 0.05 |
| ID20 | 0.99 | 0.02 | 0.98 | 1.00 | 1.00 | 0.00 |
| ID21 | 0.47 | 0.09 | 0.30 | 0.29 | 1.00 | 0.04 |
| ID22 | 0.74 | 0.06 | 0.70 | 0.73 | 1.00 | 0.02 |
| ID23 | 0.37 | 0.02 | 0.51 | 0.36 | 0.24 | 0.07 |
| ID24 | 0.88 | 0.01 | 0.84 | 0.90 | 0.93 | 0.05 |
| ID25 | 0.00 | 0.00 | 0.00 | 0.00 | 0.00 | 0.00 |
| ID26 | 0.43 | 0.10 | 0.19 | 0.29 | 1.00 | 0.06 |
| ID27 | 0.95 | 0.02 | 0.84 | 1.00 | 1.00 | 0.00 |
| ID28 | 0.73 | 0.04 | 0.80 | 0.75 | 0.75 | 0.01 |
| ID29 | 0.17 | 0.05 | 0.28 | 0.17 | 0.10 | 0.01 |
| ID30 | 0.90 | 0.01 | 0.85 | 0.93 | 0.93 | 0.05 |
| ID31 | 0.98 | 0.00 | 0.95 | 1.00 | 1.00 | 0.00 |
| ID32 | 0.98 | 0.05 | 0.93 | 1.00 | 1.00 | 0.00 |
| ID33 | 0.00 | 0.00 | 0.00 | 0.00 | 0.00 | 0.00 |
| ID34 | 0.78 | 0.03 | 0.84 | 0.82 | 1.00 | 0.02 |
| ID35 | 0.71 | 0.03 | 0.77 | 0.74 | 1.00 | 0.05 |
| ID36 | 0.59 | 0.03 | 0.68 | 0.53 | 0.37 | 0.01 |
| ID37 | 0.66 | 0.10 | 0.61 | 0.50 | 1.00 | 0.02 |
| ID38 | 0.59 | 0.05 | 0.64 | 0.59 | 1.00 | 0.04 |
| ID39 | 0.76 | 0.09 | 0.71 | 0.57 | 1.00 | 0.02 |
| ID40 | 0.67 | 0.05 | 0.77 | 0.57 | 0.80 | 0.02 |
| ID41 | 0.75 | 0.05 | 0.72 | 0.75 | 1.00 | 0.02 |
| ID42 | 0.97 | 0.04 | 0.74 | 1.00 | 1.00 | 0.00 |
| ID43 | 0.15 | 0.09 | 0.00 | 0.00 | 0.00 | 0.06 |
| ID44 | 0.98 | 0.03 | 0.76 | 1.00 | 1.00 | 0.00 |
| ID45 | 0.03 | 0.05 | 0.03 | 0.04 | 1.00 | 0.33 |
| ID46 | 0.60 | 0.03 | 0.62 | 0.68 | 0.62 | 0.03 |
| ID47 | 0.79 | 0.08 | 0.72 | 0.55 | 1.00 | 0.02 |
| ID48 | 0.88 | 0.01 | 0.76 | 1.00 | 1.00 | 0.00 |
| ID49 | 0.08 | 0.03 | 0.08 | 0.11 | 1.00 | 0.71 |
| ID50 | 0.36 | 0.08 | 0.36 | 0.24 | 1.00 | 0.07 |
| ID51 | 0.52 | 0.10 | 0.32 | 0.22 | 1.00 | 0.04 |
| ID52 | 0.50 | 0.07 | 0.29 | 0.50 | 1.00 | 0.04 |
| ID53 | 0.20 | 0.07 | 0.33 | 0.10 | 1.00 | 0.13 |
| ID54 | 0.61 | 0.12 | 0.31 | 0.29 | 0.33 | 0.01 |
| ID55 | 0.35 | 0.08 | 0.25 | 0.25 | 1.00 | 0.08 |
| ID56 | 0.82 | 0.06 | 0.61 | 0.80 | 0.67 | 0.00 |
| ID57 | 0.00 | 0.00 | 0.00 | 0.00 | 0.00 | 0.00 |
| ID58 | 0.17 | 0.04 | 0.20 | 0.21 | 0.71 | 0.26 |
| ID59 | 0.00 | 0.00 | 0.00 | 0.00 | 0.00 | 0.00 |
| ID60 | 0.16 | 0.03 | 0.08 | 0.36 | 0.25 | 0.02 |
| ID61 | 0.00 | 0.00 | 0.00 | 0.00 | 0.00 | 0.00 |
| ID62 | 0.42 | 0.06 | 0.44 | 0.41 | 0.86 | 0.07 |
| ID63 | 0.54 | 0.09 | 0.25 | 0.57 | 0.50 | 0.01 |
| ID64 | 0.62 | 0.02 | 0.59 | 0.65 | 0.50 | 0.03 |
| ID65 | 0.31 | 0.04 | 0.24 | 0.39 | 1.00 | 0.18 |
| ID66 | 0.69 | 0.11 | 0.52 | 0.50 | 1.00 | 0.02 |
| ID67 | 0.00 | 0.06 | 0.00 | 0.00 | 0.00 | 0.13 |
| ID68 | 0.78 | 0.15 | 0.33 | 0.50 | 0.33 | 0.00 |

### G.14 RESILIENCE TO NOISE

A critical aspect of any clinical environment are the perturbations and noise in the recorder signal. The SWEC iEEG dataset is collected in a real-world environment and is pre-processed following the indications of an expert neurologist, to guarantee a clinically-relevant scenario to test any model, including our MVPFormer against.

In addition to the inherent noise of the iEEG signal, we now further disturb the recordings using white gaussian noise to evaluate the resilience of MVPFormer to additional perturbations. Specifically, we add noise to achieve an SNR of 30dB (Table 32), 40dB (Table 33), 50dB (Table 34), and 60dB (Table 35) to understand the behavior of MVPFormer.

Table 31: **Results on seizure detection with noise.** We evaluate MVPFormer's performance on the seizure detection task at varying levels of injected white gaussian noise.

| SNR | Kappa | Episodic | | | Raw |
|---|---|---|---|---|---|
| | | f1 | sens | fp/h | f1 |
| None | 0.61 | 0.59 | 0.72 | 0.15 | 0.51 |
| 60dB | 0.58 | 0.54 | 0.71 | 0.12 | 0.49 |
| 50dB | 0.54 | 0.50 | 0.72 | 0.13 | 0.47 |
| 40dB | 0.36 | 0.34 | 0.74 | 0.46 | 0.31 |
| 30dB | 0.12 | 0.12 | 0.71 | 1.22 | 0.10 |

A summary of the results is presented in Table 31. Remarkably, the performance remains above the expert-level threshold until 40dB of SNR, indicating that MVPFormer is resilient to noise. In particular, while sensitivity remains high, the number of false positives rapidly increases. This is consistent with the fact that seizures often appear as high-frequency activity, which can mislead the model when in a noisy environment.

Table 32: **Details of seizure detection results of MVPFormer-S with 30dB SNR.** Kappa is the inter-rater agreement. The classification metrics report the raw and episodic metrics relevant for the seizure classification task. The similarity reports the breakdown of the cosine similarity in each of the considered scenarios.

| | | | **Classification metrics** | | | |
| | | | Raw | Episodic | | |
| **Subject** | **Kappa** | 95% CI | f1-score | f1-score | sensitivity | fp/h |
|---|---|---|---|---|---|---|
| ID19 | 0.01 | 0.00 | 0.02 | 0.28 | 0.62 | 2.07 |
| ID20 | 0.05 | 0.04 | 0.09 | 0.04 | 1.00 | 0.36 |
| ID21 | 0.01 | 0.03 | 0.01 | 0.02 | 1.00 | 0.78 |
| ID22 | 0.08 | 0.04 | 0.06 | 0.12 | 1.00 | 0.36 |
| ID23 | -0.04 | 0.01 | 0.00 | 0.06 | 0.15 | 2.90 |
| ID24 | 0.06 | 0.01 | 0.06 | 0.12 | 0.50 | 2.24 |
| ID25 | 0.01 | 0.02 | 0.00 | 0.02 | 0.50 | 2.19 |
| ID26 | 0.00 | 0.01 | 0.00 | 0.01 | 1.00 | 3.11 |
| ID27 | 0.00 | 0.00 | 0.00 | 0.02 | 0.75 | 3.54 |
| ID28 | 0.23 | 0.05 | 0.26 | 0.35 | 0.75 | 0.15 |
| ID29 | 0.03 | 0.01 | 0.02 | 0.05 | 0.26 | 2.03 |
| ID30 | 0.01 | 0.00 | 0.02 | 0.35 | 0.56 | 1.05 |
| ID31 | 0.08 | 0.01 | 0.09 | 0.18 | 1.00 | 3.54 |
| ID32 | 0.02 | 0.03 | 0.02 | 0.02 | 1.00 | 0.79 |
| ID33 | 0.23 | 0.10 | 0.00 | 0.00 | 0.00 | 0.06 |
| ID34 | 0.01 | 0.01 | 0.01 | 0.04 | 1.00 | 2.40 |
| ID35 | 0.03 | 0.01 | 0.05 | 0.05 | 0.86 | 2.10 |
| ID36 | 0.00 | 0.00 | 0.01 | 0.13 | 0.45 | 2.13 |
| ID37 | 0.01 | 0.02 | 0.01 | 0.01 | 1.00 | 1.61 |
| ID38 | 0.33 | 0.05 | 0.48 | 0.34 | 1.00 | 0.11 |
| ID39 | 0.42 | 0.07 | 0.40 | 0.36 | 1.00 | 0.05 |
| ID40 | 0.06 | 0.03 | 0.09 | 0.08 | 0.80 | 0.42 |
| ID41 | 0.00 | 0.01 | 0.01 | 0.03 | 1.00 | 2.49 |
| ID42 | 0.10 | 0.04 | 0.15 | 0.14 | 1.00 | 0.28 |
| ID43 | 0.77 | 0.05 | 0.57 | 0.80 | 1.00 | 0.02 |
| ID44 | 0.21 | 0.09 | 0.00 | 0.00 | 0.00 | 0.08 |
| ID45 | 0.01 | 0.03 | 0.00 | 0.02 | 1.00 | 0.80 |
| ID46 | 0.18 | 0.03 | 0.14 | 0.25 | 0.43 | 0.31 |
| ID47 | 0.00 | 0.01 | 0.00 | 0.01 | 1.00 | 2.80 |
| ID48 | 0.90 | 0.01 | 0.77 | 1.00 | 1.00 | 0.00 |
| ID49 | 0.01 | 0.01 | 0.01 | 0.05 | 1.00 | 1.48 |
| ID50 | 0.02 | 0.02 | 0.02 | 0.02 | 1.00 | 0.94 |
| ID51 | 0.22 | 0.08 | 0.11 | 0.11 | 1.00 | 0.11 |
| ID52 | 0.04 | 0.02 | 0.05 | 0.03 | 1.00 | 0.99 |
| ID53 | 0.01 | 0.02 | 0.01 | 0.01 | 1.00 | 2.18 |
| ID54 | 0.00 | 0.01 | 0.00 | 0.01 | 1.00 | 4.02 |
| ID55 | 0.00 | 0.01 | 0.00 | 0.01 | 1.00 | 2.60 |
| ID56 | 0.01 | 0.01 | 0.01 | 0.02 | 0.67 | 1.79 |
| ID57 | -0.01 | 0.03 | 0.01 | 0.07 | 0.08 | 0.19 |
| ID58 | 0.00 | 0.01 | 0.00 | 0.01 | 0.14 | 1.86 |
| ID59 | 0.16 | 0.06 | 0.11 | 0.20 | 0.13 | 0.03 |
| ID60 | 0.26 | 0.02 | 0.27 | 0.11 | 0.50 | 1.16 |
| ID61 | 0.02 | 0.04 | 0.00 | 0.00 | 0.00 | 0.35 |
| ID62 | 0.13 | 0.05 | 0.18 | 0.13 | 1.00 | 0.40 |
| ID63 | 0.13 | 0.03 | 0.12 | 0.12 | 1.00 | 0.31 |
| ID64 | 0.01 | 0.01 | 0.02 | 0.19 | 0.45 | 1.79 |
| ID65 | 0.02 | 0.03 | 0.01 | 0.03 | 0.25 | 0.87 |
| ID66 | 0.62 | 0.14 | 0.28 | 0.25 | 0.50 | 0.02 |
| ID67 | 0.03 | 0.02 | 0.05 | 0.02 | 0.50 | 1.47 |
| ID68 | 0.57 | 0.09 | 0.32 | 0.40 | 0.67 | 0.03 |

Table 33: **Details of seizure detection results of MVPFormer-S with 40dB SNR.** Kappa is the inter-rater agreement. The classification metrics report the raw and episodic metrics relevant for the seizure classification task. The similarity reports the breakdown of the cosine similarity in each of the considered scenarios.

| | | | **Classification metrics** | | | |
| | | | Raw | Episodic | | |
| **Subject** | **Kappa** | 95% CI | f1-score | f1-score | sensitivity | fp/h |
|---|---|---|---|---|---|---|
| ID19 | 0.01 | 0.00 | 0.02 | 0.28 | 0.44 | 1.24 |
| ID20 | 0.05 | 0.04 | 0.11 | 0.03 | 1.00 | 0.39 |
| ID21 | 0.03 | 0.04 | 0.04 | 0.04 | 1.00 | 0.42 |
| ID22 | 0.81 | 0.05 | 0.74 | 0.73 | 1.00 | 0.02 |
| ID23 | 0.11 | 0.02 | 0.10 | 0.12 | 0.15 | 1.06 |
| ID24 | 0.50 | 0.02 | 0.52 | 0.55 | 0.86 | 0.44 |
| ID25 | 0.01 | 0.02 | 0.00 | 0.03 | 0.50 | 1.11 |
| ID26 | 0.01 | 0.02 | 0.01 | 0.02 | 1.00 | 1.27 |
| ID27 | 0.04 | 0.02 | 0.03 | 0.05 | 0.88 | 1.65 |
| ID28 | 0.18 | 0.04 | 0.26 | 0.21 | 0.75 | 0.32 |
| ID29 | 0.53 | 0.03 | 0.44 | 0.27 | 0.29 | 0.19 |
| ID30 | 0.30 | 0.01 | 0.27 | 0.40 | 0.93 | 1.76 |
| ID31 | 0.79 | 0.01 | 0.80 | 0.81 | 1.00 | 0.19 |
| ID32 | 0.20 | 0.07 | 0.21 | 0.15 | 1.00 | 0.11 |
| ID33 | 0.54 | 0.15 | 0.00 | 0.00 | 0.00 | 0.02 |
| ID34 | 0.36 | 0.04 | 0.53 | 0.33 | 1.00 | 0.19 |
| ID35 | 0.42 | 0.04 | 0.61 | 0.44 | 0.86 | 0.13 |
| ID36 | 0.19 | 0.02 | 0.17 | 0.21 | 0.37 | 0.77 |
| ID37 | 0.05 | 0.04 | 0.08 | 0.05 | 1.00 | 0.36 |
| ID38 | 0.93 | 0.03 | 0.95 | 0.91 | 1.00 | 0.01 |
| ID39 | 0.90 | 0.07 | 0.83 | 0.80 | 1.00 | 0.01 |
| ID40 | 0.83 | 0.04 | 0.84 | 0.73 | 0.80 | 0.01 |
| ID41 | 0.02 | 0.01 | 0.02 | 0.03 | 1.00 | 2.29 |
| ID42 | 0.87 | 0.05 | 0.84 | 0.80 | 1.00 | 0.01 |
| ID43 | 0.93 | 0.03 | 0.71 | 1.00 | 1.00 | 0.00 |
| ID44 | 0.72 | 0.21 | 0.00 | 0.00 | 0.00 | 0.01 |
| ID45 | 0.01 | 0.02 | 0.01 | 0.02 | 1.00 | 0.80 |
| ID46 | 0.37 | 0.04 | 0.33 | 0.41 | 0.52 | 0.16 |
| ID47 | 0.01 | 0.02 | 0.01 | 0.01 | 1.00 | 1.63 |
| ID48 | 0.88 | 0.01 | 0.73 | 1.00 | 1.00 | 0.00 |
| ID49 | 0.20 | 0.04 | 0.25 | 0.26 | 1.00 | 0.24 |
| ID50 | 0.99 | 0.03 | 0.89 | 1.00 | 1.00 | 0.00 |
| ID51 | 0.38 | 0.10 | 0.22 | 0.18 | 1.00 | 0.06 |
| ID52 | 0.09 | 0.03 | 0.18 | 0.06 | 1.00 | 0.57 |
| ID53 | 0.03 | 0.04 | 0.07 | 0.03 | 1.00 | 0.50 |
| ID54 | 0.00 | 0.01 | 0.00 | 0.02 | 1.00 | 1.85 |
| ID55 | 0.02 | 0.02 | 0.02 | 0.02 | 1.00 | 1.16 |
| ID56 | 0.47 | 0.07 | 0.43 | 0.36 | 0.67 | 0.05 |
| ID57 | 0.00 | 0.00 | 0.00 | 0.00 | 0.00 | 0.00 |
| ID58 | 0.07 | 0.06 | 0.07 | 0.07 | 0.14 | 0.15 |
| ID59 | 0.18 | 0.06 | 0.10 | 0.24 | 0.13 | 0.00 |
| ID60 | 0.32 | 0.02 | 0.18 | 0.33 | 0.38 | 0.14 |
| ID61 | 0.24 | 0.05 | 0.17 | 0.36 | 0.67 | 0.13 |
| ID62 | 0.78 | 0.05 | 0.60 | 0.93 | 1.00 | 0.00 |
| ID63 | 0.64 | 0.05 | 0.67 | 0.73 | 1.00 | 0.02 |
| ID64 | 0.56 | 0.02 | 0.52 | 0.62 | 0.60 | 0.19 |
| ID65 | 0.39 | 0.06 | 0.35 | 0.43 | 0.62 | 0.07 |
| ID66 | 0.79 | 0.13 | 0.38 | 0.50 | 0.50 | 0.00 |
| ID67 | 0.03 | 0.02 | 0.05 | 0.02 | 0.50 | 1.43 |
| ID68 | 0.52 | 0.11 | 0.14 | 0.25 | 0.33 | 0.02 |

Table 34: **Details of seizure detection results of MVPFormer-S with 50dB SNR.** Kappa is the inter-rater agreement. The classification metrics report the raw and episodic metrics relevant for the seizure classification task. The similarity reports the breakdown of the cosine similarity in each of the considered scenarios.

| | | | **Classification metrics** | | | |
| | | | Raw | Episodic | | |
| **Subject** | **Kappa** | 95% CI | f1-score | f1-score | sensitivity | fp/h |
|---|---|---|---|---|---|---|
| ID19 | 0.01 | 0.00 | 0.02 | 0.33 | 0.50 | 1.15 |
| ID20 | 0.76 | 0.10 | 0.82 | 0.33 | 1.00 | 0.01 |
| ID21 | 0.50 | 0.09 | 0.38 | 0.31 | 1.00 | 0.04 |
| ID22 | 0.97 | 0.03 | 0.89 | 1.00 | 1.00 | 0.00 |
| ID23 | 0.11 | 0.03 | 0.15 | 0.09 | 0.06 | 0.19 |
| ID24 | 0.91 | 0.01 | 0.87 | 0.93 | 0.93 | 0.02 |
| ID25 | 0.13 | 0.06 | 0.05 | 0.15 | 0.25 | 0.07 |
| ID26 | 0.03 | 0.04 | 0.02 | 0.05 | 1.00 | 0.47 |
| ID27 | 0.68 | 0.04 | 0.66 | 0.73 | 1.00 | 0.04 |
| ID28 | 0.86 | 0.03 | 0.87 | 0.86 | 0.75 | 0.00 |
| ID29 | 0.52 | 0.03 | 0.49 | 0.31 | 0.29 | 0.13 |
| ID30 | 0.74 | 0.01 | 0.73 | 0.76 | 0.78 | 0.17 |
| ID31 | 0.98 | 0.00 | 0.92 | 1.00 | 1.00 | 0.00 |
| ID32 | 0.87 | 0.09 | 0.76 | 0.67 | 1.00 | 0.01 |
| ID33 | 0.50 | 0.15 | 0.00 | 0.00 | 0.00 | 0.02 |
| ID34 | 0.79 | 0.03 | 0.85 | 0.75 | 1.00 | 0.03 |
| ID35 | 0.71 | 0.04 | 0.75 | 0.75 | 0.86 | 0.03 |
| ID36 | 0.46 | 0.03 | 0.58 | 0.43 | 0.30 | 0.04 |
| ID37 | 0.46 | 0.09 | 0.42 | 0.29 | 1.00 | 0.05 |
| ID38 | 0.90 | 0.03 | 0.95 | 0.91 | 1.00 | 0.01 |
| ID39 | 0.99 | 0.03 | 0.89 | 1.00 | 1.00 | 0.00 |
| ID40 | 0.82 | 0.04 | 0.88 | 0.80 | 0.80 | 0.00 |
| ID41 | 0.23 | 0.04 | 0.29 | 0.22 | 1.00 | 0.25 |
| ID42 | 0.99 | 0.02 | 0.92 | 1.00 | 1.00 | 0.00 |
| ID43 | 0.93 | 0.03 | 0.71 | 1.00 | 1.00 | 0.00 |
| ID44 | 0.89 | 0.15 | 0.32 | 0.67 | 0.50 | 0.00 |
| ID45 | 0.01 | 0.02 | 0.00 | 0.01 | 1.00 | 0.99 |
| ID46 | 0.40 | 0.04 | 0.42 | 0.45 | 0.33 | 0.02 |
| ID47 | 0.04 | 0.04 | 0.07 | 0.05 | 1.00 | 0.38 |
| ID48 | 0.86 | 0.01 | 0.70 | 1.00 | 1.00 | 0.00 |
| ID49 | 0.44 | 0.05 | 0.49 | 0.55 | 1.00 | 0.07 |
| ID50 | 0.98 | 0.05 | 0.81 | 1.00 | 1.00 | 0.00 |
| ID51 | 0.73 | 0.12 | 0.55 | 0.40 | 1.00 | 0.02 |
| ID52 | 0.26 | 0.05 | 0.42 | 0.19 | 1.00 | 0.15 |
| ID53 | 0.26 | 0.08 | 0.30 | 0.14 | 1.00 | 0.09 |
| ID54 | 0.01 | 0.01 | 0.01 | 0.02 | 1.00 | 1.41 |
| ID55 | 0.12 | 0.06 | 0.12 | 0.12 | 1.00 | 0.20 |
| ID56 | 0.72 | 0.07 | 0.54 | 0.67 | 0.67 | 0.01 |
| ID57 | 0.00 | 0.00 | 0.00 | 0.00 | 0.00 | 0.00 |
| ID58 | 0.33 | 0.12 | 0.00 | 0.00 | 0.00 | 0.01 |
| ID59 | 0.11 | 0.06 | 0.08 | 0.12 | 0.07 | 0.00 |
| ID60 | 0.23 | 0.03 | 0.15 | 0.50 | 0.38 | 0.02 |
| ID61 | 0.41 | 0.08 | 0.08 | 0.29 | 0.17 | 0.00 |
| ID62 | 0.82 | 0.04 | 0.64 | 1.00 | 1.00 | 0.00 |
| ID63 | 0.79 | 0.04 | 0.75 | 0.89 | 1.00 | 0.01 |
| ID64 | 0.62 | 0.02 | 0.66 | 0.65 | 0.50 | 0.03 |
| ID65 | 0.63 | 0.06 | 0.43 | 0.67 | 0.62 | 0.01 |
| ID66 | 0.84 | 0.16 | 0.38 | 0.50 | 0.50 | 0.00 |
| ID67 | 0.12 | 0.04 | 0.23 | 0.07 | 0.25 | 0.22 |
| ID68 | 0.65 | 0.13 | 0.22 | 0.33 | 0.33 | 0.01 |

Table 35: **Details of seizure detection results of MVPFormer-S with 60dB SNR.** Kappa is the inter-rater agreement. The classification metrics report the raw and episodic metrics relevant for the seizure classification task. The similarity reports the breakdown of the cosine similarity in each of the considered scenarios.

| | | | **Classification metrics** | | | |
| | | | Raw | Episodic | | |
| **Subject** | **Kappa** | 95% CI | f1-score | f1-score | sensitivity | fp/h |
|---|---|---|---|---|---|---|
| ID19 | 0.01 | 0.00 | 0.02 | 0.33 | 0.50 | 1.15 |
| ID20 | 0.87 | 0.06 | 0.91 | 0.57 | 1.00 | 0.01 |
| ID21 | 0.74 | 0.11 | 0.49 | 0.50 | 1.00 | 0.02 |
| ID22 | 0.97 | 0.02 | 0.91 | 1.00 | 1.00 | 0.00 |
| ID23 | 0.10 | 0.03 | 0.17 | 0.10 | 0.06 | 0.09 |
| ID24 | 0.94 | 0.01 | 0.90 | 0.96 | 0.93 | 0.00 |
| ID25 | 0.18 | 0.09 | 0.00 | 0.00 | 0.00 | 0.06 |
| ID26 | 0.03 | 0.03 | 0.02 | 0.04 | 1.00 | 0.62 |
| ID27 | 0.85 | 0.03 | 0.78 | 0.89 | 1.00 | 0.01 |
| ID28 | 0.81 | 0.04 | 0.85 | 0.86 | 0.75 | 0.00 |
| ID29 | 0.44 | 0.03 | 0.44 | 0.28 | 0.29 | 0.17 |
| ID30 | 0.68 | 0.01 | 0.68 | 0.70 | 0.70 | 0.20 |
| ID31 | 0.98 | 0.00 | 0.93 | 1.00 | 1.00 | 0.00 |
| ID32 | 1.00 | 0.02 | 0.93 | 1.00 | 1.00 | 0.00 |
| ID33 | 0.51 | 0.15 | 0.00 | 0.00 | 0.00 | 0.02 |
| ID34 | 0.76 | 0.03 | 0.87 | 0.82 | 1.00 | 0.02 |
| ID35 | 0.74 | 0.04 | 0.79 | 0.80 | 0.86 | 0.02 |
| ID36 | 0.43 | 0.03 | 0.52 | 0.40 | 0.28 | 0.06 |
| ID37 | 0.61 | 0.08 | 0.54 | 0.40 | 1.00 | 0.03 |
| ID38 | 0.92 | 0.02 | 0.95 | 0.91 | 1.00 | 0.01 |
| ID39 | 0.98 | 0.03 | 0.89 | 1.00 | 1.00 | 0.00 |
| ID40 | 0.88 | 0.04 | 0.91 | 0.89 | 0.80 | 0.00 |
| ID41 | 0.29 | 0.04 | 0.37 | 0.29 | 1.00 | 0.18 |
| ID42 | 0.98 | 0.02 | 0.87 | 1.00 | 1.00 | 0.00 |
| ID43 | 0.89 | 0.04 | 0.56 | 1.00 | 1.00 | 0.00 |
| ID44 | 0.88 | 0.11 | 0.32 | 0.67 | 0.50 | 0.00 |
| ID45 | 0.01 | 0.02 | 0.00 | 0.01 | 1.00 | 0.94 |
| ID46 | 0.40 | 0.04 | 0.43 | 0.47 | 0.33 | 0.01 |
| ID47 | 0.04 | 0.05 | 0.08 | 0.06 | 1.00 | 0.31 |
| ID48 | 0.87 | 0.01 | 0.70 | 1.00 | 1.00 | 0.00 |
| ID49 | 0.54 | 0.05 | 0.52 | 0.63 | 1.00 | 0.05 |
| ID50 | 0.96 | 0.05 | 0.81 | 1.00 | 1.00 | 0.00 |
| ID51 | 0.83 | 0.13 | 0.67 | 0.50 | 1.00 | 0.01 |
| ID52 | 0.35 | 0.05 | 0.52 | 0.25 | 1.00 | 0.11 |
| ID53 | 0.46 | 0.09 | 0.40 | 0.22 | 1.00 | 0.05 |
| ID54 | 0.01 | 0.01 | 0.01 | 0.01 | 0.67 | 1.49 |
| ID55 | 0.46 | 0.09 | 0.35 | 0.36 | 1.00 | 0.05 |
| ID56 | 0.72 | 0.07 | 0.52 | 0.67 | 0.67 | 0.01 |
| ID57 | 0.00 | 0.00 | 0.00 | 0.00 | 0.00 | 0.00 |
| ID58 | 0.30 | 0.12 | 0.00 | 0.00 | 0.00 | 0.01 |
| ID59 | 0.20 | 0.06 | 0.13 | 0.24 | 0.13 | 0.00 |
| ID60 | 0.26 | 0.03 | 0.15 | 0.50 | 0.38 | 0.02 |
| ID61 | 0.35 | 0.09 | 0.08 | 0.29 | 0.17 | 0.00 |
| ID62 | 0.84 | 0.04 | 0.67 | 1.00 | 1.00 | 0.00 |
| ID63 | 0.70 | 0.05 | 0.67 | 0.80 | 1.00 | 0.01 |
| ID64 | 0.65 | 0.02 | 0.67 | 0.67 | 0.50 | 0.00 |
| ID65 | 0.70 | 0.06 | 0.51 | 0.77 | 0.62 | 0.00 |
| ID66 | 0.88 | 0.09 | 0.53 | 0.80 | 1.00 | 0.00 |
| ID67 | 0.15 | 0.05 | 0.24 | 0.10 | 0.25 | 0.14 |
| ID68 | 0.76 | 0.13 | 0.26 | 0.40 | 0.33 | 0.01 |

### G.15 EVALUATION OF MAXIMUM PERFORMANCE

To better characterize MVPFormer's ability to generalize to unseen subjects, we perform the seizure detection task on 40 subjects in two different scenarios. First, we use a model that is pre-trained on those 40 subjects (see Figure 21 and Table 36) to determine MVPFormer's maximum performance on the manual selection of channels. Second, we use a model for which those 40 subjects are unseen (see Figure 22 and Table 37). As expected, with a Kappa score of 0.73 the model trained on the testing subjects achieves superior agreement even to human experts, and can therefore be seen as having learned the training set. On the other hand, as seen with previous results as well, in the unseen subject scenario MVPFormer reaches a Kappa score of 0.56, indicating a high degree of generalization.

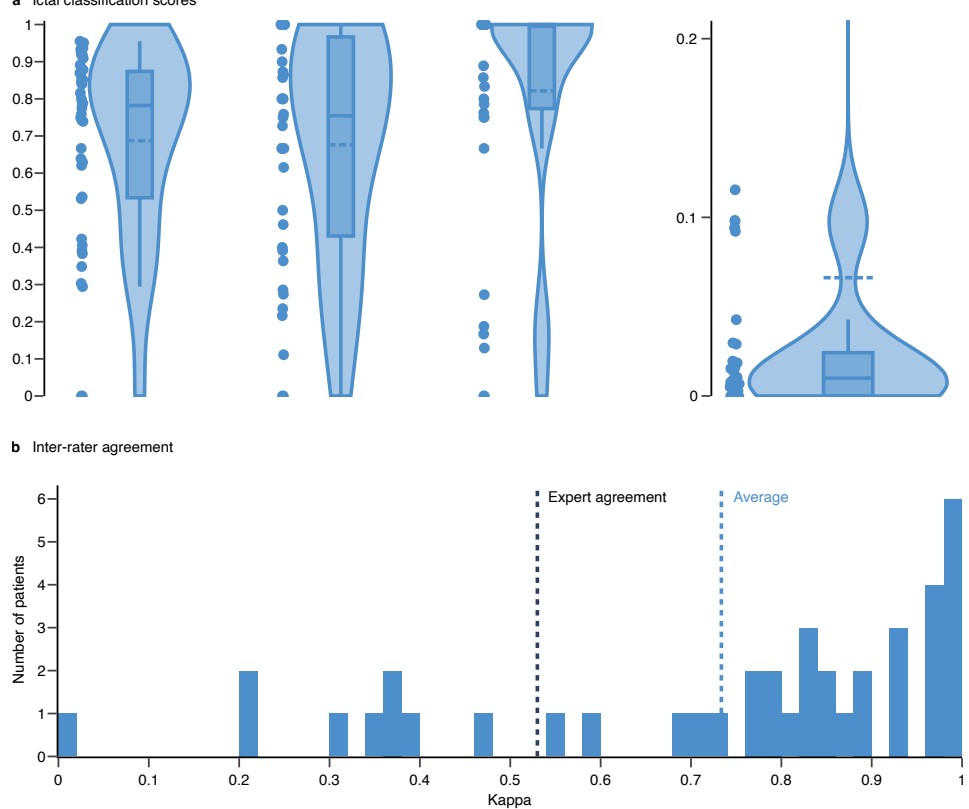

Figure 21: **Seizure detection with 58 subject pre-training and evaluation on 40 pre-trained subjects on the manual selection of channels. (a)** Seizure detection results of MVPFormer-S on 40 previously seen subjects using the manual selection of channels: the F1-score, sensitivity, and fp/h are reported. The performance metrics are notably improved due to testing on previously seen subjects. **(b)** Cohen's kappa is used to measure the agreement between the artificial assistant and the human expert. The average kappa is 0.73, notably improved from the baseline.

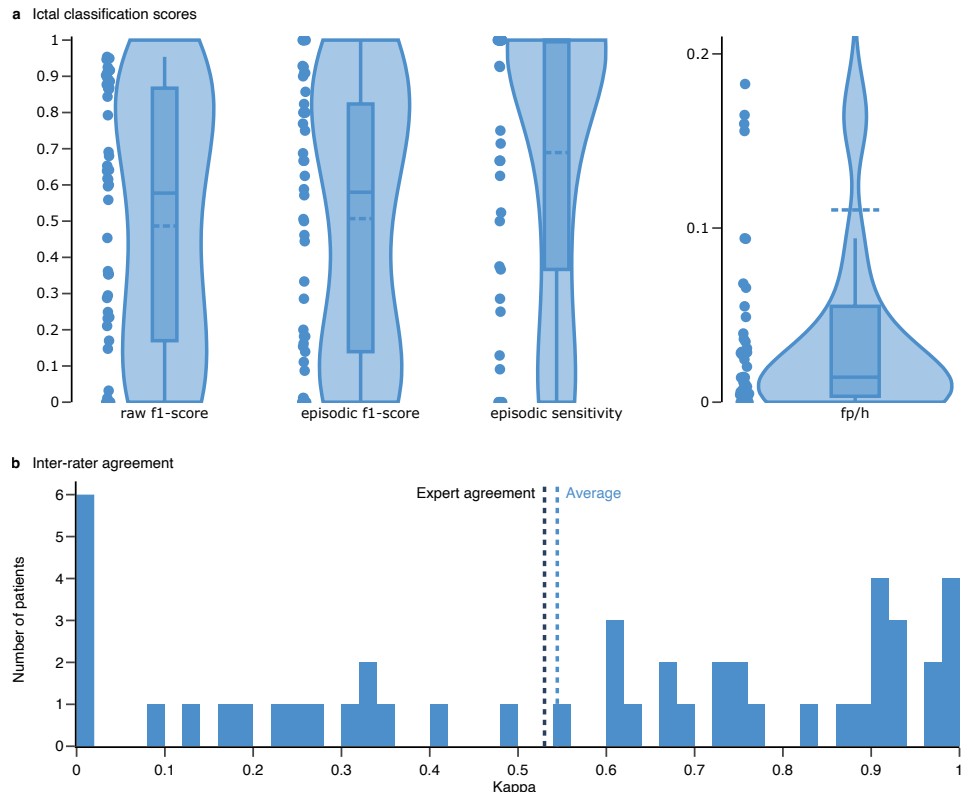

Figure 22: **Seizure detection with 18 subjects pre-training on a selection of 40 unseen subjects and the manual selection of channels. (a)** Seizure detection results of MVPFormer-S on 40 unseen subjects which are part of the training set for the 58-subjects model using the manual selection of channels: the F1-score, sensitivity, and fp/h are reported. The raw and episodic F1-scores are notably lower here with respect to the 58-subjects model. This is expected given the 58-subject model is pre-trained on these subjects. These results are a subset of those presented in the Results section. **(b)** Cohen's kappa is used to measure the agreement between the artificial assistant and the human expert. The average kappa is 0.56, competitive with expert agreement but, as expected, reduced from the 58-subjects pre-trained model.

Table 36: **Details of seizure detection results of MVPFormer-S with 58 subject pre-training and evaluation on 40 pre-trained subjects using the manual selection of channels.** Kappa is the inter-rater agreement. The classification metrics report the raw and episodic metrics relevant for the seizure classification task.

| | | | Classification metrics | | | |
| | | | Raw | Episodic | | |
| Subject | Kappa | 95% CI | f1-score | f1-score | sensitivity | fp/h |
|---|---|---|---|---|---|---|
| ID19 | 0.36 | 0.02 | 0.29 | 0.29 | 0.19 | 0.09 |
| ID20 | 0.92 | 0.05 | 0.92 | 0.67 | 1.00 | 0.01 |
| ID21 | 0.75 | 0.12 | 0.62 | 0.50 | 1.00 | 0.02 |
| ID22 | 0.77 | 0.06 | 0.78 | 0.73 | 1.00 | 0.02 |
| ID23 | 0.36 | 0.02 | 0.39 | 0.39 | 0.27 | 0.09 |
| ID24 | 0.69 | 0.02 | 0.64 | 0.76 | 0.79 | 0.10 |
| ID25 | 0.43 | 0.15 | 0.00 | 0.00 | 0.00 | 0.01 |
| ID26 | 0.98 | 0.04 | 0.80 | 1.00 | 1.00 | 0.00 |
| ID27 | 0.80 | 0.04 | 0.74 | 0.80 | 0.75 | 0.01 |
| ID28 | 0.62 | 0.05 | 0.54 | 0.67 | 0.75 | 0.03 |
| ID29 | 0.24 | 0.05 | 0.30 | 0.22 | 0.13 | 0.01 |
| ID30 | 0.85 | 0.01 | 0.79 | 0.87 | 0.89 | 0.10 |
| ID31 | 0.97 | 0.00 | 0.94 | 1.00 | 1.00 | 0.00 |
| ID32 | 0.99 | 0.03 | 0.92 | 1.00 | 1.00 | 0.00 |
| ID33 | 0.00 | 0.00 | 0.00 | 0.00 | 0.00 | 0.00 |
| ID34 | 0.88 | 0.03 | 0.88 | 0.90 | 1.00 | 0.01 |
| ID35 | 0.92 | 0.02 | 0.89 | 0.93 | 1.00 | 0.01 |
| ID36 | 0.26 | 0.04 | 0.42 | 0.27 | 0.17 | 0.02 |
| ID37 | 0.93 | 0.07 | 0.85 | 0.80 | 1.00 | 0.01 |
| ID38 | 0.98 | 0.01 | 0.95 | 1.00 | 1.00 | 0.00 |
| ID39 | 0.87 | 0.06 | 0.84 | 0.80 | 1.00 | 0.01 |
| ID40 | 0.77 | 0.05 | 0.82 | 0.62 | 0.80 | 0.02 |
| ID41 | 0.99 | 0.02 | 0.94 | 1.00 | 1.00 | 0.00 |
| ID42 | 0.99 | 0.01 | 0.95 | 1.00 | 1.00 | 0.00 |
| ID43 | 0.83 | 0.05 | 0.79 | 0.80 | 1.00 | 0.02 |
| ID44 | 0.98 | 0.02 | 0.87 | 1.00 | 1.00 | 0.00 |
| ID45 | 0.91 | 0.13 | 0.67 | 0.67 | 1.00 | 0.01 |
| ID46 | 0.80 | 0.02 | 0.76 | 0.86 | 0.76 | 0.00 |
| ID47 | 0.93 | 0.07 | 0.84 | 0.86 | 1.00 | 0.00 |
| ID48 | 0.97 | 0.01 | 0.93 | 1.00 | 1.00 | 0.00 |
| ID49 | 0.67 | 0.05 | 0.74 | 0.67 | 1.00 | 0.04 |
| ID50 | 0.81 | 0.08 | 0.75 | 0.67 | 1.00 | 0.01 |
| ID51 | 0.71 | 0.12 | 0.53 | 0.40 | 1.00 | 0.02 |
| ID52 | 0.36 | 0.05 | 0.63 | 0.24 | 1.00 | 0.12 |
| ID53 | 0.99 | 0.04 | 0.91 | 1.00 | 1.00 | 0.00 |
| ID54 | 0.55 | 0.09 | 0.35 | 0.36 | 0.67 | 0.03 |
| ID55 | 0.98 | 0.03 | 0.85 | 1.00 | 1.00 | 0.00 |
| ID56 | 0.85 | 0.05 | 0.87 | 0.75 | 1.00 | 0.02 |
| ID57 | 0.20 | 0.01 | 0.38 | 0.11 | 0.83 | 1.74 |
| ID58 | 0.40 | 0.04 | 0.41 | 0.46 | 0.86 | 0.09 |
| ID59 | 0.00 | 0.00 | 0.00 | 0.00 | 0.00 | 0.00 |
| ID60 | 0.16 | 0.03 | 0.08 | 0.36 | 0.25 | 0.02 |
| ID61 | 0.00 | 0.00 | 0.00 | 0.00 | 0.00 | 0.00 |
| ID62 | 0.42 | 0.06 | 0.44 | 0.41 | 0.86 | 0.07 |
| ID63 | 0.54 | 0.09 | 0.25 | 0.57 | 0.50 | 0.01 |
| ID64 | 0.62 | 0.02 | 0.59 | 0.65 | 0.50 | 0.03 |
| ID65 | 0.31 | 0.04 | 0.24 | 0.39 | 1.00 | 0.18 |
| ID66 | 0.69 | 0.11 | 0.52 | 0.50 | 1.00 | 0.02 |
| ID67 | 0.00 | 0.06 | 0.00 | 0.00 | 0.00 | 0.13 |
| ID68 | 0.78 | 0.15 | 0.33 | 0.50 | 0.33 | 0.00 |

Table 37: **Details of seizure detection results of MVPFormer-S with 18 subjects pre-training on a selection of 40 unseen subjects using the manual selection of channels.** Kappa is the inter-rater agreement. The classification metrics report the raw and episodic metrics relevant for the seizure classification task.

| | | | **Classification metrics** | | | |
| | | | Raw | Episodic | | |
| **Subject** | **Kappa** | 95% CI | f1-score | f1-score | sensitivity | fp/h |
| ID19 | -0.05 | 0.02 | 0.00 | 0.00 | 0.00 | 0.97 |
| ID20 | 0.98 | 0.03 | 0.87 | 0.80 | 1.00 | 0.00 |
| ID21 | 0.92 | 0.07 | 0.86 | 0.80 | 1.00 | 0.00 |
| ID22 | 0.98 | 0.02 | 0.89 | 1.00 | 1.00 | 0.00 |
| ID23 | 0.26 | 0.03 | 0.17 | 0.14 | 0.09 | 0.16 |
| ID24 | 0.91 | 0.01 | 0.90 | 0.93 | 0.93 | 0.02 |
| ID25 | 0.35 | 0.12 | 0.00 | 0.00 | 0.00 | 0.03 |
| ID26 | 0.08 | 0.06 | 0.03 | 0.11 | 1.00 | 0.18 |
| ID27 | 0.99 | 0.01 | 0.93 | 1.00 | 1.00 | 0.00 |
| ID28 | 0.71 | 0.04 | 0.68 | 0.75 | 0.75 | 0.01 |
| ID29 | 0.25 | 0.05 | 0.29 | 0.20 | 0.13 | 0.03 |
| ID30 | 0.92 | 0.01 | 0.88 | 0.93 | 0.93 | 0.05 |
| ID31 | 0.98 | 0.00 | 0.95 | 1.00 | 1.00 | 0.00 |
| ID32 | 1.00 | 0.02 | 0.92 | 1.00 | 1.00 | 0.00 |
| ID33 | 0.00 | 0.05 | 0.00 | 0.00 | 0.00 | 0.26 |
| ID34 | 0.89 | 0.03 | 0.91 | 0.90 | 1.00 | 0.01 |
| ID35 | 0.85 | 0.03 | 0.87 | 0.82 | 1.00 | 0.03 |
| ID36 | 0.50 | 0.03 | 0.62 | 0.51 | 0.37 | 0.03 |
| ID37 | 0.74 | 0.09 | 0.69 | 0.50 | 1.00 | 0.02 |
| ID38 | 0.94 | 0.02 | 0.95 | 0.91 | 1.00 | 0.01 |
| ID39 | 0.92 | 0.05 | 0.89 | 0.80 | 1.00 | 0.01 |
| ID40 | 0.95 | 0.03 | 0.95 | 0.91 | 1.00 | 0.00 |
| ID41 | 0.61 | 0.05 | 0.65 | 0.67 | 1.00 | 0.04 |
| ID42 | 0.86 | 0.05 | 0.84 | 0.80 | 1.00 | 0.01 |
| ID43 | 0.90 | 0.03 | 0.64 | 1.00 | 1.00 | 0.00 |
| ID44 | 0.00 | 0.00 | 0.00 | 0.00 | 0.00 | 0.00 |
| ID45 | 0.01 | 0.02 | 0.00 | 0.01 | 1.00 | 1.06 |
| ID46 | 0.60 | 0.04 | 0.56 | 0.69 | 0.52 | 0.00 |
| ID47 | 0.26 | 0.08 | 0.29 | 0.16 | 1.00 | 0.09 |
| ID48 | 0.68 | 0.02 | 0.62 | 0.80 | 0.67 | 0.00 |
| ID49 | 0.85 | 0.03 | 0.79 | 0.86 | 1.00 | 0.01 |
| ID50 | 0.69 | 0.09 | 0.60 | 0.44 | 1.00 | 0.03 |
| ID51 | 0.35 | 0.09 | 0.23 | 0.15 | 1.00 | 0.07 |
| ID52 | 0.23 | 0.05 | 0.35 | 0.18 | 1.00 | 0.16 |
| ID53 | 0.12 | 0.06 | 0.21 | 0.09 | 1.00 | 0.16 |
| ID54 | 0.01 | 0.02 | 0.01 | 0.01 | 0.67 | 1.72 |
| ID55 | 0.37 | 0.08 | 0.35 | 0.29 | 1.00 | 0.07 |
| ID56 | 0.72 | 0.08 | 0.60 | 0.67 | 0.67 | 0.01 |
| ID57 | 0.00 | 0.00 | 0.00 | 0.00 | 0.00 | 0.00 |
| ID58 | 0.22 | 0.07 | 0.25 | 0.18 | 0.29 | 0.09 |

G.16 EFFECTS OF THE NUMBER OF CHANNELS ON THE BRAIN TREEBANK DATASET

To better evaluate the robustness of MVPFormer to an increasing number of channels, we evaluate the four tasks of the BrainTreeBank with a range of 10 to 50 channels. The performance of MVP-Former moderately increases with no reduction with the channel number, as reported in PopT (Chau et al., 2025) as well, indicating that our model is robust to the number of channels.

Table 38: **Effects of the number of channels on the four tasks of the Brain TreeBank dataset.** Evaluation of the performance of MVPFormer with respect to number of channels for fine-tuning and testing.

| Channels | Pitch | Volume | Onset | Speech |
|---|---|---|---|---|
| 10 | 0.81 (0.01) | 0.85 (0.01) | 0.86 (0.02) | 0.87 (0.02) |
| 20 | 0.82 (0.01) | 0.87 (0.01) | 0.87 (0.02) | 0.88 (0.02) |
| 30 | 0.82 (0.02) | 0.87 (0.01) | 0.87 (0.02) | 0.89 (0.02) |
| 40 | 0.83 (0.01) | 0.87 (0.01) | 0.87 (0.02) | 0.89 (0.02) |
| 50 | 0.83 (0.01) | 0.88 (0.01) | 0.87 (0.02) | 0.90 (0.02) |

## G.17 EVALUATION ON TRADITIONAL LONG-TERM FORECASTING TASK

To provide a comprehensive evaluation of MVPA, we compare MVPFormer with existing SOTA architectures on a classical long-term forecasting task. Table 39 reports the results of MVPFormer, the vanilla Transformer (Vaswani et al., 2017), PatchTST (Nie et al., 2023), TimesFM (Das et al., 2024), TimeMixer (Wang et al., 2024b), and WPMixer (Chau et al., 2025) on the ETTh1, ETTh2, and Weather datasets (Zhou et al., 2021; Wu et al., 2021). These datasets represent a well-known benchmark that allows us to decouple MVPFormer from the specific clinical setting. The lookback window is fixed to 96, while the forecast is performed at lengths of 96, 192, 336, and 720. These settings are well established in the literature (Wang et al., 2024b). MVPFormer notably surpasses the vanilla Transformer and is competitive with established architectures designed specifically for this task, achieving the best or second best result in most cases. Moreover, MVPFormer is on average 2x faster to train than the vanilla Transformer and 1.4x faster than PatchTST — still slower than TimeMixer, which a fully MLP-based model —, making it an excellent choice in many scenarios. Therefore, we have shown that MVPFormer and MVPA have a wide applicability and transfer their performance from the clinical task — for which they were designed — to more general time-series tasks as well.

Table 39: **Classical time-series forecasting benchmark. MVPFormer is compared with multiple SOTA architectures on the time-series forecasting task using the ETTh1, ETTh2, and Weather datasets.** The lookback window is fixed at 96 and the forecasting length varies between 96 to 720. The vanilla Transformer is also included as a point of comparison. In bold are the best MSE results, in italics are the second best. MVPFormer notably outperforms the vanilla Transformer and is competitive with all baselines, having either the best or second best result in most cases.

| Model | | MVPFormer (ours) | | Transformer | | PatchTST | | TimesFM | | TimeMixer | | WPMixer | |
|---|---|---|---|---|---|---|---|---|---|---|---|---|---|
| Metric | Length | MSE | MAE | MSE | MAE | MSE | MAE | MSE | MAE | MSE | MAE | MSE | MAE |
| ETTh1 | 96 | *0.38* | 0.40 | 0.83 | 0.72 | ***0.38*** | 0.40 | 0.39 | 0.41 | **0.37** | 0.40 | 0.39 | 0.40 |
| | 192 | 0.45 | 0.44 | 0.96 | 0.78 | **0.43** | 0.43 | 0.46 | 0.44 | *0.44* | 0.43 | **0.43** | 0.42 |
| | 336 | *0.49* | 0.46 | 1.04 | 0.83 | **0.47** | 0.46 | *0.49* | 0.45 | 0.50 | 0.46 | 0.49 | 0.45 |
| | 720 | **0.49** | 0.48 | 1.16 | 0.86 | 0.52 | 0.51 | *0.50* | 0.48 | **0.49** | 0.48 | *0.49* | 0.47 |
| ETTh2 | 96 | *0.30* | 0.35 | 2.64 | 1.30 | 0.31 | 0.35 | *0.30* | 0.45 | **0.29** | 0.35 | *0.30* | 0.35 |
| | 192 | **0.37** | 0.40 | 3.48 | 1.47 | *0.38* | 0.40 | 0.37 | 0.40 | **0.37** | 0.39 | **0.37** | 0.40 |
| | 336 | **0.42** | 0.43 | 4.07 | 1.62 | *0.43* | 0.44 | 0.43 | 0.44 | *0.43* | 0.44 | **0.42** | 0.43 |
| | 720 | **0.43** | 0.43 | 3.28 | 1.52 | *0.45* | 0.45 | 0.44 | 0.45 | 0.47 | 0.47 | *0.45* | 0.46 |
| Weather | 96 | *0.17* | 0.22 | 0.33 | 0.38 | *0.17* | 0.22 | *0.17* | 0.21 | **0.16** | 0.21 | *0.17* | 0.21 |
| | 192 | **0.21** | 0.26 | 0.51 | 0.50 | *0.22* | 0.26 | *0.22* | 0.26 | **0.21** | 0.25 | *0.22* | 0.25 |
| | 336 | *0.28* | 0.30 | 0.62 | 0.56 | *0.28* | 0.30 | *0.28* | 0.30 | **0.26** | 0.29 | **0.26** | 0.30 |
| | 720 | **0.35** | 0.35 | 0.91 | 0.70 | *0.36* | 0.35 | **0.35** | 0.35 | **0.35** | 0.35 | **0.35** | 0.35 |

### G.18 ABLATION OF THE THREE COMPONENTS ON TRADITIONAL LONG-TERM FORECASTING TASK

MVPA is composed of three components (content, time, and channel attention) which process different aspects of the time-series in parallel. We evaluate the impact of the components on the long-term forecasting tasks as above. Table 40 reports the results of full MVPA (all three components), content-only attention, time-only attention, channel-only attention, and no attention on the ETTh1, ETTh2, and Weather datasets (Zhou et al., 2021; Wu et al., 2021).

Full MVPA is consistently the better performer, except on the 96 and 720 lengths of the ETTh2 dataset. Notably, MVPA outperforms all variants by a greater margin on the Weather dataset, which has the most number of channels (21 vs 7 of ETTh1 and ETTh2). This suggests that strongly multivariate time-series provide a considerable advantage to MVPA, which is consistent with its design.

Table 40: **Ablation of the three components on the classical time-series forecasting benchmark. MVPA is compared with ablated variants of its three components on the time-series forecasting task using the ETTh1, ETTh2, and Weather datasets.** The lookback window is fixed at 96 and the forecasting length varies between 96 to 720. The vanilla Transformer is also included as a point of comparison. In bold are the best MSE results.

| Model | | **MVPA** | | Content-only | | Time-only | | Channel-only | | None | |
|---|---|---|---|---|---|---|---|---|---|---|---|
| Metric | Length | MSE | MAE | MSE | MAE | MSE | MAE | MSE | MAE | MSE | MAE |
| ETTh1 | 96 | **0.38** | 0.40 | 0.39 | 0.40 | 0.39 | 0.40 | 0.39 | 0.40 | 0.41 | 0.40 |
| | 192 | **0.45** | 0.44 | **0.45** | 0.44 | **0.45** | 0.44 | **0.45** | 0.45 | 0.46 | 0.45 |
| | 336 | **0.49** | 0.46 | **0.49** | 0.46 | **0.49** | 0.46 | **0.49** | 0.46 | 0.50 | 0.49 |
| | 720 | **0.49** | 0.48 | **0.49** | 0.48 | **0.49** | 0.48 | **0.49** | 0.48 | **0.49** | 0.48 |
| ETTh2 | 96 | 0.30 | 0.35 | 0.31 | 0.35 | **0.29** | 0.35 | 0.30 | 0.35 | 0.30 | 0.35 |
| | 192 | **0.37** | 0.40 | **0.37** | 0.39 | 0.38 | 0.40 | **0.37** | 0.40 | 0.38 | 0.39 |
| | 336 | **0.42** | 0.43 | 0.43 | 0.43 | **0.42** | 0.43 | **0.42** | 0.43 | 0.43 | 0.43 |
| | 720 | 0.43 | 0.45 | **0.42** | 0.44 | 0.43 | 0.44 | **0.42** | 0.44 | 0.47 | 0.47 |
| Weather | 96 | **0.17** | 0.22 | 0.19 | 0.23 | 0.19 | 0.23 | 0.18 | 0.22 | 0.19 | 0.23 |
| | 192 | **0.21** | 0.26 | 0.23 | 0.26 | 0.23 | 0.26 | 0.22 | 0.26 | 0.23 | 0.26 |
| | 336 | **0.28** | 0.30 | 0.29 | 0.30 | 0.29 | 0.30 | 0.28 | 0.30 | 0.29 | 0.30 |
| | 720 | **0.35** | 0.35 | 0.36 | 0.35 | 0.36 | 0.35 | **0.35** | 0.35 | 0.36 | 0.35 |

