# OpenReview forum: "A foundation model with multi-variate parallel attention to generate neuronal activity"
_ICLR.cc/2026/Conference — ICLR 2026 Poster_

### Official Review · Reviewer_dxZg · 2025-10-31

**Soundness:** 3
**Presentation:** 3
**Contribution:** 3
**Rating:** 6
**Confidence:** 3

**Summary:**

This paper introduces Multi-Variate Parallel Attention (MVPA), a novel self-attention mechanism that addresses the challenges of multi-variate time-series data with heterogeneous channel configurations. The MVPA mechanism is designed to handle time-series signals that vary across different subjects, particularly in clinical domains like intracranial electroencephalography (iEEG). The model efficiently separates attention into three components: content-based, time-based, and channel-based attention, enabling flexible processing of data without relying on fixed channel positions or global positional encodings. The authors apply MVPA to develop MVPFormer, a foundation model for human electrophysiology, which is trained on the Long-term iEEG dataset, the largest publicly available iEEG corpus. MVPFormer achieves superior generalization across subjects and outperforms state-of-the-art (SOTA) models in several clinical tasks such as seizure detection, while also excelling in general time-series forecasting and classification tasks.

**Strengths:**

1. The introduction of the Multi-Variate Parallel Attention (MVPA) mechanism is innovative and addresses the challenge of heterogeneous channel configurations in multi-variate time-series data. The way MVPA separates content, temporal, and spatial attention is novel and can be generalized to other time-series domains beyond iEEG.

2. The MVPFormer model, powered by MVPA, shows impressive performance in several iEEG-related tasks, including seizure detection, outperforming existing models. The model demonstrates expert-level performance on the Long-term iEEG dataset and outperforms SOTA methods across various clinical benchmarks.

3. The paper releases the Long-term iEEG dataset, the largest publicly available iEEG dataset to date, containing nearly 10,000 hours of recordings. This is a significant contribution to the research community, addressing the issue of data scarcity in iEEG research.

4. The commitment to open-source the dataset, code, and weights is a major advantage for reproducibility and allows other researchers to build upon this work.

5. The use of MVPA and MVPFormer is not limited to iEEG but is shown to generalize well to classical time-series forecasting and classification tasks, offering a broader impact in the field of time-series modeling.

**Weaknesses:**

1. While the results on seizure detection are impressive, the paper lacks a detailed real-world scenario evaluation, particularly regarding how the model would perform in a clinical setting with real-time data or noisy recordings. The authors should consider adding practical deployment considerations and edge-case performance, such as handling low-quality signals or data interruptions.

2. MVPA introduces a level of computational complexity, especially in the time- and channel-based terms. While the paper provides solutions to mitigate this, it would benefit from more in-depth comparisons with simpler models that might achieve similar performance with less computational overhead.

3. Although MVPFormer performs well in seizure detection and on Brain TreeBank tasks, the paper could benefit from broader evaluation across more clinical tasks (e.g., epilepsy classification or other cognitive tasks), especially those commonly encountered in clinical settings.

4.While the Long-term iEEG dataset is a valuable contribution, it has limitations, such as lack of electrode location information, which may limit its utility in some clinical contexts. The paper mentions this but does not fully address how future versions of the dataset might overcome this limitation.

5. While MVPFormer outperforms existing methods like Brant-2, BrainBERT, and others, a more detailed analysis of how these models compare in terms of generalization across different subjects and datasets would strengthen the argument for MVPA’s superiority. Specific examples of failure modes in the comparison would be helpful.

**Questions:**

1. How does MVPA perform in scenarios with significantly different time series compared to iEEG, particularly in domains like financial time series or sensor data? Could the model’s flexibility be leveraged for other domains?

2. While the results show strong generalization across subjects, could the model handle extremely varied electrode setups (e.g., patients with unusual electrode configurations)? How does MVPA cope with potential signal distortions caused by non-standard setups?

3. With the Long-term iEEG dataset being very large (10,000 hours), what are the limitations in terms of processing and inference time when scaling to even larger datasets, especially in real-time clinical settings?

4. Could MVPFormer's real-world clinical application be affected by variability in electrode placement (e.g., anatomical differences across patients)? Have you tested MVPFormer on data from patients who have had non-standard electrode placements due to medical conditions?

5.While MVPFormer is open-source, how are you ensuring patient privacy and safety when providing access to the dataset and model weights? Are there any restrictions on data sharing due to privacy concerns that might hinder the broader clinical adoption of this model?

---

> ### Author Response · Authors · 2025-11-21
>
> We thank the Reviewer for highlighting the impressive results of MVPFormer, and for strongly appreciating our commitment to open science.
>
> As suggested by the Reviewer, we now provide further benchmarks on noisier version of the Long-term iEEG dataset to better characterize the resilience of MVPFormer to noise. We also report runtime and memory comparisons against more attention variants to identify the key advantages of MVPA. Moreover, we clarify the unclear aspects of the Long-term iEEG dataset and address all questions raised by the Reviewer.
>
> ---
>
> ## W1. Low-quality signals
>
> MVPFormer is designed with clinical relevance as a primary objective, as testified by the curation of the largest clinical iEEG dataset yet. This dataset was collected in a hospital environment and was subjected to the same pre-processing as would be used by epileptologists during routine clinical evaluations. Nonetheless, as part of our extensive evaluation suite, we test multiple datasets (Long-term iEEG, MAYO, FNUSA, Brain TreeBank) with varying levels of noise. In particular, while MVPFormer surpasses all baselines on the high-SNR Long-term iEEG dataset, it is also state-of-the-art (SOTA) on the lower-SNR MAYO and FNUSA, albeit with a smaller margin.
>
> Following the Reviewer's suggestion, we now test MVPFormer on the Long-term iEEG dataset while **varying the amount of noise, from 30dB to 60dB SNR**. We report the results in Appendix G.14 Resilience to noise and provide a summary here for discussion.
>
> |          | Episodic  |       |       |       | Raw    |
> | :------- | :-------- | :---- | :---- | :---- | :---- |
> | SNR      | Kappa ↑ | f1 ↑    | sens ↑  | fp/h ↓  | f1 ↑    |
> | None     | 0\.61     | 0\.59 | 0\.72 | 0\.15 | 0\.51 |
> | 60dB     | 0.58         | 0.54     | 0.71     | 0.12     | 0.49    |
> | 50dB     | 0\.54     | 0\.50 | 0\.72 | 0\.13 | 0\.47 |
> | 40dB     | 0\.36     | 0\.34 | 0\.74 | 0\.46 | 0\.31 |
> | 30dB | 0\.12     | 0\.12 | 0\.71 | 1\.22 | 0\.10 |
>
> As can be seen, MVPFormer is highly tolerant to noise and **maintains performance in the range of human experts up to 50dB,** giving further evidence regarding its potential clinical utility. In particular, while sensitivity remains high when increasing noise, the number of false positives rapidly increases. This is consistent with the fact that ictal periods often appear as high-frequency activity, which can mislead the model when in a noisy environment.

---

> ### Author Response · Authors · 2025-11-21
>
> ## W2. Computational overhead
>
> The Reviewer is correct in that MVPA provides greater expressiveness at the cost of more computation, but the efficient MVPA computation we employ, together with the Triton implementation, aims exactly at mitigating this.
>
> Following the Reviewer and Reviewer cFg6's suggestion, we now expand this comparison to include a form of linear attention (Nystromformer [1]), naive vanilla attention, and FlashAttention 2. All results have been moved to their own separate Section and can be found in Appendix A.7 Performance comparison, and an excerpt is reported below for discussion.
>
> The Table below reports the results for the naive implementation.
>
> | T \ C | 1 | 10 | 20 | 30 | 40 | 50 |
> |---|---|---|---|---|---|---|
> | 1 | 1.56 us / 0.02 GB | 1.75 us / 0.05 GB | 1.90 us / 0.06 GB | 1.96 us / 0.07 GB | 2.03 us / 0.08 GB | 2.12 us / 0.10 GB |
> | 10 | 1.72 us / 0.05 GB | 1.81 us / 0.18 GB | 2.59 us / 0.43 GB | 5.11 us / 0.81 GB | 7.53 us / 1.31 GB | 11.83 us / 1.92 GB |
> | 20 | 1.69 us / 0.06 GB | 2.64 us / 0.43 GB | 7.59 us / 1.30 GB | 15.29 us / 2.64 GB | 25.30 us / 4.47 GB | 38.15 us / 6.75 GB |
> | 30 | 1.77 us / 0.07 GB | 5.20 us / 0.81 GB | 15.27 us / 2.64 GB | 34.52 us / 5.54 GB | 67.64 us / 9.50 GB | 111.33 us / 14.52 GB |
> | 40 | 1.84 us / 0.08 GB | 7.68 us / 1.31 GB | 25.15 us / 4.47 GB | 67.48 us / 9.50 GB | 108.71 us / 16.43 GB | 161.07 us / 25.25 GB |
> | 50 | 1.74 us / 0.10 GB | 11.86 us / 1.92 GB | 37.77 us / 6.75 GB | 110.74 us / 14.52 GB | 160.72 us / 25.25 GB | 278.60 us / 38.90 GB |
>
> The Table below reports the results for the Triton implementation. Both runtime and memory consumption are significantly lower than the naive implementation, confirming the efficient nature of this implementation.
>
> | T \ C | 1 | 10 | 20 | 30 | 40 | 50 |
> |---|---|---|---|---|---|---|
> | 1 | 1.34 us / 0.02 GB | 1.40 us / 0.05 GB | 1.39 us / 0.06 GB | 1.42 us / 0.07 GB | 1.41 us / 0.08 GB | 1.49 us / 0.09 GB |
> | 10 | 1.41 us / 0.05 GB | 1.45 us / 0.14 GB | 1.78 us / 0.24 GB | 2.83 us / 0.34 GB | 4.10 us / 0.45 GB | 4.53 us / 0.56 GB |
> | 20 | 1.42 us / 0.06 GB | 1.69 us / 0.24 GB | 4.09 us / 0.44 GB | 5.66 us / 0.65 GB | 10.04 us / 0.87 GB | 12.10 us / 1.09 GB |
> | 30 | 1.42 us / 0.07 GB | 2.92 us / 0.34 GB | 6.00 us / 0.65 GB | 12.09 us / 0.96 GB | 18.93 us / 1.29 GB | 23.98 us / 1.63 GB |
> | 40 | 1.47 us / 0.08 GB | 3.98 us / 0.45 GB | 12.43 us / 0.87 GB | 16.17 us / 1.29 GB | 27.29 us / 1.73 GB | 37.68 us / 2.19 GB |
> | 50 | 1.44 us / 0.09 GB | 4.79 us / 0.55 GB | 11.69 us / 1.08 GB | 23.45 us / 1.62 GB | 36.28 us / 2.19 GB | 60.36 us / 2.75 GB |
>
> As can be seen, FlashMVPA enjoys very favourable scaling against all alternatives, comparable to FlashAttention 2, while also including information about the structure of the time-series at the attention level. Moreover, the naive implementation of MVPA and vanilla attention are actually very close in terms of both runtime and memory, indicating that MVPA's overhead is minimal.
>
> Finally, we provide an in-depth comparison with a simpler model in MV-Llama, which makes use of the lighter vanilla attention but is otherwise identical to MVPFormer in all other aspects. The results on all tasks confirm that MVPA, while being more computationally expensive, is highly beneficial to downstream performance.

---

> ### Author Response · Authors · 2025-11-21
>
> ## W3. Additional tasks
>
> As part of the current setup, we report results on 5 tasks (seizure detection, discrimination of pitch, discrimination of volume level, classification of sentence onset, and classification of speech) and on 4 datasets (Long-term iEEG, MAYO, FNUSA, and Brain TreeBank). These are the most widely available tasks for iEEG, and we couldn't assess other high-quality alternatives. A summary Table of all tasks, datasets, and results is reported below.
>
> |                   |           | SWEC      |           | MAYO      | FNUSA     | Brain TreeBank |           |           |           |
> | :---------------- | :-------- | :-------- | :-------- | :-------- | :-------- | :------------- | :-------- | :-------- | :-------- |
> |                   |           | Seizure   |           | Seizure   | Seizure   | Pitch          | Volume    | Onset     | Speech    |
> | Model             | Attention | Kappa ↑     | f1 ↑        | f1 ↑       | f1 ↑       | acc ↑           | acc ↑      | acc ↑      | acc ↑      |
> | MVPFormer         | MVPA      | **0\.61** | **0\.59** | **0\.36** | **0\.46** | /              | /         | /         | /         |
> | MVPFormer-S       | MVPA      | 0\.57     | 0\.53     | 0\.35     | 0\.46     | **0\.83**      | **0\.88** | **0\.87** | **0\.90** |
> | MV-Llama          | Vanilla   | 0\.11     | 0\.01     | /         | /         | 0\.63          | 0\.77     | 0\.80     | 0\.81     |
> | Brant             | Vanilla   | /         | /         | /         | /         | 0\.61          | 0\.74     | 0\.80     | 0\.80     |
> | Brant-2           | Vanilla   | 0\.08     | 0\.00     | 0\.19     | 0\.46     | /              | /         | /         | /         |
> | BrainBERT         | Vanilla   | 0\.00     | 0\.00     | /         | /         | 0\.59          | 0\.66     | 0\.70     | 0\.71     |
> | PopT&dagger; | Vanilla   | /         | /         | /         | /         | 0\.74          | 0\.87     | _**0\.90**_     | **_0\.93_**     |
> | PopT              | Vanilla   | /         | /         | /         | /         | 0\.62          | 0\.76     | 0\.81     | 0\.83     |
>
> However, we also evaluate the validity of our approach in the generic time-series setting, to extend MVPA's applicability even beyond its design domain. We test 3 forecasting tasks and 10 classification tasks, reaching SOTA performance across the board.
>
> ---
>
> ## W4. Limitations of the Long-term iEEG dataset
>
> We take great care to ensure our Long-term iEEG dataset is of the highest quality and fully respects all relevant safety and privacy best practices. As the Reviewer very correctly points out in Q5, these concerns are a high priority for any research in the clinical domain.
>
> For this reason, we are unable to provide information about the location of the electrodes, as such metadata could potentially be misused to violate deidentification (as mentioned by Reviewer cFg6).
>
> ---
>
> ## W5. Comparisons with baselines
>
> We provide the full subject-by-subject breakdown of the results on the Long-term iEEG dataset in Appendix G Additional results, which helps characterize the cross-subject generalization capabilities of the models. In particular, while MVPFormer shows strong zero-shot generalization, other models mostly fail to generalize. As an example, BrainBERT never predicts a seizure on the unseen patients, while Brant-2's behaviour is more nuanced. We have added this specification to Section 5.2.1 Seizure detection task.
>
> At the same time, the Long-term iEEG dataset is the design dataset of MVPFormer, and thus might bias results. Therefore, we also compare the models on fully unseen datasets: MAYO, FNUSA, and Brain TreeBank. Also in this setting, MVPFormer achieves SOTA performance even against other models specifically designed for the task at hand (PopT on the Brain TreeBank). For these datasets, the results are always reported aggregated, such that a subject-by-subject analysis is unfortunately not possible.

---

> ### Author Response · Authors · 2025-11-21
>
> ## Q1. Time-series different from iEEG
>
> The Reviewer raises an important point about the applicability of MVPA beyond the iEEG domain. Following the Reviewer and Reviewer oKKd's suggestion, we move the results on the general time-series to Section 6 Evaluation on general time-series, to further highlight the importance of these results.
>
> In particular, we test MVPFormer against 5 models on 3 general forecasting tasks and 10 classification tasks, ranging from weather data, to sensors in power plants and chemical plants, to traffic data. On this large variety of tasks and time-series, MVPFormer once again showcase SOTA performance, indicating that the model's flexibility generalizes well to many other domains. A summary of the forecasting tasks is presented in the Table below.
>
> | Model   |        | MVPFormer |      | Transformer |      | PatchTST |      | TimesFM  |      | TimeMixer |      | WPMixer  |      |
> |---------|--------|-----------|------|-------------|------|----------|------|----------|------|-----------|------|----------|------|
> | Dataset | Length | MSE ↓      | MAE ↓ | MSE ↓        | MAE ↓ | MSE ↓     | MAE ↓ | MSE ↓     | MAE ↓  | MSE ↓      | MAE ↓ | MSE ↓     | MAE ↓ |
> | ETTh1   | Avg.   | **0.45**  | 0.45 | 1.00        | 0.80 | **0.45** | 0.45 | **0.45** | 0.45 | **0.45**  | 0.44 | **0.45** | 0.44 |
> | ETTh2   | Avg.   | **0.38**  | 0.41 | 3.37        | 1.48 | 0.39     | 0.41 | **0.38** | 0.41 | 0.39      | 0.41 | **0.38** | 0.41 |
> | Weather | Avg.   | **0.25**      | 0.28 | 0.59        | 0.53 | 0.26 | 0.28 | 0.26 | 0.28 | **0.25**      | 0.28 | **0.25**     | 0.28 |
>
> ---
>
> ## Q2. Varied electrode setup
>
> While EEG setups are standardized thanks to the 10-20 systems, unfortunately, no such system exists for intracranial EEG (iEEG). All implantations are fully customized to the subject based on their clinical needs, with placement and number of electrodes varying significantly. In the case of the Long-term iEEG dataset, the number of channels ranges from 29 to 128, and the placement of the electrodes, while not publicly available, is also highly heterogeneous as is usually the case.
>
> Therefore, MVPFormer was specifically developed to handle such heterogeneous setups, with the channel component autonomously learning a map of the channel connectivity (see Appendix G.11 Channel connectivity map). This heterogeneity is well reflected in the classification performance of MVPFormer (see Appendix G.10 Patient classification difficulty), with some subjects being notably more difficult to classify than others.
>
> ---
>
> ## Q3. Dataset scaling
>
> Currently, a full pre-training and inference run takes approximately two weeks with 8xNvidia A100 GPUs using our optimized Triton-based FlashMVPA. The Reviewer's concern is well placed, and is precisely the reason for the development of FlashMVPA, as the naive implementation of MVPA would be too slow to be practical.
>
> Given further potential optimizations to FlashMVPA, increasing the size of the dataset beyond what we have currently available should provide feasible even with with much less computational resources as compared to what is being used to train LLMs. Therefore, we do not foresee this to be a bottleneck for frontier electrophylogical model training.
>
> Real-time inference is, in contrast, a more notable constraint. Based on what we have observed regarding the scaling of the models (see Appendix G.4 Effects of the scale of the model), a smaller model could prove sufficient in many tasks, providing a lower-cost alternative to the full MVPFormer. Other techniques, such as transfer learning or distillation, could also prove useful to improve the scaling at the cost of minor performance impact.
>
> ---
>
> ## Q4. Non-standard setups
>
> As mentioned in W1 and Q2, MVPFormer is specifically designed to cope with channel variability. Indeed, no "standard" setup exists for iEEG implantations. Therefore, all recording in the Long-term iEEG dataset come from subjects with highly heterogeneous setups in terms of number and placement of the electrodes.

---

> ### Author Response · Authors · 2025-11-21
>
> ## Q5. Open-sourcing
>
> We thank the Reviewer for underscoring the importance of safety and privacy when open-sourcing clinical models and datasets.
>
> On the one hand, we have followed all relevant procedures for the collection of the dataset in terms of patient consent and approval, and have ensured strict adherence to the IRB mandate. For example, as also mentioned in W4, we remove all information regarding the location of the electrodes to prevent any leakage. On the other hand, we have taken all necessary mitigations to reduce the risk of misuse of our model, while still providing the model and data to all in the spirit of open science.
>
> We strongly emphasize the restrictions on the usage of both the model and the dataset in the respective public webpages. As stated in license available in the supplementary material, **our dataset and models may only be used for research purposes, and may not be used for any other purpose, especially clinical use.** Our disclaimers have undergone review at both the Institution collecting the data and the Institution where the model was trained.
>
> Beyond the usage restrictions mentioned above, no other restrictions are placed on data sharing.
>
> [1] Xiong, Yunyang, et al. "Nyströmformer: A nyström-based algorithm for approximating self-attention." AAAI (2021).

---

> > ### Author Response · Authors · 2025-11-28
> >
> > Thank you once again for your valuable feedback and dedicated service as a reviewer. We have carefully considered your input and made significant updates to the manuscript in response. We genuinely value your perspective and encourage you to review our rebuttal and the revised manuscript. We hope our revisions address your concerns and help you reassess our work. Please don not hesitate to reach out if further clarification or discussion would be helpful.

---

### Official Review · Reviewer_qWbc · 2025-11-01

**Soundness:** 3
**Presentation:** 3
**Contribution:** 4
**Rating:** 8
**Confidence:** 5

**Summary:**

This paper presents a new method for learning representations of multi-channel intracranial activity. The input is one embedding vector per segment of activity, augmented with position vectors for time and for channel. Gains in efficiency can be had by reusing attention computations for channels that share the same time. During pretraining, a discriminative loss is used. Downstream evaluation is done on a seizure detection task and audio-linguistic tasks. Additionally, the paper releases 10K hours of iEEG recordings.

**Strengths:**

●	Overall, this paper represents a very strong contribution to the community.
●	Releases a large amount of data. I am personally not aware of a larger publicly available iEEG dataset. This is a big boon to the community. Especially so, since many other foundation models for intracranial signal train on private data, e.g., BrainWave.
●	Evaluation is thorough: the authors use their own seizure detection task as well as the Brain Treebank tasks.
●	Performance on the epilepsy detection task exceeds the current state of the art

**Weaknesses:**

●	Am I misunderstanding something? The paper refers to a "generative" objective, but the loss seems to be discriminative, i.e., an InfoNCE loss? The output of the model is in the embedding space, not neural activity, correct?
●	Line 364: The claim is that the choice of objective is justified by an ablation. But if I read appendix G.14 correctly, it seems that there is only justification for doing pretraining, not the specific type of pretraining, i.e., some other choice of pre-training objective. This is not a major weakness, but more precise wording is probably needed.
●	In the related works section, it would be good to discuss various factored approaches that are proposed for spatio-temporal data. For example, for processing movies: https://arxiv.org/pdf/2106.05968.

**Questions:**

●	For comparison, you can cite BrainWave: A Brain Signal Foundation Model for Clinical Applications https://arxiv.org/pdf/2402.10251. They have 35K hours of data. But most of it is EEG and most of it is private.
●	Figure 2: In the legend, what is "two-step"? It looks like this is defined in Fig 7 of the appendix, but it would be good to have that in the main text somewhere.
●	Line 329: "We must also consider that our evaluation setup involves many more subjects and ictal events than are reported for human experts," — does this mean that the expert annotations contain false negatives? I.e., periods of activity that are not marked as seizures, but should be?

---

> ### Author Response · Authors · 2025-11-21
>
> We thank the Reviewer for highlighting the strength of our contribution and the thoroughness of our evaluation. We are happy to read that our dataset is recognized as highly valuable to the community at large.
>
> We have improved the clarity of the passages suggested by the Reviewer to increase the readability of our paper.
>
> ---
>
> ## W1. Generative objective
>
> The Reviewer is indeed correct that the contrastive loss is similar to InfoNCE, and that the prediction is made on the model's latent space. As we mention in Section 3 MVPFormer, we therefore generate future embeddings, rather than raw signals. Therefore, our model is more similar to continuous-embedding [1] generative models or flow-based [2] generative models. Nonetheless, our MVPA mechanism is not limited to this setup and we are actively working towards making a fully end-to-end generation of raw signals.
>
> We further clarify this aspect in Section 3 MVPFormer.
>
> ---
>
> ## W2. Pre-training objective
>
> The ablation mentioned by the Reviewer is specifically to evaluate whether a foundation setting, with a base generative pre-training task, is useful. All our models are built with the contrastive loss function for the initial pre-training task. Following the Reviewer's suggestion, we clarify this aspect in Section 5.2.1 Seizure detection task.
>
> ---
>
> ## W3. Related works
>
> We thank the Reviewer for bringing this interesting paper to our attention. We strive to include all relevant works, including many other multi-dimensional attention mechanisms. We have now added both the mentioned paper [3] and ViViT [4] to our comprehensive comparison Table in Appendix A.2 Comparison with alternative attention mechanisms. In particular, space-time mixing [3] reduces the computational complexity of the full quadratic self-attention by only computing attention across the channels, and then performing a simple averaging in time. ViViT instead uses sequential encoders to disentangle space and time. In contrast, MVPA fully processes both aspects of the signal in parallel at the attention level.
>
> ---
>
> ## Q1. BrainWave
>
> BrainWave [5] is, in fact, a new revision of Brant-2 [6]. Following the Reviewer's suggestion, we now also mention BrainWave's iEEG dataset explicitly in Section 4 Long-term iEEG dataset.
>
> ---
>
> ## Q2. Caption Fig. 2
>
> As mentioned by the Reviewer, the two-step target is the signal twice removed in the future, i.e. the five seconds of iEEG signal coming after the true target. We have now updated Fig. 2's caption to properly explain the two-step scheme.
>
> ---
>
> ## Q3. Evaluation setup
>
> In our dataset, each patient is fully annotated by an expert, to ensure the highest possible level of data quality. However, comparative studies involving the annotations by multiple experts of the same episodes (which inform our Kappa score baseline) usually involve only a few tens of hand-picked samples. This is due to the already pressing workload of epileptologists. In contrast, our evaluation routine consists of many hundreds of episodes picked at random from any patient, making our scenario more challenging that human comparisons.
>
> We now make this clearer in Section 5.2.1 Seizure detection task.
>
> [1] Tschannen, Michael, Cian Eastwood, and Fabian Mentzer. "Givt: Generative infinite-vocabulary transformers." ECCV (2024).
>
> [2] Li, Bohan, et al. "On the sentence embeddings from pre-trained language models." arXiv preprint arXiv:2011.05864 (2020).
>
> [3] Arnab, Anurag, et al. "Vivit: A video vision transformer." Proceedings of the IEEE/CVF international conference on computer vision (2021).
>
> [4] Bulat, Adrian, et al. "Space-time Mixing Attention for Video Transformer." NeurIPS (2021).
>
> [5] Yuan, Zhizhang, et al. "Brainwave: A brain signal foundation model for clinical applications." arXiv preprint arXiv:2402.10251v7 (2024).
>
> [6] Yuan, Zhizhang, et al. "Brant-2: Foundation model for brain signals." arXiv preprint arXiv:2402.10251v4 (2024).

---

> > ### Comment · Reviewer_qWbc · 2025-11-25
> >
> > I thank the authors for their detailed answers. I apologize for bringing this up late in the review process, but could I ask for a little more technical detail in describing the seizure detection task? I’ve read appendices E and F, but I still don’t have a precise idea of how I would reproduce this task, given the data. From what I can tell: the inputs to the seizure detection task are 5 seconds (?) of activity. The model classifies this activity as ictal or non-ictal. The dataset is created by segmenting the entire (?) recording into 5 second segments. Ideally, accompanying code would clarify these points. I think clarifying this task would help reproducibility. In any case, I will continue to advocate for this paper’s acceptance. It is clearly a strong contribution.

---

> > > ### Author Response · Authors · 2025-11-26
> > >
> > > We sincerely thank the Reviewer for appreciating and advocating for our work, and are glad to revise any unclear aspect of our manuscript.
> > >
> > > The seizure classification task is conducted analogously to the pre-training task, according to each model's specification. In the case of MVPFormer, we segment the whole dataset in windows of 5s*100 (approximately 8 minutes), with a stride of 5 seconds. The ictal/non-ictal decision is taken on the last 5 second window, with the previous 99 windows used as context. For comparison, Brant-2 uses 16 windows of 3 seconds (less than a minute), again with a stride of 3 seconds. The longer context length is a notable advantage of MVPFormer and highlights the importance of the computational efficiency of MVPA.
> > >
> > > We have added this clarification to Appendix E Generation of results. Our public code repository contains the data-loading logic (in particular, the dataset segmentation) and configuration files that also clarify this aspect.

---

### Official Review · Reviewer_cFg6 · 2025-11-01

**Soundness:** 3
**Presentation:** 2
**Contribution:** 3
**Rating:** 4
**Confidence:** 3

**Summary:**

The paper proposes Multi-Variate Parallel Attention (MVPA), an attention mechanism that splits attention into content, temporal, and channel components to better handle heterogeneous multi-channel time series. Building on MVPA, the authors train MVPFormer, a generative pre-trained foundation model for iEEG on a newly released “Long-term iEEG” corpus of about 10k recording hours. MVPFormer generalizes zero-shot across patients for seizure detection and performs competitively on four Brain TreeBank decoding tasks, while MVPA also matches or exceeds strong baselines on standard time-series forecasting and classification benchmarks.

Main contributions include:
1 MVPA: a self-attention variant that separately models content, time, and channel structure to support generalization under variable, heterogeneous channels.

2 MVPFormer: a generative, MVPA-powered foundation model for electrophysiology that outperforms vanilla-attention baselines for seizure detection and improves over a matched discriminative model.

3 Long-term iEEG dataset: release of a large public corpus, roughly 9,300 to 10,000 hours across 68 subjects, enabling open data, code, and weights for the community.

**Strengths:**

1 MVPA factorizes self-attention into content, time, and channel to handle heterogeneous, variable-channel signals that standard attention struggles with. MVPFormer’s generative pretraining and the Long-term iEEG corpus add fresh angles on both model and data.

2 The method is solid  and scalable, with a coherent pretraining recipe followed by light adaptation.

3. The three MVPA components and their roles in the logits are well-explained with figures. Pretraining and fine-tuning protocols are modular, and the dataset description is specific enough to judge external validity.

4  Addressing variable, heterogeneous channels is a core deployment blocker, and MVPA tackles it directly. The approach likely transfers beyond iEEG, while the large open corpus and a usable pretrained model can accelerate community progress.

**Weaknesses:**

1 Zero-shot tests use manual channel selection at inference, and preprocessing, post-processing, and thresholds are not harmonized across baselines. The reported gains may stem from pipeline differences rather than the core method.


2 Overstated “expert-level” claim: A single Kappa threshold from prior work is used instead of a same-dataset, same-protocol human comparison. No per-subject confidence intervals or significance tests are reported.

3 The MVPA decomposition lacks a rigorous derivation and error bounds. How summed logits are scaled or weighted is unclear, and notation/dimensions are inconsistent in places.

4: The stated complexity does not square with a local-window content term, and there are no reproducible runtime or memory curves versus sequence length, channel count, or window size. Missing direct comparisons with other efficient attention variants.

5: Pretraining choices (negative sampling strategy, temperature, hard negatives, sample count) are not systematically ablated. Cross-dataset protocols differ, and strong baselines are not adapted for channel heterogeneity, weakening SOTA claims.

**Questions:**

1 Can you provide a fully automatic end-to-end inference pipeline with no manual channel selection, and report side-by-side results on the same test set? Use identical preprocessing, post-processing, and thresholding across all models, select thresholds on a shared validation split, and include per-subject distributions with 95% CIs plus clear leakage controls (window overlap, session boundaries).

2 Can you give a rigorous derivation from dual-encoding attention to the sum of content, time, and channel terms, stating the conditions under which cross terms are dropped and providing error bounds? Please clarify how the summed logits are scaled or weighted (and whether weights are learnable). Show ablations that remove each term, swap relative for absolute encodings, and vary channel count; add a synthetic study to demonstrate when each component is required.

3 Can you release reproducible runtime and memory curves versus sequence length T, channel count C, and local window L, reflecting the expected O(T·C·L) behavior for the content term? Provide controlled throughput comparisons on the same hardware and batch size against strong efficient-attention baselines (axial or factorized attention, linear-time variants, FlashAttention with GQA), with scripts and fixed seeds.

4 Can you supply a same-dataset, same-protocol head-to-head with human experts, including inter-rater agreement, event-level error breakdowns, and statistical tests? Please adapt strong baselines for channel heterogeneity, enforce a unified evaluation protocol across datasets, and add few-shot curves to separate pretraining benefits from protocol artifacts. If the method still leads under these stricter comparisons, the “expert-level” statement becomes defensible.

**Details Of Ethics Concerns:**

Privacy, security, safety: Long-duration human iEEG can be re-identified when linked with site or timing metadata, and the paper does not document a de-identification audit, adversarial linkage tests, or access controls.

Responsible research practice: IRB approval and consent for open release are not clearly stated, label provenance and annotator compensation are unspecified.

---

> ### Author Response · Authors · 2025-11-21
>
> We thank the Reviewer for highlighting the transferability and freshness of our approach, and we are glad to read that our figures and explanations were appreciated.
>
> Following the Reviewer's suggestion, we now provide a fully standardized end-to-end evaluation protocol on the Long-term iEEG dataset and report all its results for the tested models. We also increase the number of attention baselines compared in our runtime and memory benchmarks, to better capture the characteristics of the considered alternatives. Moreover, we strengthen the characterization of MVPA and improve the clarity of all aspects mentioned by the Reviewer.
>
> ---
>
> ## W1. Evaluation pipeline
>
> For all evaluated models (MVPFormer-M, MVPFormer-S, MV-Llama, Brant-2, and BrainBERT) on the seizure detection tasks we use the exact same pre-processing and post-processing pipeline, tailoring the details following each model's original paper. Specifically, we use the published pre-trained weights for Brant-2 and BrainBERT to ensure consistency and to give all models fair representation. Therefore, some input parameters are adjusted to match their pre-training routine. The Table below reports all the used techniques:
>
> |  |  | MVPFormer-M | MVPFormer-S | MV-Llama | Brant-2 | BrainBERT |
> |---|---|---|---|---|---|---|
> | Pre-processing | **Channel selection [NEW]** | **Automatic** | **Automatic** | **Automatic** | **Automatic** | **Automatic** |
> |  | Patch length | 5s | 5s | 5s | 3s&dagger; | 5s&dagger; |
> |  | Number of patches | 100 | 100 | 100 | 16&dagger; | 1&dagger; |
> |  | Sampling rate | 512Hz | 512Hz | 512Hz | 512Hz | 512Hz |
> | Post-processing | Minimum episode length | 20s | 20s | 20s | 20s | 20s |
> |  | Merging threshold | 5m | 5m | 5m | 5m | 5m |
> |  | Minimum responses | 5 | 5 | 5 | 5 | 5 |
> |  | Threshold | argmax | argmax | argmax | argmax | argmax |
>
> As can be seen, all parameters are equal except when made necessary by the model's characteristics (represented with a &dagger;). The training and testing subjects are also kept identical between the different models (in some instances, a subject cannot be evaluated by fixed-channel models such as Brant-2, as they do not have enough channels).
>
> Moreover, following the Reviewer's suggestion, we now perform an automatic channel selection step based on variance and kurtosis, to eliminate any possible bias in this step as well. Specifically, we rank the channels based on the first 30 minutes of activity using the formula $variance/(1 + kurtosis)$ and keep only the channels that are between the 1st and 99th percentile of variance and below the 95th percentile of kurtosis. For all patients, we use 32 channels for compatibility with the fixed-channel models.
>
> A summary of the results is presented in the Table below. The performance of MVPFormer actually increases with this setup to a Kappa score of 0.61 and 0.15 fp/h, with 81% of the subjects having less than 0.05 fp/h.
>
> | Model | Attention | Kappa ↑ | f1 ↑ | sens ↑ | fp/h ↓ |
> |---|---|---|---|---|---|
> | MVPFormer | MVPA | 0.61 | 0.59 | 0.72 | 0.15 |
> | MVPFormer-S | MVPA | 0.57 | 0.53 | 0.71 | 0.12 |
> | Brant-2 | Vanilla | 0.06 | 0.01 | 0.01 | 0.11 |
> | BrainBERT | Vanilla | 0.00 | 0.00 | 0.00 | 0.00 |
> | MV-Llama | Vanilla | 0.11 | 0.01 | 0.01 | 0.02 |
>
> We have now replaced all results with manual channel selection with the results obtained with automatic channel selection in Section 5.2.1 Seizure detection task. We have added a detailed explanation about the automatic channel selection in Appendix E.1 Channel selection.
>
> For the Brain TreeBank tasks, we follow the pre-processing and post-processing routine reported in the PopT paper [1], once again to ensure the comparability of our results.
>
> ---
>
> ## W2. Expert-level performance
>
> We collect multiple sources [2] [3] [4] to estimate the Kappa score among human experts, albeit on different datasets than our Long-term iEEG dataset. However, the Reviewer is correct in that comparative studies involving multiple expert annotations of the same episodes usually involve only a few tens of hand-picked samples. In contrast, our evaluation routine consists of many hundreds of episodes picked at random from any patient, making our scenario more challenging than human comparisons.
>
> Unfortunately, a human-to-human comparison on the Long-term iEEG dataset is not possible, as it would involve multiple expert epileptologists annotating thousands of hours of recordings, which is incompatible with their already pressing workload.
>
> To strengthen the statistical validity of our results, we now also report all 95% confidence intervals for the Kappa scores in Appendix G Additional results. An excerpt for MVPFormer-M is reported below.
>
> | Subject | Kappa ↑ | 95% CI ↓ |
> |---|---|---|
> | ID19 | 0.00 | 0.00 |
> | ID20 | 0.87 | 0.05 |
> | ID21 | 0.82 | 0.09 |
> | ID22 | 0.99 | 0.02 |
> | ID23 | 0.12 | 0.03 |
> | ID24 | 0.92 | 0.01 |
> | ID25 | 0.00 | 0.00 |

---

> ### Author Response · Authors · 2025-11-21
>
> ## W3. MVPA clarity
>
> We have strived to detail every aspect of MVPA in Section 2 MVPA and Appendix A Details on multi-variate parallel attention (MVPA). To address the Reviewer's concerns, we have now specified in Section 2 MVPA which cross-terms are cancelled to retrieve MVPA and the relative errors with respect to a full quadratic computation, and have added the full softmax equation to clarify the logits weighting. We have also double-checked all notations to ensure consistency, and kindly ask the Reviewer to point out any more issues in case they find any.
>
> ---
>
> ## W4. MVPA complexity
>
> We specify the derivation of the subquadratic complexity of MVPA in Appendix A.3 Efficient computation of MVPA. Given time length $T$, channel count $C$, and a local window $L $the full complexity becomes $O(T^2\times C + T\times C^2 + L^2\times C)$. This simplifies to the stated $O(T^2\times C + T\times C^2)$ when $L \ll T$.
>
> We had also included a comparison in runtime and memory usage with respect to T and C between a naive implementation of MVPA and the optimized Triton version. Now, following the Reviewer and Reviewer dxZg's suggestion, we expand this comparison to include a form of linear attention (Nystromformer [5]), naive vanilla attention, and FlashAttention 2. All results have been moved to their own separate Section and can be found in Appendix A.7 Performance comparison, and an excerpt is reported below for discussion.
>
> The Table below reports the results for the naive implementation.
>
> | T \ C | 1 | 10 | 20 | 30 | 40 | 50 |
> |---|---|---|---|---|---|---|
> | 1 | 1.56 us / 0.02 GB | 1.75 us / 0.05 GB | 1.90 us / 0.06 GB | 1.96 us / 0.07 GB | 2.03 us / 0.08 GB | 2.12 us / 0.10 GB |
> | 10 | 1.72 us / 0.05 GB | 1.81 us / 0.18 GB | 2.59 us / 0.43 GB | 5.11 us / 0.81 GB | 7.53 us / 1.31 GB | 11.83 us / 1.92 GB |
> | 20 | 1.69 us / 0.06 GB | 2.64 us / 0.43 GB | 7.59 us / 1.30 GB | 15.29 us / 2.64 GB | 25.30 us / 4.47 GB | 38.15 us / 6.75 GB |
> | 30 | 1.77 us / 0.07 GB | 5.20 us / 0.81 GB | 15.27 us / 2.64 GB | 34.52 us / 5.54 GB | 67.64 us / 9.50 GB | 111.33 us / 14.52 GB |
> | 40 | 1.84 us / 0.08 GB | 7.68 us / 1.31 GB | 25.15 us / 4.47 GB | 67.48 us / 9.50 GB | 108.71 us / 16.43 GB | 161.07 us / 25.25 GB |
> | 50 | 1.74 us / 0.10 GB | 11.86 us / 1.92 GB | 37.77 us / 6.75 GB | 110.74 us / 14.52 GB | 160.72 us / 25.25 GB | 278.60 us / 38.90 GB |
>
> The Table below reports the results for the Triton implementation. Both runtime and memory consumption are significantly lower than the naive implementation, confirming the efficient nature of this implementation.
>
> | T \ C | 1 | 10 | 20 | 30 | 40 | 50 |
> |---|---|---|---|---|---|---|
> | 1 | 1.34 us / 0.02 GB | 1.40 us / 0.05 GB | 1.39 us / 0.06 GB | 1.42 us / 0.07 GB | 1.41 us / 0.08 GB | 1.49 us / 0.09 GB |
> | 10 | 1.41 us / 0.05 GB | 1.45 us / 0.14 GB | 1.78 us / 0.24 GB | 2.83 us / 0.34 GB | 4.10 us / 0.45 GB | 4.53 us / 0.56 GB |
> | 20 | 1.42 us / 0.06 GB | 1.69 us / 0.24 GB | 4.09 us / 0.44 GB | 5.66 us / 0.65 GB | 10.04 us / 0.87 GB | 12.10 us / 1.09 GB |
> | 30 | 1.42 us / 0.07 GB | 2.92 us / 0.34 GB | 6.00 us / 0.65 GB | 12.09 us / 0.96 GB | 18.93 us / 1.29 GB | 23.98 us / 1.63 GB |
> | 40 | 1.47 us / 0.08 GB | 3.98 us / 0.45 GB | 12.43 us / 0.87 GB | 16.17 us / 1.29 GB | 27.29 us / 1.73 GB | 37.68 us / 2.19 GB |
> | 50 | 1.44 us / 0.09 GB | 4.79 us / 0.55 GB | 11.69 us / 1.08 GB | 23.45 us / 1.62 GB | 36.28 us / 2.19 GB | 60.36 us / 2.75 GB |
>
> As can be seen, FlashMVPA enjoys very favourable scaling against all alternatives, comparable to FlashAttention 2, while also including information about the structure of the time-series at the attention level.
>
> ---
>
> ## W5. Pre-training ablations
>
> We perform pre-training ablations on the most impactful design choices of MVPFormer, namely: number of parameters (Appendix G.4 Effects of the scale of the model), choice of attention (Appendix G.5 Effects of the attention mechanism), scale of the pre-training dataset (Appendix G.9 Effects of the scale of the pre-training dataset), and number of channels (Appendix G.8 Effects of the selection of channels).
>
> On the recommendation of the Reviewer, we now also add an ablation between manual and automatic selection of channels. Finally, following the suggestion of Reviewer dxZg, we also include ablations regarding the resilience of the model to the noise in the input signal (see Appendix G.14 Resilience to noise). Due to the resources required to fully pretrain and obtain new results (approximately two weeks with 8xNvidia A100 GPUs), it is not feasible to ablate all aspects of the model and the pre-training and fine-tuning routines.
>
> Finally, all protocols are consistent between models to ensure fair comparisons, using the training routing and the pre-training weights specified for both BrainBERT and Brant-2, and now also an automated channel selection routine as suggested by the Reviewer.

---

> ### Author Response · Authors · 2025-11-21
>
> ## Q1. End-to-end inference pipeline
>
> As mentioned in W1, we use a consistent pipeline for all models, harmonized to use all equal pre- and post-processing steps up to the dissimilarity in the models' intrinsic characteristics. In particular, for BrainBERT and Brant-2 we use the publicly available pre-trained weights to ensure fairness.
>
> Following the Reviewer's suggestion, we also implement an automated channel selection based on variance and kurtosis, and are glad to report that we find an increase in performance for MVPFormer. The full results can be found in the Table above in W1 and in Appendix G.3 Conventional evaluation.
>
> The per-subject performance breakdown can be found in Appendix G Additional results, to which we now add the 95% confidence interval to increase the statistical validity of our results. Specifically, there is no leakage between training and testing events, as the subjects are strictly not shared between the two. Except for cases where subjects do not have enough channels to be evaluated by fixed-channel models such as Brant-2, the training and testing subjects are identical for all models.
>
> ---
>
> ## Q2. Derivation of the three components
>
> We are glad to read that the Reviewer found the clarity of our explanation of the three components to be one of the strengths of our paper, and have now reworked Section 2. MVPA to address all remaining concerns.
>
> Specifically, as mentioned in W2, we have provided the complete characterization of the dropped terms, and have added the complete softmax equation to clarify the scaling of the logits.
>
> Following the suggestions of the Reviewer and Reviewer oKKd, we have now added an ablation of the three components on the traditional time-series forecasting tasks. The results are reported in Section 6.3 Ablation of the three components (and also below for ease of reading). They indicate that all three components are necessary to achieve the full performance of MVPA.
>
> |  | MVPA | Content-only | Time-only | Channel-only | None |
> |---|---|---|---|---|---|
> |  | MSE ↓ | MSE ↓ | MSE ↓ | MSE ↓ | MSE ↓ |
> | ETTh1 | **0.45** | 0.46 | 0.46 | 0.46 | 0.47 |
> | ETTh2 | **0.38** | 0.38 | 0.38 | 0.38 | 0.40 |
> | Weather | **0.25** | 0.27 | 0.27 | 0.26 | 0.27 |
>
> Interestingly, the performance drop between MVPA and the ablated variants is greater for the Weather dataset, which has 21 channels against the 7 of ETTh1 and ETTh1, giving further indication that the combination of the components gives MVPA more powerful processing of strongly multi-variate time-series.
>
> On the seizure detection task of the Long-term iEEG dataset, we provide ablations of the channel count and channel selection in Appendix G.8 Effects of the selection of channels, as discussed in W1 and Q1.
>
> Unfortunately, it is not possible to ablate the relative encoding scheme of MVPA, as it is an intrinsic characteristic of the technique, as inherited by the Transformer-XL relative characterization.
>
> ---
>
> ## Q3. Runtime and memory curves
>
> We previously provided a comparison in runtime and memory usage with respect to T and C between a naive implementation of MVPA and the optimized Triton version.
>
> As mentioned in W4, we now expand this comparison to include a form of linear attention (Nystromformer [5]), naive vanilla attention, and FlashAttention 2. The full results are in Appendix A.7 Performance comparison, and an excerpt can be found in W4. FlashMVPA enjoys very favourable scaling against all alternatives, comparable to FlashAttention 2, while also including information about the structure of the time-series at the attention level. These results were all gathered on the same hardware and the same script, with fixed seeds.

---

> ### Author Response · Authors · 2025-11-21
>
> ## Q4. Human-to-human comparison
>
> While we collect multiple comparative human-to-human studies [2] [3] [4] to obtain a meaningful baseline Kappa score, the Reviewer correctly identifies expert comparisons as a challenge in the field.
>
> Unfortunately, a human-to-human comparison on our Long-term iEEG dataset would require tremendous resources for a clinical study on almost 10,000 hours of data by multiple expert epileptologists. Such a control experiment would be extremely valuable to the field at large, but no study of this size exists and they are currently not feasible. For reference, existing works use less than 50 hours of recordings.
>
> [1] Chau, Geeling, et al. "Population Transformer: Learning Population-level Representations of Neural Activity." ICLR (2025).
>
> [2] Halford, Jonathan J., et al. "Inter-rater agreement on identification of electrographic seizures and periodic discharges in ICU EEG recordings." Clinical Neurophysiology (2015).
>
> [3] Grant, Arthur C., et al. "EEG interpretation reliability and interpreter confidence: a large single-center study." Epilepsy & Behavior (2014).
>
> [4] Quigg, Mark, et al. "Interrater reliability in interpretation of electrocorticographic seizure detections of the responsive neurostimulator." Epilepsia (2015).
>
> [5] Xiong, Yunyang, et al. "Nyströmformer: A nyström-based algorithm for approximating self-attention." AAAI (2021).
>
> ---
>
> ## Ethics concerns
>
> We thank the Reviewer for underscoring the importance of safety and privacy when open-sourcing clinical models and datasets.
>
> On the one hand, we have followed all relevant procedures for the collection of the dataset in terms of patient consent and approval, and have ensured strict adherence to the IRB mandate. For example, as also mentioned by Reviewer dxZg, we remove all information regarding the location of the electrodes to prevent any leakage. Given the thorough deidentification process followed, no other restrictions are placed on the dataset, including access control.
>
> On the other hand, we have taken all necessary mitigations to reduce the risk of misuse of our model, while still providing the model and data to all in the spirit of open science. As stated in the paper, all recordings were annotated by an expert neurologist in the context of a clinical study.
>
> Moreover, we strongly emphasize the restrictions on the usage of both the model and the dataset in the respective public webpages. As stated in license available in the supplementary material, **our dataset and models may only be used for research purposes, and may not be used for any other purpose, especially clinical use.** Our disclaimers have undergone review at both the Institution collecting the data and the Institution where the model was trained.

---

> ### Comment · Reviewer_cFg6 · 2025-11-26
>
> The authors have provided a very solid report in rebuttal: the weakness I raised has been well-addressed. The method is now demonstrably robust, and the 10k-hour dataset remains a major community asset. I will raise my score accordingly.

---

### Official Review · Reviewer_oKKd · 2025-11-01

**Soundness:** 2
**Presentation:** 3
**Contribution:** 2
**Rating:** 4
**Confidence:** 3

**Summary:**

This paper introduces Multi-variate Parallel Attention (MVPA), which disentangles content, temporal, and spatial attention to handle heterogeneous time-series with varying channel configurations, and applies it to build MVPFormer, an iEEG foundation model achieving SOTA results on seizure detection and decoding tasks.

**Strengths:**

- The paper is well-written with a clear motivation for the problem and an architectural approach.

- The strong architecture design enables practical foundation models for clinical iEEG

- MVPA's decomposition of attention into content, temporal, and spatial components is novel and specifically addresses the real-world challenge of heterogeneous channel configurations in clinical data.

- The model demonstrates superior performance on seizure detection across three datasets and competitive results on Brain TreeBank decoding tasks and standard time-series benchmarks.

**Weaknesses:**

- The model is trained only on iEEG data, whereas foundation models typically leverage diverse datasets across multiple domains and modalities.
- The model is fine-tuned on target tasks, so calling evaluation on "unseen subjects" zero-shot is inaccurate. Normally, a true zero-shot would require no task-specific training data. Should be called fewshots?
- Section 5.3 abruptly shifts to generic time-series forecasting and classification tasks, creating a disjointed narrative that dilutes the paper's clinical focus.
- The paper is missing an ablation study of the proposed attention and the tree components in the main body.

**Questions:**

What specific criteria define a foundation model in your view, and how does MVPFormer satisfy these conditions, given that it's trained on a single modality (iEEG) from one domain and requires fine-tuning for downstream tasks rather than demonstrating broad zero-shot generalization?

---

> ### Author Response · Authors · 2025-11-21
>
> We thank the Reviewer for appreciating the clarity and relevance of our work, especially the clear motivation, novelty, and strength of our multi-variate parallel attention (MVPA).
>
> As suggested by the Reviewer, we now make use of the additional space to include further ablations in the main body of the paper and to better introduce the motivations behind our classical time-series benchmarks.
>
> ---
>
> ## W1. Foundation model
>
> The Reviewer raises a very relevant point: while MVPFormer is indeed a unimodal foundation model (much like many current language-only LLMs, such as GPT-OSS), the potential for multi-modality in the field of time-series is large. MVPFormer is trained following the prototypical generative foundation model scheme, with vast amounts of data and no task-specific objective, requiring only little finetuning to generalize to different tasks. In particular, we showcase zero-shot generalization to unseen patients (albeit, as correctly stated, with finetuning) in the seizure detection task, and generalization to four more neural discrimination tasks on a different dataset (Brain TreeBank). Therefore, MVPFormer is built to be a foundation model for iEEG. At the same time, recent work [1] has shown that cross-training between iEEG and EEG modalities might be highly beneficial; therefore, a multimodal foundation model is a research area we are actively investigating.
>
> Moreover, we show that the core mechanism of MVPFormer, our multi-variate parallel attention (MVPA), generalizes seamlessly to classical time-series benchmarks as well, suggesting a wide range of applicable scenarios.
>
> We now clarify in Section 3 MVPFormer that MVPFormer is a unimodal foundation model.
>
> ---
>
> ## W2. Zero-shot classification
>
> As mentioned above, the Reviewer is indeed correct in that seizure detection requires finetuning of the base model. Zero-shot only applies in the context of cross-patient evaluation, rather than cross-task. We now strengthen this distinction in Section 1 Introduction and Section 5.2 iEEG tasks.
>
> ---
>
> ## W3. General time-series tasks
>
> Following the Reviewer's suggestion, we now place the general time-series results in their own dedicated Section 6 Evaluation on general time-series. This better introduces their relevance to validating the design of MVPA in a manner that is agnostic to the specific clinical task, highlighting its potential for wider applicability.
>
> ---
>
> ## W4. MVPA ablation
>
> We present the main ablations of MVPA against vanilla attention in two Sections of the main body:
> - Section 5: comparisons in the iEEG tasks against MV-Llama, which is an identical model to MVPFormer with MVPA replaced by standard self-attention
> - Section 6: comparisons in the traditional time-series tasks against the vanilla Transformer
>
> Following the Reviewer's and Reviewer cFg6's suggestions, we now also ablate each attention component independently in the general time-series setup. The results are reported in Section 6.3 Ablation of the three components (and also below for ease of reading). They indicate that all three components are necessary to achieve the full performance of MVPA.
>
> |  | MVPA | Content-only | Time-only | Channel-only | None |
> |---|---|---|---|---|---|
> |  | MSE ↓ | MSE ↓ | MSE ↓ | MSE ↓ | MSE ↓ |
> | ETTh1 | **0.45** | 0.46 | 0.46 | 0.46 | 0.47 |
> | ETTh2 | **0.38** | 0.38 | 0.38 | 0.38 | 0.40 |
> | Weather | **0.25** | 0.27 | 0.27 | 0.26 | 0.27 |
>
> Interestingly, the performance drop between MVPA and the ablated variants is greater for the Weather dataset, which has 21 channels against the 7 of ETTh1 and ETTh1, giving further indication that the combination of the components gives MVPA more powerful processing of strongly multi-variate time-series.
>
> ---
>
> ## Q1. Foundation model
>
> As discussed in W1, MVPFormer is designed as a foundation model by virtue of its task-agnostic generative pre-training objective, which is then capable of generalizing to novel tasks by finetuning. We show that MVPFormer achieves state-of-the-art performance on five tasks across four different datasets, showcasing strong results across a variety of scenarios. This paradigm is standard both in medical [2] and language [3] models, as it often yields the most competitive results.
>
> We also wish to highlight that we are actively contributing to the research of multi-modal medical models with the public release of our Long-term iEEG dataset, the first iEEG dataset large enough for training foundation models.
>
> [1] Carzaniga, Francesco S., et al. "The Case for Cleaner Biosignals: High-fidelity Neural Compressor Enables Transfer from Cleaner iEEG to Noisier EEG." ICLR (2025).
>
> [2] Yuan, Zhizhang, et al. "Brant-2: Foundation Model for Brain Signals" arXiv preprint arXiv:2402.1025 (2024).
>
> [3] Hu, Edward J., et al. "Lora: Low-rank adaptation of large language models." ICLR (2022).

---

> > ### Author Response · Authors · 2025-11-28
> >
> > Thank you once again for your valuable feedback and dedicated service as a reviewer. We have carefully considered your input and made significant updates to the manuscript in response. We genuinely value your perspective and encourage you to review our rebuttal and the revised manuscript. Given the positive feedback from other reviewers, we hope our revisions address your concerns and provide the clarity needed to reconsider your assessment. Please do not hesitate to reach out if further clarification or discussion would be helpful.

---

### Author Response · Authors · 2025-11-21

We are deeply grateful to all Reviewers for their dedicated service and for providing thoughtful and interesting feedback.

We have taken all the reviews to heart and have updated the manuscript to reflect all the changes made following the Reviewers' suggestions. In addition to the individual responses, we wish to highlight here the main improvements to our work.

As suggested by Reviewers oKKd and cFg6, we have added ablations of all three components of our multi-variate parallel attention (MVPA) on the 3 general time-series forecasting tasks.

|  | MVPA | Content-only | Time-only | Channel-only | None |
|---|---|---|---|---|---|
|  | MSE ↓ | MSE ↓ | MSE ↓ | MSE ↓ | MSE ↓ |
| ETTh1 | **0.45** | 0.46 | 0.46 | 0.46 | 0.47 |
| ETTh2 | **0.38** | 0.38 | 0.38 | 0.38 | 0.40 |
| Weather | **0.25** | 0.27 | 0.27 | 0.26 | 0.27 |

The results show that **all three components are necessary to achieve the full performance of MVPA**, and that the difference is more pronounced when the signal is more strongly multi-variate (21 channels of Weather vs 7 channels of ETTh1/2).

As suggested by Reviewer cFg6, we have added an automated channel selection step, producing a fully automated end-to-end evaluation pipeline that ensures maximum fairness in comparisons.

| Model | Attention | Kappa ↑ | f1 ↑ | sens ↑ | fp/h ↓ |
|---|---|---|---|---|---|
| MVPFormer | MVPA | 0.61 | 0.59 | 0.72 | 0.15 |
| MVPFormer-S | MVPA | 0.57 | 0.53 | 0.71 | 0.12 |
| Brant-2 | Vanilla | 0.06 | 0.01 | 0.01 | 0.11 |
| BrainBERT | Vanilla | 0.00 | 0.00 | 0.00 | 0.00 |
| MV-Llama | Vanilla | 0.11 | 0.01 | 0.01 | 0.02 |

Remarkably, the introduction of **this pipeline increases the performance of our models to 0.61 and 0.57 Kappa scores**, respectively. We also follow Reviewer cFg6's suggestion and include 95% confidence intervals in all our seizure detection results on the Long-term iEEG dataset, strengthening the statistical validity of our evaluation.

We also implement Reviewer dxZg's suggestion to evaluate MVPFormer's resilience to noise, which is a fundamental characteristic in potentially noisy clinical environments.

|          | Episodic  |       |       |       | Raw    |
| :------- | :-------- | :---- | :---- | :---- | :---- |
| SNR      | Kappa ↑ | f1 ↑    | sens ↑  | fp/h ↓  | f1 ↑    |
| None     | 0\.61     | 0\.59 | 0\.72 | 0\.15 | 0\.51 |
| 60dB     | 0.58         | 0.54     | 0.71     | 0.12     | 0.49    |
| 50dB     | 0\.54     | 0\.50 | 0\.72 | 0\.13 | 0\.47 |
| 40dB     | 0\.36     | 0\.34 | 0\.74 | 0\.46 | 0\.31 |
| 30dB | 0\.12     | 0\.12 | 0\.71 | 1\.22 | 0\.10 |

Once more, the results show that **our model is designed with clinical relevance in mind, as it maintains expert-level performance up to 50dB SNR**.

In the context of this review period, we have also updated our runtime and memory usage baselines to include other attention variants, such as linear attention and Flash Attention. We have reworked all relevant aspects of the manuscript as suggested by the Reviewers, and are convinced that our work is now better and clearer thanks to all the feedback.

We remain available to address any further concerns and are looking forward to engage in interesting discussions with all the Reviewers.

---

### Author Response · Authors · 2025-12-03

We wish to thank all Reviewers, Area Chairs, and the whole organizing Committee for their work and dedication despite the difficult circumstances. We are motivated by the positive feedback we have received and are glad to have engaged in the initial fruitful discussion with the Reviewers. We are glad that Reviewers recognized and appreciated the novelty of our multi-variate parallel attention (MVPA), and that Reviewers cFg6, qWbc, and dxZg highlighted the thoroughness and cohesiveness of our evaluation pipeline. Further, we are confident that our Long-term iEEG dataset will be a notable contribution to the community, as underlined by Reviewers cFg6, qWbc, and dxZg.

We believe our work has been much improved during this rebuttal period, and we summarize here the main modifications, highlighting how they address the Reviewers' comments.

---

## 1. Fully automated evaluation pipeline

_Reviewer oKKd, Reviewer cFg6_

**Feedback.** Reviewer cFg6 suggests we perform a fully end-to-end automated evaluation, including channel selection. Reviewer oKKd also points out that our pipeline is zero-shot in the subjects, rather than the task.

**Response.** We have added an automated channel selection step using entropy and kurtosis statistics. Now the pipeline is fully automated and can be entirely reproduced with our code and data, addressing any concerns regarding finetuning and zero-shot performance.

**Results.** The new evaluation pipeline increases our Kappa score in the seizure detection task from **0.57 to 0.61**, further highlighting the adaptability of MVPA and improving on our previous state-of-the-art result (the best non-MVPA model reaches 0.11 Kappa score).

---

## 2. Performance on generic time-series tasks

_Reviewer oKKd, Reviewer dxZg_

**Feedback.** Reviewer oKKd and Reviewer dxZg note that our results on the generic time-series tasks warrant more thorough introduction.

**Response.** We have moved the results to their own Section 6 Evaluation on general time-series, to ensure they are clearly highlighted as they provide strong evidence of the performance of MVPA outside of its design domain of human electrophysiological data.

---

## 3. Ablation of the three components of MVPA

_Reviewer oKKd, Reviewer cFg6_

**Feedback.** Reviewer oKKd and Reviewer cFg6 suggest to ablate the three components of MVPA, to better isolate their contribution to the overall performance.

**Response.** We ablate the three components (content, channel, and time) of MVPA in the classical time-series forecasting tasks, using the same setup as previous results.

**Results.** We show that **all three components are necessary to achieve the full performance of MVPA**, and that the difference is more pronounced when the signal is more strongly multi-variate (21 channels of Weather vs 7 channels of ETTh1/2).

---

## 4. Detailed statistical results

_Reviewer cFg6_

**Feedback.** Reviewer cFg6 suggests we provide confidence intervals and detailed head-to-head results for each subject in the seizure detection task.

**Response.** We add the 95% confidence intervals on the Kappa scores to all our evaluations, adding it to the detail subject-by-subject breakdown we had already reported in Appendix G Additional results. The results for MVPFormer and all other tested models can now be compared head-to-head on a variety of key statistics, including: Kappa score, F1-score, sensitivity, confidence intervals, and more.

---

## 5. Runtime and memory profiling

_Reviewer cFg6, Reviewer dxZg_

**Feedback.** Reviewer cFg6 requests runtime and memory usage curves for MVPA, FlashMVPA, and other alternatives, and Reviewer dxZg also suggests benchmarking against simpler models.

**Response.** We add to the figures we previously reported in Appendix A Details on multi-variate parallel attention (MVPA) comparisons with vanilla attention, FlashAttention 2, and a linear attention variant (Nystromformer). All results were collected using reproducible scripts with fixed seeds, multiple runs, and all other common techniques to reduce the randomness of the results.

**Results.** FlashMVPA enjoys very favourable scaling against all alternatives, comparable to FlashAttention 2, while also including information about the structure of the time-series at the attention level. In particular, our results reinforce the sub-quadratic nature of our optimized implementation.

---

> ### Author Response · Authors · 2025-12-03
>
> ## 6. Additional related works
>
> _Reviewer qWbc_
>
> **Feedback.** Reviewer qWbc indicates further papers that are relevant to our work, including approaches for spatio-temporal data.
>
> **Response.** We add the suggested papers to Section 7 Related works, and expand Appendix A1. Comparison with alternative attention mechanisms to include them as well.
>
> ---
>
> ## 7. Resilience to noise
>
> _Reviewer dxZg_
>
> **Feedback.** Reviewer dxZg recommends us to evaluate the effect of signal noise on MVPFormer, to further validate its clinical viability.
>
> **Response.** We previously provided results on the seizure detection task on our Long-term iEEG dataset (high SNR) and the MAYO and FNUSA datasets (low SNR). Now, we systematically evaluate the performance of MVPFormer as noise in the recording increases from 30dB to 60dB, to simulate higher level of disturbance that mimic sub-optimal clinical environments.
>
> **Results.** MVPFormer maintains expert-level performance (0.54 Kappa score) up to 50dB SNR, showing that our model is designed with clinical relevance in mind.

---

### Meta-Review · Area_Chair_M8XX · 2026-01-01

**Summary:**

This paper introduces a foundation model for iEEG trained on the largest publicly available dataset, which makes a strong combined contribution in methodology, scale, and open resources. Initial reviewer concerns regarding evaluation fairness, missing ablations, statistical rigour, computational efficiency, and ethics were convincingly addressed in the rebuttal. Following authors' revisions, the method is well-validated, technically sound, and of clear value to both the neuroscience and time-series/AI communities.

**Reviewer Concerns:**

The authors adequately addressed the major technical concerns raised by reviewers. In particular, they resolved issues regarding evaluation issues by introducing a fully automated and standardised end-to-end pipeline with consistent preprocessing and channel selection across baselines, as well as additional results support.  But direct human-to-human comparisons on the same dataset are still not clear.

**Reviewer Scores:**

Overall, post-discussion from reviewers and authors rebutall and revisions would converge toward a consensus acceptance.

---

### Decision · Program_Chairs · 2026-01-26

Accept (Poster)